**Subject Category:**
Biology (whole organism)

ecology/environmental science/biogeography

hydrothermal vents, chemosynthetic ecosystem, resurgent cone, stable isotopes, Gammaproteobacteria, biogeography

**Author for correspondence:**
Katrin Linse
e-mail kl@bas.ac.uk

†Present address: Biodiversity, Evolution and Adaptation, British Antarctic Survey, High Cross, Madingley Road, Cambridge CB3 0ET, UK.
‡Present address: Department of Applied Sciences, Faculty of Health and Life Sciences, University of Northumbria, Ellison Building, Newcastle Upon Tyne NE1 8ST, UK.
¶Present address: REV Ocean, Oksenøyveien 10, 1366 Lysaker, Norway.
‖Present address: College of Marine Science, University of South Florida, 140 7th Avenue South, St Petersburg, FL 33701, US.
§Present address: Bundeswehr Institute of Microbiology, Neuherbergstr 11, 80937 Munich, Germany.

# Fauna of the Kemp Caldera and its upper bathyal hydrothermal vents (South Sandwich Arc, Antarctica)

Katrin Linse[1,†], Jonathan T. Copley[2], Douglas P. Connelly[3], Robert D. Larter[1], David A. Pearce[1,‡], Nick V. C. Polunin[4], Alex D. Rogers[5,¶], Chong Chen[6], Andrew Clarke[1], Adrian G. Glover[7], Alastair G. C. Graham[1,‖], Veerle A. I. Huvenne[3], Leigh Marsh[2], William D. K. Reid[4], C. Nicolai Roterman[5], Christopher J. Sweeting[4], Katrin Zwirglmaier[1,§] and Paul A. Tyler[2]

[1]British Antarctic Survey, High Cross, Madingley Road, Cambridge CB3 0ET, UK
[2]Ocean and Earth Science, University of Southampton, Waterfront Campus, Southampton SO14 3ZH, UK
[3]National Oceanography Centre, European Way, Southampton SO14 3ZH, UK
[4]School of Natural and Environmental Sciences, Newcastle University, Ridley Building, Newcastle upon Tyne NE1 7RU, UK
[5]Department of Zoology, University of Oxford, South Parks Road, Oxford OX1 3PS, UK
[6]X-STAR, Japan Agency for Marine-Earth Science and Technology (JAMSTEC), 2-15 Natsushima-cho, Yokosuka 237-0061, Kanagawa Pref. Japan
[7]Life Sciences Department, Natural History Museum, Cromwell Road, London SW7 5BD, UK

KL, 0000-0003-3477-3047; JTC, 0000-0003-3333-4325; DPC, 0000-0002-8353-116X; RDL, 0000-0002-8414-7389; NVCP, 0000-0002-1480-8794; CC, 0000-0002-5035-4021; AGG, 0000-0002-9489-074X; AGCG, 0000-0002-2880-2908; VAIH, 0000-0001-7135-6360; LM, 0000-0002-4613-1538; WDKR, 0000-0003-0190-0425; CNR, 0000-0002-6636-6078; KZ, 0000-0001-8598-9572

Faunal assemblages at hydrothermal vents associated with island-arc volcanism are less well known than those at vents on mid-ocean ridges and back-arc spreading centres. This study characterizes chemosynthetic biotopes at active hydrothermal vents discovered at the Kemp Caldera in the South Sandwich Arc. The caldera hosts sulfur and anhydrite vent chimneys in 1375–1487 m depth, which emit sulfide-rich fluids with temperatures up to 212°C, and the microbial community of

water samples in the buoyant plume rising from the vents was dominated by sulfur-oxidizing Gammaproteobacteria. A total of 12 macro- and megafaunal taxa depending on hydrothermal activity were collected in these biotopes, of which seven species were known from the East Scotia Ridge (ESR) vents and three species from vents outside the Southern Ocean. Faunal assemblages were dominated by large vesicomyid clams, actinostolid anemones, *Sericosura* sea spiders and lepetodrilid and cocculinid limpets, but several taxa abundant at nearby ESR hydrothermal vents were rare such as the stalked barnacle *Neolepas scotiaensis*. Multivariate analysis of fauna at Kemp Caldera and vents in neighbouring areas indicated that the Kemp Caldera is most similar to vent fields in the previously established Southern Ocean vent biogeographic province, showing that the species composition at island-arc hydrothermal vents can be distinct from nearby seafloor-spreading systems. $\delta^{13}$C and $\delta^{15}$N isotope values of megafaunal species analysed from the Kemp Caldera were similar to those of the same or related species at other vent fields, but none of the fauna sampled at Kemp Caldera had $\delta^{13}$C values, indicating nutritional dependence on Epsilonproteobacteria, unlike fauna at other island-arc hydrothermal vents.

## 1. Introduction

The presence of active volcanoes on the Antarctic continent and several islands in the Southern Ocean was recorded by early polar explorers, such as Sir James Clark Ross who observed an eruption of Mount Erebus in 1841 [1]. These clearly visible volcanoes have been the subject of geological and biological research for a long time (e.g. [2,3]). The presence of volcanic and measurable tectonic activity, such as eruptions, steam fields, fumaroles and earthquakes on the Antarctic Peninsula, Marie Byrd Land, Victoria Land, the islands of the South Shetlands and South Sandwich Arc and their surrounding seas (e.g. [2,4]), has implied the presence of sub-ice and submarine volcanism and hydrothermal activity [5–8].

In the Antarctic and Southern Ocean marine environment, shallow-water hydrothermal fumaroles occur in the caldera of Deception Island and are characterized by depleted marine life with low species richness and faunal abundance compared to other Antarctic shallows (e.g. [9–11]). Hot vents and cold seeps have been discovered in the Bransfield Strait in 990–1500 m depth [12–17], in the Larsen B ice shelf area, eastern Antarctic Peninsula, in 215–850 m water depth [18–20], and on the shelf of South Georgia in 250–350 m water depth [21,22], but the fauna associated with these locations mostly comprise elements of typical Antarctic and Southern Ocean shelf fauna, with two exceptions associated with chemosynthetic environments including the siboglinid polychaete *Sclerolinum contortum* at Hook Ridge, Bransfield Strait [23] and a large vesicomyid clam at Larsen B [22,24–26].

Hydrothermal vent fields with abundant novel fauna have been discovered on the back-arc spreading centre of the East Scotia Ridge (ESR) in 2400–2600 m depth, and on the Australian Antarctic Ridge (AAR) in 1800–1900 m depth [27,28]. High-temperature 'black-smoker' venting occurs at two vent fields, on ESR segments E2 and E9 [27,29,30], which are inhabited by vent-specific species including the yeti crab *Kiwa tyleri*, the gastropods *Gigantopelta chessoia* and *Lepetodrilus concentricus*, the stalked barnacle *Neolepas scotiaensis*, the seven-armed sea star *Paulasterias tyleri* and undescribed actinostolid anemones [31–36], in assemblages supported by chemoautotrophic carbon fixation [37–40]. To date, the AAR sites have not been visited and imaged by remotely operated vehicles (ROVs), but specimens of associated vent fauna *Kiwa araonae* and *Paulasterias tyleri* have been collected, indicating the presence of faunal assemblages supported by chemosynthesis [28,41].

Situated on the Sandwich Plate, the South Sandwich Arc of the Southern Ocean comprises actively erupting volcanic South Sandwich Islands and their associated seamounts [42,43]. The minor tectonic Sandwich Plate is separated in the west from the Scotia Plate by the ESR, a back-arc spreading centre formed by the subduction of South American Plate on the eastern margin of the microplate, which also forms the South Sandwich Trench [42,44,45]. To the south, the Sandwich Plate is bordered by the Antarctic Plate [42]. The active volcanic arc ranges from northern Protector Shoal seamounts via eleven volcanic islands to seamounts in the south, one of the southernmost being the Kemp Seamount [42]. Submarine hydrothermal vents have been reported from this island arc at Adventure Crater [46], Protector Shoal seamounts and Quest Caldera [47], and Kemp Caldera near the Kemp Seamount [29,48], which is the focus of this study. The Kemp Caldera is situated within a restricted zone of the South Georgia and the South Sandwich Islands Marine Protected Area (SGSSI MPA), where both research and fisheries activities are regulated by the SGSSI government [49]. In the SGSSI MPA, commercial benthic bottom trawling is banned and longline fisheries are restricted to depths between

700 and 2250 m (www.gov.gs). The SGSSI government is in the process of enacting legislation to protect the environment of the SGSSI MPA from any future mining or hydrocarbon activity (exploration and exploitation) to align with the Madrid Protocol of the Antarctic Treaty, and elucidating the faunal assemblages associated with hydrothermal vents in the SGSSI MPA informs that legislative effort.

The ecology of deep-sea hydrothermal vents associated with island-arc volcanism is less well known compared to vents on mid-ocean ridges and back-arc spreading centres, despite island-arcs extending over a total distance of 22 000 km in the oceans [50,51]. Island-arc hydrothermal vents in the Pacific and Caribbean exhibit contrasting vent fluid geochemistry to hydrothermal systems on seafloor spreading centres. Hydrothermal vents in the Mariana arc produce fluids with low pH, high $CO_2$ and low $H_2S$ concentrations compared to 'black smoker' vents on mid-ocean ridges [51]. At vents on the Loihi Seamount of the Hawaiian archipelago, low concentrations of $H_2S$ favour microbial communities dominated by Fe-oxidizing bacteria rather than sulfide-oxidizers [52]. Hydrothermal vents on the Nafanua cone in the crater of Vailulu'u Seamount in the Samoan archipelago produce fluids with pH 2.7 and droplets of liquid $CO_2$; dead midwater metazoans are abundant on the seafloor around those vents, where an acid-tolerant polynoid polychaete feeds on bacteria colonizing animal carcasses [53]. A similar 'dead zone' of midwater shrimp has been observed around hydrothermal vents in the summit crater of Kick'em Jenny submarine volcano in the Lesser Antilles arc of the Caribbean [54], which also hosts a cold-seep ecosystem on a debris avalanche deposit on its western flank [55].

Environmental differences between island-arc hydrothermal systems and those on seafloor spreading centres may result in differences in their associated faunal assemblages, but few faunal assemblages have been characterized from island-arc vent systems, and none have been previously described from the Southern Ocean. Although some species found at ESR vents have also been reported from the Kemp Caldera, such as the limpet *Lepetodrilus concentricus*, the stalked barnacle *N. scotiaensis*, the sea star *P. tyleri* and the sea spiders *Sericosura* spp. [33–35,56], the faunal assemblage associated with this island-arc hydrothermal system has not yet been described in detail. The aims of this study are, therefore, to define the different biotopes in the hydrothermally active areas of the caldera, characterize the trophic ecology of the faunal assemblage and determine the levels of similarity between the fauna of this island-arc vent system and assemblages at hydrothermal vents on seafloor spreading centres in neighbouring regions.

We hypothesize that the chemosynthetic fauna present in the venting sites of the Kemp Caldera will be similar to that of the ESR segments E2 and E9, based on their relatively close distance of only approximately 440 km and approximately 90 km away, respectively.

## 2. Material and methods

Data for this study were collected during two expeditions at sea: expedition JR224 on board the *Royal Research Ship* (*RRS*) *James Clark Ross* in February 2009 [57] and expedition JC42 on board the *RRS James Cook* in January 2010 [58].

### 2.1. Hydro-acoustic data

Multibeam swathe bathymetry surveys of the caldera were conducted by the *RRS James Clark Ross*'s hull-mounted Kongsberg SIMRAD EM120 multibeam echo sounder during expedition JR224 and by the SIMRAD SM2000 high-resolution (200 kHz) multibeam echosounder mounted on ROV *Isis* during expedition JC42 [27,58,59]. ROV-mounted swathe data were processed using the IFREMER software package CARAIBES [59].

### 2.2. Water column and hydrothermal fluid sampling

On JC42, water column samples were collected using a titanium frame with 24 externally sprung Niskin bottles, specifically designed for sampling waters with low levels of trace metals and nutrients. The frame also included a Seabird + 911 CTD, a light scattering sensor (LSS) and a reductive potential (Eh) detector. The 10 l Niskin bottles were Teflon lined, with Teflon taps and non-metallic parts, any metallic components were constructed using titanium or high-quality stainless steel. Water samples were analysed for particulate, dissolved and soluble concentrations of metals and oxyanions [29] as well as microbial diversity (detailed below).

Hydrothermal fluid sampling was achieved using titanium (Ti) samplers on the ROV, equipped with an inductively coupled link (ICL) high-temperature sensor to ensure the collection of high-quality samples. In the case of diffuse flow fluid sampling or sampling of friable chimney structures, the Ti

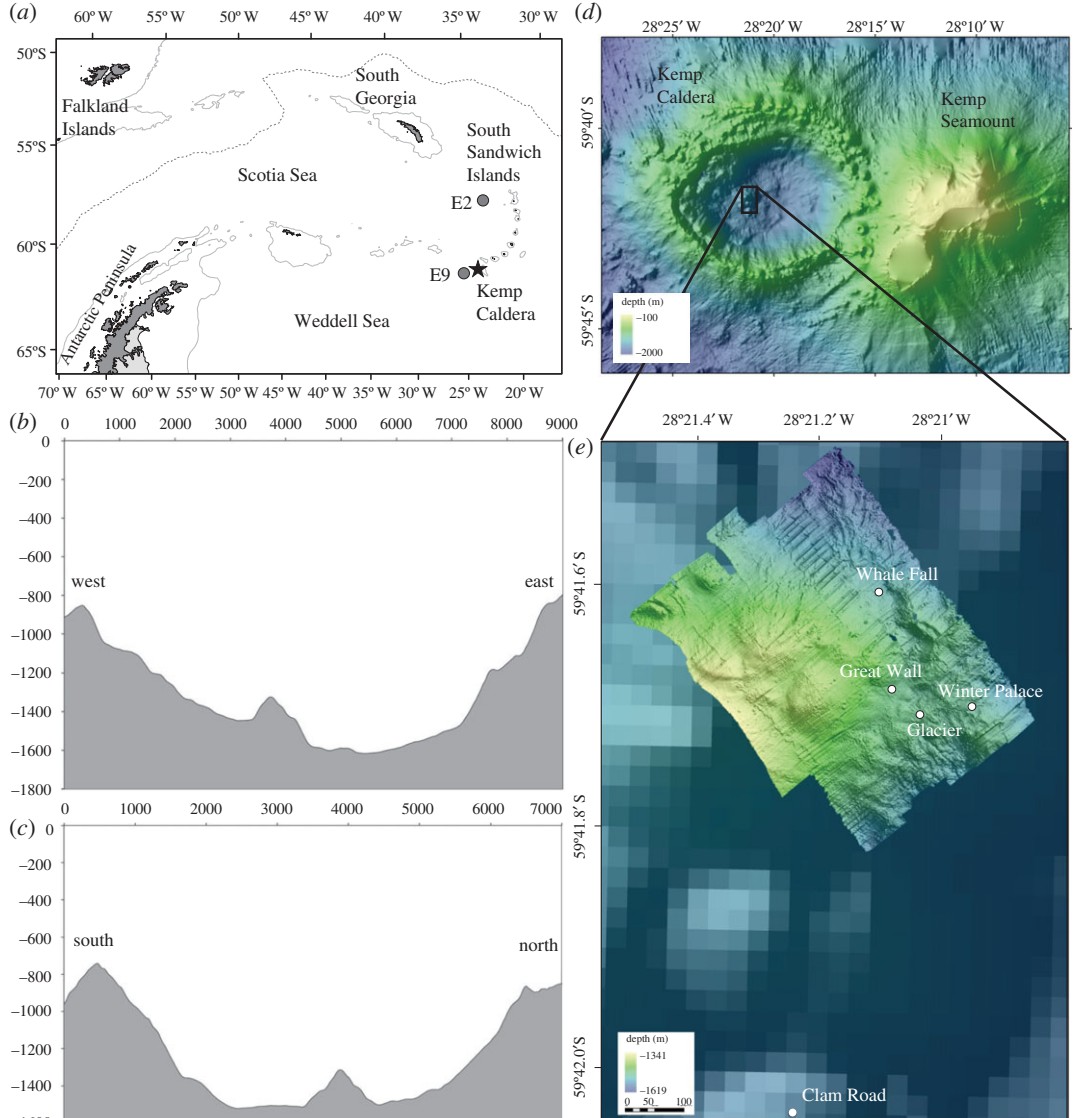

**Figure 1.** Map of the Southern Ocean; (*a*) location of Kemp Caldera in the Southern Ocean; (*b*) bathymetric profile, west to east through the Kemp Caldera; (*c*) bathymetric profile, south to north through the Kemp Caldera; (*d*) map of Kemp Seamount and the Kemp Caldera showing the resurgent cone; (*e*) sampling sites in the vicinity of the resurgent cone in the Kemp Caldera.

samplers were used in conjunction with a specially constructed Ti diffuse sampler, which was used to prevent entrainment of surrounding seawater into the path of the fluid during sampling [27].

## 2.3. Seabed imagery

During expedition JR224, the towed seabed high-resolution imaging platform (SHRIMP) was deployed once in the caldera near the Kemp Seamount to collect video imagery of the seabed that enabled identification of megafaunal (greater than 3 cm) benthic invertebrates [57]. SHRIMP was equipped with three video-only recording cameras, a forward-looking Simrad PAL colour charge-coupled device (CCD) camera type OE1364, a downward-looking Bowtech PAL colour CCD camera type L3C-550 and a downward-looking three-chip Panasonic camcorder.

In 2010, during expedition JC42 the ROV *Isis* was deployed for 8 dives with a total of 118 h deployment time at the Kemp Caldera (figure 1; electronic supplementary material, table 1). For seafloor imagery (photos, video and frame grabs of videos) ROV *Isis* was equipped with a three-chip CCD video camera (Insite Pacific Atlas), a 1080i high-definition video camera (Insite Pacific Mini Zeus) on a pan-and-tilt-mount and a 3.3-megapixel stills camera (Insite Scorpio) [59]. Additionally, two lasers, 0.1 m apart, were mounted parallel to the focal axis of the video camera to provide scale in images. Footage from the video cameras was recorded on DVCAM tapes and DVD and from the still camera on a memory card [27].

## 2.4. Macro- and megafauna sample treatment

Benthic invertebrates were collected by either the ROV *Isis*'s suction sampler or scoop and brought to the surface in ambient seawater. Once on board, samples were immediately transferred to seawater in a temperature-controlled laboratory set to +4°C where individuals were dissected and either frozen or stored in molecular grade ethanol for molecular analysis, frozen for isotope analysis or fixed in 10% seawater formalin for morphological analysis.

### 2.4.1. Stable isotopes

For stable isotope analysis of the food web structure in and outside the venting areas in the Kemp Caldera, different tissues were used for different species: tube feet in the asteroids, tentacles in the sea anemones and foot muscle tissue in the vesicomyid clam. Pycnogonids were sampled whole while the cocculinid gastropods were removed from their shells and sampled whole. Faunal tissue samples were freeze-dried and ground to homogeneous powder using a pestle and mortar. Aliquots of fauna were tested for carbonates prior to analysis. The samples did not effervesce and therefore no acidification was carried out. Dual $\delta^{13}C$ and $\delta^{15}N$ were measured by elemental analysis–isotope ratio mass spectrometry (EA-IRMS) using a Roboprep-CN sample preparation module coupled to a Europa Scientific 20–20 IRMS on 1.0 mg of sample. $\delta^{34}S$ was measured using a SERCON elemental analyser coupled to a Europa Scientific 20–20 IRMS on 2 mg of sample with an additional 4 mg of vanadium pentoxide as a catalyst. All analyses were carried out by Iso-Analytical (Crewe, UK). Stable isotope ratios were expressed in delta ($\delta$) notation as parts per thousand (‰). An external reference material of freeze-dried and ground deep-sea fish white muscle (*Antimora rostrata*) was also analysed ($\delta^{13}C$, $n = 28$, $-18.82‰ \pm 0.10$ s.d.; $\delta^{15}N$, $n = 28$, $13.11‰ \pm 0.38$ s.d.; $\delta^{34}S$, $n = 7$, $18.56‰ \pm 0.44$ s.d.).

## 2.5. Multivariate analysis of faunal similarity with vent fields in neighbouring regions

The species inventory for the site was compared with species lists compiled from published literature for 15 well-studied vent fields in neighbouring regions: the ESR (E2 and E9 vent fields), the Indian Ocean (Longqi, Duanqiao, Tiancheng on the Southwest Indian Ridge; Kairei, Edmond and Solitaire fields on the Central Indian Ridge) and Mid-Atlantic Ridge (Lucky Strike, Rainbow, Broken Spur, TAG, Snake Pit, Ashadze-1 and Logatchev fields), updating the dataset previously published by Copley *et al*. [60] with subsequently published records of additional sites and taxa [61–64].

Meiofauna were not sampled at the Kemp Caldera during this study, and meiofaunal taxa were, therefore, excluded from species lists of other vent fields for comparison, following the protocol of Copley *et al*. [60], as their true absence cannot be inferred reliably from the literature for each vent field. 'Non-vent' taxa (defined as species originally described from non-chemosynthetic environments) were also excluded for the same reason, as such 'normal' deep-sea taxa on the periphery of vent fields are not consistently included in published species inventories of vent fields. The omission of these variably recorded groups, therefore, ensures equivalent datasets from each vent field for comparative analysis by only considering the presence/absence of macro- and megafaunal taxa considered to be endemic to chemosynthetic environments.

The identities of taxa were recorded to species level in the dataset where possible, and indeterminate species of the same genus at different sites were conservatively assigned to separate taxonomic units to avoid potential false conflation of faunal similarity. In total, the resulting database of vent fauna (presented in the electronic supplementary material) contains 329 records of 159 taxa across 16 vent fields. A similarity matrix between vent fields was calculated from taxon presence/absence records using Sørensen's Index. Hierarchical agglomerative clustering using group-average linkage, and non-metric multidimensional scaling (MDS) with a 5% proportion of the metric MDS solution to reduce sample-point collapse, were applied to the similarity matrix using PRIMER v. 7 to produce a dendrogram and two-dimensional ordination representing similarity relationships.

## 2.6. Microbiological sample treatment

For the analysis of the microbial biodiversity in the water column, 30 l of water from one sample within the buoyant vent plume, at a depth of 1355 m (about 60 m above the seafloor) taken with CTD Niskin bottles were filtered through a 0.2 µm pore size nitrocellulose filter (Whatman GE Healthcare). The water at the sampling depth showed a strong decrease in the redox potential of 342 mV with

concomitant increase in LSS of 0.538 compared to surface water. The filters were frozen at −80°C until further analysis. DNA was extracted from the filters using a phenol/chloroform protocol [42]. To test for the presence of endosymbiont in the vesicomyid clams, gill tissue was frozen at −80°C. DNA from gill tissue was extracted with phenol/chloroform as above, but using a longer Proteinase K digest (1 h at 37°C followed by 1 h at 50°C).

The near full-length 16S rRNA gene was amplified by PCR using the universal bacterial primers 27F [65] and 1492R [66] and universal archaeal primers ARCH46F [67] and UA1406R [68]. PCR conditions were 3 min at 94°C, followed by 30 cycles of 60 s at 94°C, 45 s at 50°C, 90 s at 72°C and a final elongation of 5 min at 72°C. PCR products were cloned into the pCR2.1 vector by TOPO TA cloning (Invitrogen-Life Technologies, USA), following the manufacturer's recommendations and plated on LB-Ampicillin plates containing X-gal for blue-white screening. White clones were checked for correct insert size by PCR using the plasmid primers M13F and M13R. A total of 134 clones from the filters and 131 clones from the clams' gill tissues were sequenced bi-directionally by Sanger sequencing at LGC Genomics (Berlin, Germany). The sequences were trimmed, QC checked and overlapping ends assembled with the software package Geneious (www.geneious.com) to obtain the near full-length 16S rRNA sequence and subsequently aligned to a reference database (SILVA 102 www.arb-silva.de) and phylogenetically identified within arb [69]. Sequences have been submitted to GenBank (accession numbers MK736312–MK736351).

For flow cytometry, 1.5 ml of five water samples taken with CTD Niskin bottles at different points within the vent plume of each water sample was fixed in 0.5% (v/v) glutaraldehyde and frozen at −80°C until analysis. Samples were stained in 0.001% (v/v) SYBR green I (Sigma) for 1 h before measurement in a FACScan flow cytometer (Becton Dickinson, Oxford, UK). Flow cytometry data were analysed with CytoWin [70].

# 3. Results

## 3.1. Study area

In 2009, multibeam bathymetry surveys in the vicinity of the Vysokaya Bank, southern South Sandwich Island volcanic arc, revealed a caldera to the west of the Kemp Seamount (59°42′ S, 28°20′ W) (figure 1). This previously unseen feature measures an E–W rim-to-rim width of 8300 m, a N–S width of 6500 m and a flat bottom floor width of approximately 2500 m. The sill depth is at 900 m while the inner floor depth is approximately 1600 m resulting in a difference of approximately 700 m from rim to floor. These dimensions indicated that the depression is a caldera (i.e. formed by collapse of the sea floor into a drained magma chamber; [71]), rather than a crater (formed as a direct result of an eruption), as the latter rarely exceed 2 km in diameter and typically have lower width/depth ratios [72,73]. The structure is referred to as 'Kemp Caldera' hereafter.

The 'fresh' morphology of the structure and its lack of sedimentary infill, demonstrated by seafloor observations of exposed rock surfaces and by the presence of blocky lavas at the caldera floor, suggest a particularly young age for the caldera. Further observations within and away from the caldera included the presence of slide blocks and mass movement structures (e.g. debris flow chutes) on the flanks of the caldera and seamount, which testified to the instability of slopes both here and elsewhere along the South Sandwich arc [45,74]. The rim of the caldera was marked by a heterogeneous topography featuring multiple cones and craters, indicating post-collapse activities, which commonly occurs around caldera rims.

The topography of the inner caldera floor was virtually flat apart from the presence of a mounded feature at the centre of the caldera, interpreted here as a resurgent cone. The cone had a neighbouring bank on the western side (figure 1b,c). The resurgent cone rose approximately 250 m from the caldera floor and is approximately 1000 m in diameter. On the SE base of the resurgent cone and on the NE flank of the bank in 1375–1487 m depth, hydrothermally active areas including venting chimneys and diffuse flow sites were discovered.

The deep-water oceanography around the Vysokaya Bank, including the Kemp Caldera, is characterized by the Weddell-Scotia Confluence [75] and two main water masses, the Circumpolar Deep Water (CDW) and the Weddell Sea Deep Water (WSDW) [76,77]. The CDW is driven by the Antarctic Circumpolar Current into the eastern Scotia Sea, while WSDW in the area originates either from it overflowing the South Scotia Ridge east of the Orkney Passage or entering through the Georgia Passage [76]. The waters within the Kemp Caldera show the characteristics of modified CDW.

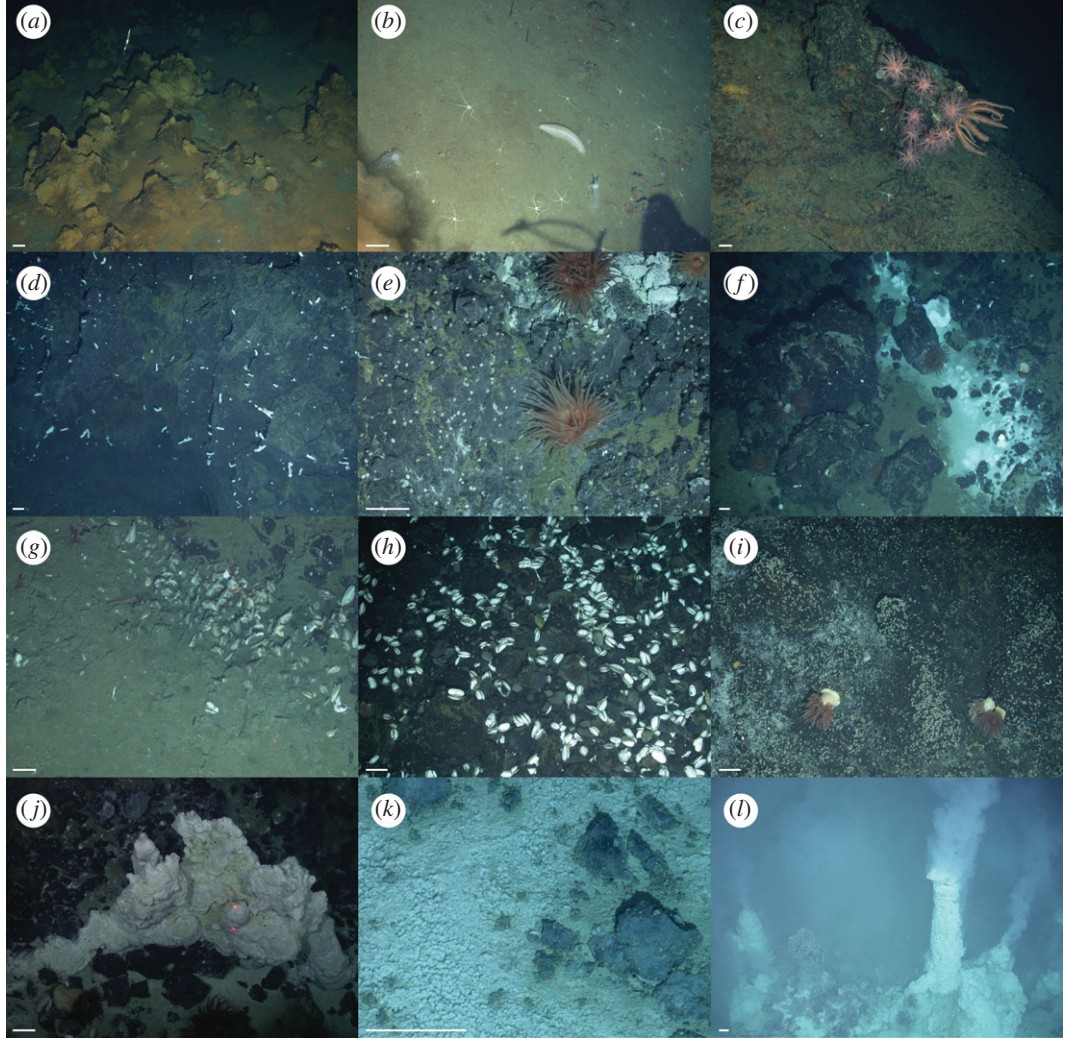

**Figure 2.** *In situ* Kemp Caldera biotopes; (*a*) ferrous sediments, 59°41.86 S, 28°21.53 W; (*b*) non-vent soft sediment with ophiuroids and holothuroids, 59°41.83 S, 28°21.52 W; (*c*) non-vent basalt with alcyonaceans and brisingid, 59°41.82 S, 28°21.51 W; (*d*) vent halo zone with halichondriids, 59°41.64 S, 28°21.13 W; (*e*) basalt in diffuse flow with actinostolid anemones, pycnogonid *Sericosura* and cocculinid limpet, 59°41.67 S, 28°21.10 W; (*f*) basalt next to 'Great Wall' with actinostolids, *Sericosura*, cocculinids and vesicomyid clams *Archivesica* s.l. *puertodeseadoi*, 59°41.69 S, 28°21.10 W; (*g*) 'Fine Sediment in Diffuse Flow' with *Archivesica* s.l. *puertodeseadoi*, *Sericosura*, and cocculinids, 59°41.66 S, 28°20.99 W; (*h*) 'Clam Road' with *Archivesica* s.l. *puertodeseadoi* and cocculinids, 59°42.03 S, 28°21.23 W; (*i*) 'Coarse Sediment in Diffuse Flow' with cocculinids, *Sericosura* and actonostilids, 59°42.05 S, 28°21.22 W; (*j*) 'Great Wall', sulfur structure with white bacterial mats, 59°41.68 S, 28°21.09 W; (*k*) 'Precipitated Sediment' with cocculinids and *Lepetodrilus concentricus*, 59°41.67 S, 28°21.14 W; (*l*) 'Winter Palace', chimneys covered in anhydrite and with *Lepetodrilus concentricus*, 59°41.69 S, 28°20.97 W. The white bar is approximately 10 cm.

## 3.2. Hydrothermal vent biotopes and their fauna

Fauna inside the Kemp Caldera was studied by analysing underwater imagery and samples collected from specific faunal assemblages, and their habitats are described as follows. The focus was on the benthic fauna of the identified hydrothermally actives areas, characterized by the presence of 'white smoking' venting, diffuse flow, precipitates and/or bacterial mats, while an overview is given on the observed benthic and pelagic fauna from surrounding non-venting environments. The video analyses defined eight assemblage types, classified as biotopes, under the influence of hydrothermal activities based either on the dominating macro- and megafauna or on the type of substrate (figures 2 and 3, tables 1 and 2). A further chemosynthetic biotope discovered in the Kemp Caldera adjacent to the resurgent cone, the decomposing skeleton of an Antarctic minke whale, has been described by Amon *et al*. [78].

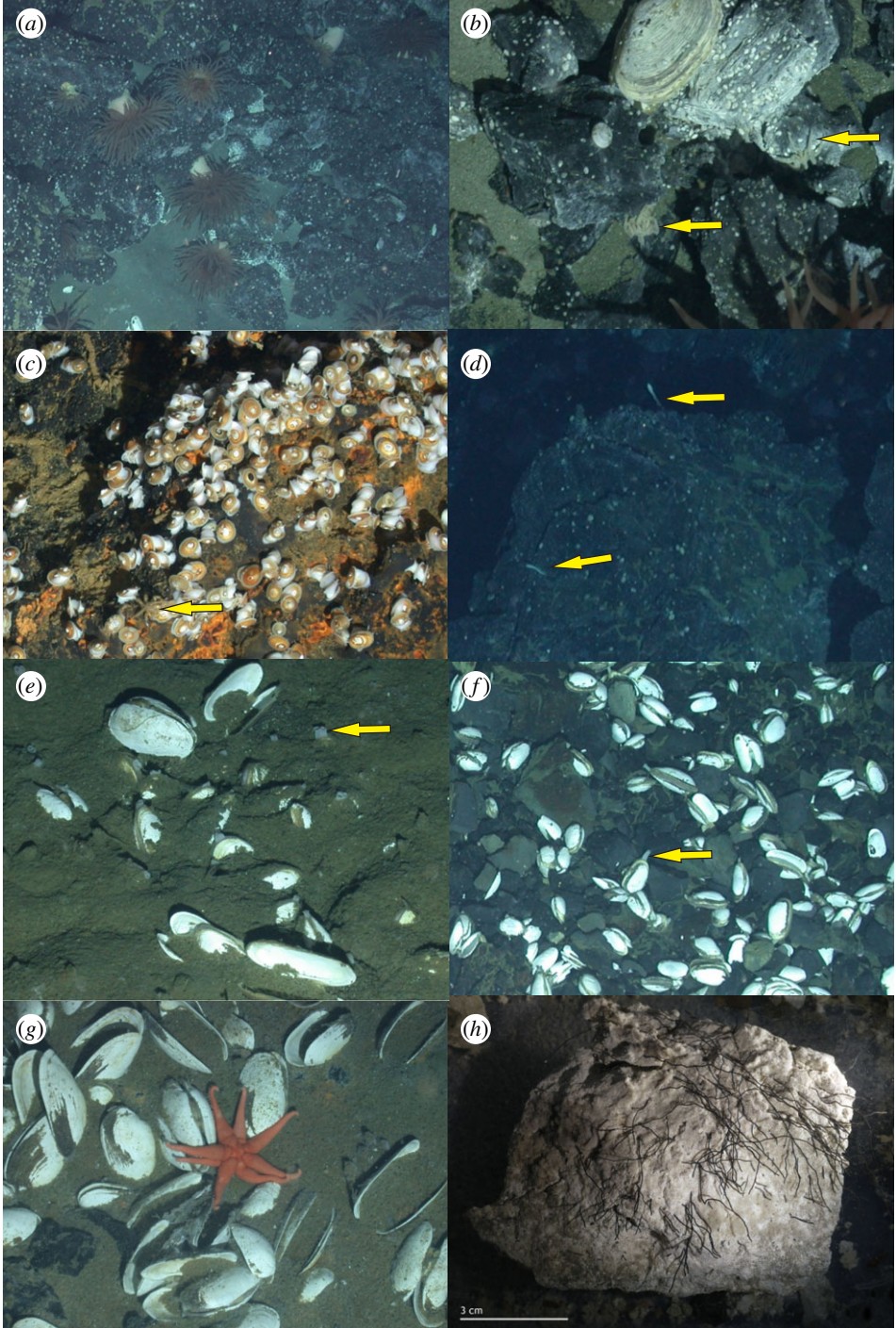

**Figure 3.** *In situ* Kemp Caldera chemosynthetic fauna; (*a*) actinostolid sp. and cocculinid limpet at 'Clam Road', 59°42.05 S, 28° 21.23 W; (*b*) *Sericosura bamberi* (yellow arrow), cocculinid limpet, *Lepetodrilus concentricus* at 'Basalt next to Great Wall', 59°41.67 S, 28°21.09 W; (*c*) cocculinid limpet and *Sericosura bamberi* (yellow arrow) at 'Coarse Sediment in Diffuse Flow', 59°42.06 S, 28° 21.24 W; (*d*) *Neolepas scotiaensis* (yellow arrow) at 'Basalt in Diffuse Flow', 59°42.66 S, 28°20.98 W; (*e*) burrowed, live *Archivesica* s.l. *puertodeseadoi* with siphons visible (yellow arrow) at 'Fine Sediment in Diffuse Flow', 59°41.67 S, 28°21.01 W; (*f*) epilithically living *Archivesica* s.l. *puertodeseadoi* with siphons visible (yellow arrow) at 'Clam Road', 59°42.02 S, 28°21.23 W; (*g*) *Paulasterias tyleri* at 'Fine Sediment in Diffuse Flow', 59°41.71 S, 28°21.07 W; (*h*) *Sclerolinum* sp. at 'Precipitated Sediment', 59°41.68 S, 28°20.99 W.

### 3.2.1. Descriptions of the Kemp Biotopes

#### 3.2.1.1. Great Wall (GW)

'Great Wall' (figure 2*j*) was defined by sulfurous structures, which include pure bright yellow crystalline sulfur, emerging from the seafloor, which hosted no macrofauna. The maximum temperature measured

at 'GW' was 21°C. The macrofauna seen in close proximity to the biotope 'GW' belonged to assemblages associated with basalt rocks and boulders.

### 3.2.1.2. Winter Palace (WP)
This biotope was characterized by an area of dense 'white smoking' sulfide chimneys covered in bacterial mats (figure 2*l*) and emitting fluids with temperatures ranging from 103 to 212°C [48]. Pelagic animals, including squid or shrimp, swimming through the white smoke were observed to change colour from red to white and then drop to the seafloor. The only macro- or megafaunal species collected from chimney samples and observed living on the chimney surfaces was the vent limpet *Lepetodrilus concentricus* (figure 3*b*). The shell of limpets collected from these chimneys were covered in anhydrite and the limpets left what appeared to be 'home scars' around their sitting positions on the chimneys.

### 3.2.1.3. Basalt next to Great Wall (BGW)
Basalt outcrops, including rocks and boulders, and the soft sediment between them (figure 2*f*), next to the 'GW' structure, were characterized by the rare presence of large, infaunal specimens of the vesicomyid clam *Archivesica* s.l. *puertodeseadoi* (figure 3*e*), occasional large, dark red actinostolid anemones (figure 3*a*) and *Sericosura* spp. sea spiders (figure 3*b,c*) together with numerous individuals of *L. concentricus* and a cocculinid limpet (figure 3*c*). A single specimen of the stalked barnacle *Neolepas scotiaensis* (figure 3*d*) was seen in this biotope.

### 3.2.1.4. Basalt in Diffuse Flow (BDF)
Rigid and continuous basalts on the slope of the resurgent cone were associated with diffuse hydrothermal flow and hosted an assemblage of abundant macrofauna (figure 2*e*). The cocculinid limpet and *Lepetodrilus conentricus* both occurred at very high abundances and both actinostolid anemones and *Sericosura* spp. were frequent. Occasionally, halichondriid sponges (figure 2*d*) were seen and several lone specimens of *Neolepas scotiaensis* were present. Towards the outer edges of this biotope, very occasional non-vent-associated fauna, such as the sessile holothurians (genus *Psolus*) and cnidarians (genus *Anthomastus*), were recorded.

### 3.2.1.5. Clam Road (CR)
The bank just south of the resurgent cone hosted the 'Clam Road' biotope (figure 2*h*), characterized by rough-edged basalt and frequent vesicomyid clams living epilithically. The most common species were the cocculinid limpet, *Lepetodrilus concentricus*, and pycnogonids of the genus *Sericosura*, with actinostolid anemones and halichondriid sponges being occasionally present. Within the finer sediment rubble collected by the suction sampler, several species of polychaetes belonging to several families were present (table 1).

### 3.2.1.6. Precipitated Sediment (PS)
Near the 'Winter Palace' biotope, large areas of flat, sedimented seafloors were covered in white precipitate with the occasional basalt rock penetrating the surface (figure 2*k*). A characteristic area for this biotope is the 'Glacier' locality (figure 1*e*). *Lepetodrilus concentricus* and *Sericosura* spp. sea spiders were seen on dense white precipitate, while the vesicomyid clam *Archivesica* s.l. *puertodeseadoi* was frequently seen burrowed in areas with thinner precipitate cover. Collections in this area revealed the presence of thyasirid bivalves and several polychaete species. Specimens of the actinostolid anemone and the cocculinid limpet were less abundant in this biotope and mostly found on the basalt outcrops. The sediments taken, coated in pale bacterial mats, in this biotope had a strong sulfur odour and contained tubes of *Sclerolinum* worms (figure 3*h*). Some of the precipitate-covered seafloor areas formed 'dead zones' in which numerous dead *Nematocarcinus* shrimps and squids were seen lying on the white precipitate. *Sericosura* spp. sea spiders were seen in clusters over individual dead shrimps and were presumed to be feeding on them.

### 3.2.1.7. Coarse Sediment in Diffuse Flow (CSDF)
The top of the bank south of the resurgent cone was influenced by diffuse venting and covered by coarse-grained anhydrite aggregates and basalt fragments (figure 2*i*), which led to the location name 'Ash Mount'. The macrofauna there was dominated by very high abundances of *Sericosura* spp. and the

**Table 1.** Species presence in biotopes of the Kemp Caldera. Abundance: 0, absent; 1, rare; 2, occasional; 3, frequent; 4, common; x, present. *, records only from hydrothermal sites. WP, Winter Palace; GW, Great Wall; PS, Precipitated Sediment; BGW, Basalt next to Great Wall; BDF, Basalt in Diffuse Flow; CR, Clam Road; CSDF, Coarse Sediment in Diffuse Flow; FSDF, Fine Sediment in Diffuse Flow; BWM, Basalt covered in white mat.

| | species | WP | GW | PS | BGW | BDF | CR | CSDF | FSDF | BWM |
|---|---|---|---|---|---|---|---|---|---|---|
| Porifera | | | | | | | | | | |
| | Halichondriidae sp.* | | | | | 2 | 2 | | 2 | 4 |
| Cnidaria | | | | | | | | | | |
| | Actinostolidae sp.* | | | 1 | 2 | 3 | 3 | 1 | | 1 |
| | Anthomastus sp. | | | | | 1 | | | | |
| Mollusca | | | | | | | | | | |
| Gastropoda | Lepetodrilus concentricus* | 3 | | 3 | 4 | 4 | 4 | 2 | | |
| | Cocculinidae sp.* | | | 2 | 4 | 4 | 4 | 4 | | 1 |
| Bivalvia | Archivesica s.l. puertodeseadoi* | | | 3 | 1 | | 3 | | 2 | |
| | Spinaxinus caldarium* | | | 1 | | | | | 1 | |
| | Parathyasira cf. dearborni | | | 1 | | | | | | |
| Cephalopoda | Octopoda sp. | | | | | | 1 | | | |
| | dead Decapodiformes | | | x | | | | | | |
| Annelida | | | | | | | | | | |
| | Sabellidae sp. | | | | | | | | | 1 |
| | Sclerolinum contortum* | | | 2 | | | | | | |
| | Nicomache lokii* | | | 3 | | | 1 | | 2 | |
| | Polynoidae spp. | | | | | | 1 | | | |
| | Terebellidae spp. | | | 2 | | | 1 | | | |
| | Spionidae spp. | | | 2 | | | 1 | | | |
| | Maldanidae spp. | | | | | | 1 | | | |
| | Amphiomidae spp. | | | | | | | | 1 | |

(Continued.)

**Table 1.** (*Continued.*)

| | | species | WP | GW | PS | BGW | BDF | CR | CSDF | FSDF | BWM |
|---|---|---|---|---|---|---|---|---|---|---|---|
| | | Chaetopteridae spp. | | | | | | 1 | | 1 | |
| Crustacea | | | | | | | | | | | |
| | Cirripedia | Neolepas scotiaensis* | | | | 1 | 1 | | | | |
| | Decapoda | Nematocarcinus lanceopes | | | | | | | | 2 | |
| | | dead Nematocarcinus | | | x | | | | | | |
| Chelicerata | | | | | | | | | | | |
| | Pycnogonida | Sericosura bamberi* | | | | 2 | 3 | 4 | 4 | | 1 |
| | | Sericosura curva* | | | | | | 1 | | | |
| | | Sericosura dimorpha* | | | | | | 1 | | | |
| | | Nymphon cf. longicoxa | | | | | | | | | 1 |
| Echinodermata | | | | | | | | | | | |
| | Ophiuroidea | indet | | | | | | | | 2 | |
| | Asteroidea | Paulasterias tyleri* | | | 1 | | | | | 1 | 1 |
| | Echinoidea | Sterechinus dentifer | | | | | 1 | | | | |
| | Holothuridea | Psolus sp. | | | | 1 | | | | | |
| Chordata | | | | | | | | | | | |
| | | dead Notolepis annulata | | | x | | | | | | |

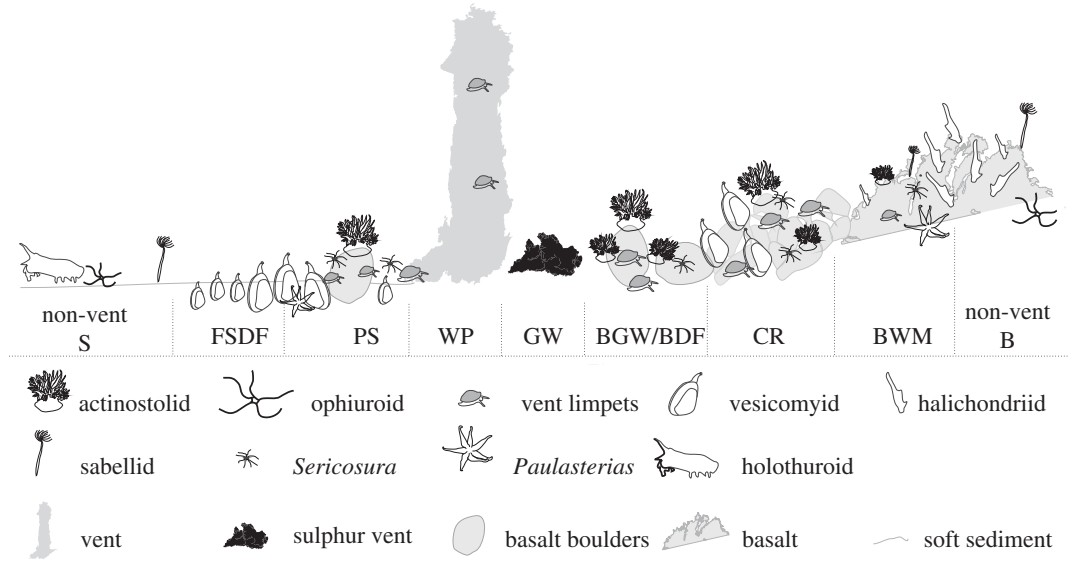

**Figure 4.** Idealized schematic of the spatial distribution of the Kemp Caldera vent field faunal assemblages with increasing distance from sulfur vent or vent fluid orifice.

cocculinid limpet, with the occasional *L. concentricus* and actinostolid anemone. No vesicomyid clams, common in the adjacent biotope 'Clam Road', were present here.

### 3.2.1.8. Fine Sediment in Diffuse Flow (FSDF)

This biotope, defined by fine sediments in the influence of the diffuse flow, was characterized by clusters of burrowing vesicomyid clams. Thyasirid bivalves [79] lived infaunally among the vesicomyid clams and the sea star *Paulasterias tyleri* was also observed on the clam beds. Individuals of the shrimp *Nematocarcinus* sp. were seen walking around on the sediment and occasionally siboglinid worm tubes and ophiuroids could be seen.

### 3.2.1.9. Basalt covered in white mat (BWM)

The 'BWM' biotope is characterized by the presence of halichondriid sponges and a fine white bacterial or mineral precipitate cover on the basalt (figure 2*d*). In this biotope, few macrofauna were seen apart from sponges and the occasional sea star *Paulasterias tyleri*. Towards the border of this zone, near areas influenced by hydrothermal activity, macrofauna associated with the hydrothermal environment including *Sericosura* spp., the cocculinid limpet, and actinostolid anemones began to appear; initially animals were rare, but they then increased in numbers as the abundance of the halichondriid sponge decreased.

Overall, the Kemp Caldera vent field showed a consistent pattern of faunal zonation, with changing assemblage types in different biotopes and with increasing distance from the hot vent fluid source (figure 4).

### 3.2.2. Vent biotope associated macro- and megafauna

In total, 26 benthic species were recovered from the chemosynthetic biotopes in the Kemp Caldera (table 1), of which ten taxa were known from non-chemosynthetic environments in the Southern Ocean, and 13 species have so far been discovered only at hydrothermally active areas.

The demosponge dominating the fauna in the 'BWM' biotope and occasionally occurring in diffuse flow areas was identified as a species belonging to the family Halichondriidae based on the morphology of the simple spicules (D. Janussen 2011, personal communication).

The actinostolid anemones with dark red tentacles were relatively large with a pedal disc greater than 10 cm diameter. The tentacles were armed with strong cnidocytes, which stung through double-layered nitrile examination gloves resulting in a numbing, tingling sensation (C.N. Roterman 2010, personal communication). This species occurred in six of the biotopes but highest abundances were seen in 'BDF' and 'CR'.

The small vent limpet *Lepetodrilus concentricus* occurred in six of the eight hydrothermal biotopes in the Kemp Caldera as well as on the natural whale fall [78], but was missing in the toxic environment of 'GW' and in the outermost area influenced by diffuse flow, the 'BWM' biotope. Molecular analysis of the barcode gene region of COI confirmed that *Lepetodrilus* limpets from the Kemp Caldera were conspecific with *L. concentricus* from the ESR vents [33,80].

The cocculinid limpet was the visually most abundant species in the hydrothermal biotopes being found in six of them and was also found on the natural whale fall. The gill morphology suggests affinity to Cocculinidae within the order Cocculinida [81,82]

The large vesicomyid clam was present in six biotopes of the Kemp Caldera, showing two different lifestyle modes; one group was seen buried in the soft sediments, while the other was seen living epilithically on basalt. Morphologically, based on shell and soft part characteristics, the specimens resemble *Archivesica* s.l. *puertodeseadoi* (Signorelli & Pastorino, 2015) (E. Krylova 2019, personal communication). To determine the presence of endosymbionts in the gill tissue, clone libraries were constructed for 16S rDNA and resulted in 16 sequences from epifaunal and 115 sequences from infaunal specimens. The sequence analysis showed that only one single symbiotic species of Gammaproteobacteria from the SUP05 cluster is present in the vesicomyid clams in the Kemp Caldera. The phylogenetic identification in the arb search indicated a species of Oceanospirillales, an endosymbiont in vesicomyid clams off Florida, as the closest relative (figure 5).

In the precipitate and diffuse flow fine sediment areas, a few specimens of thyasarid bivalves were found and identified as *Spinaxinus caldarium* and *Parathyasira* cf. *dearborni* [79]. While *P.* cf. *dearborni* did not host symbiotic bacteria, *S. caldarium* hosted endosymbiotic bacteria in their gills, which were of the same phylotype as symbiotic bacteria in *Spinaxinus emicatus* from the Gulf of Mexico [79]. Siboglinid polychaetes matching the tube morphology of *Sclerolinum contortum* were found in sulfidic sediments at 'PS' (figure 3*h*). The maldanid *Nicomache lokii* was found at 'PS', 'CR' and 'FSDF'.

The stalked neolepadid barnacles occasionally found on the basalt of the diffuse flow areas were *Neolepas scotiaensis*, a species described from the ESR vents at segment E2 and E9 and also the Kemp Caldera [34]. As only three barnacle specimens were collected in Kemp and used for taxonomy, no material was available for stable isotope analysis.

The medium-sized pycnogonids present in the Kemp hydrothermal biotopes belonged to three species of the vent-affiliated genus *Sericosura* [56]. Molecular COI sequence analysis showed that *Sericosura bamberi*, *S. curva* and *S. dimorpha* were closely related to each other and form a sister clade to *Sericosura venticola* from Northern Pacific vents [56].

The multi-armed (7- to 8-armed) forcipulatacean sea star reported from clam fields and basalts in low diffuse flow areas belong to the recently described *Paulasterias tyleri* [35]. Only four specimens of *P. tyleri* were collected in the Kemp Caldera and were used for taxonomic identification.

## 3.3. Faunal similarity of the vent assemblage at Kemp Caldera with vent fields in neighbouring regions

Multivariate analyses of the presence/absence data for 159 macrofaunal and megafaunal taxa endemic to chemosynthetic environments from 16 vent fields in the Southern, Indian and Atlantic Oceans (figure 6*a*) show that the vent fauna at Kemp Caldera is most similar to faunal assemblages at E2 and E9 hydrothermal vent fields on the ESR, but with a lower degree of similarity than that found between those two ESR vent fields (figure 6*b*). The vent fauna of Kemp Caldera was most similar to the fauna at E9, showing a 44% Sørensen similarity, followed by E2 with a 32% Sørensen similarity, while E2 and E9 exhibited 85% Sørensen similarity. At a regional scale, vent fields within each ocean (Southern, Indian and Atlantic) were more similar to each other in faunal composition than to those in other oceans (figure 6*b,c*).

## 3.4. Macro- and megafauna of non-venting environments

### 3.4.1. Pelagic fauna

A high abundance of pelagic fauna, especially nekton, was observed in the water column of the caldera and some seemed to follow the ROV *Isis*, presumably attracted to the lights of the underwater camera systems. Several species of megafaunal pelagic crustaceans were present, e.g. the Antarctic krill *Euphausia superba*, dark red mysids and the bentho-pelagic shrimp *Nematocarcinus lanceopes*, as well as

**Figure 5.** Phylogenetic position of the Kemp Caldera 16S rRNA sequences (in bold) from the water and the vesicomyid endosymbiont within the SUP05 cluster. Sequences were added to the Silva102 guide tree by parsimony. Bootstrap values (only values greater than 50 are shown) were calculated by nearest neighbour interchange within arb.

individuals of large jellyfish, similar to the genus *Poralia* (D. Lindsay 2014, personal communication) and comb jellies. In near-bottom waters different fish species were present. Once a large toothfish *Dissostichus mawsoni* was seen, while often individuals of two different macrourid species (rattails), a muraenolepid and the paralepidid *Notolepis annulata* were observed in non-venting areas. Three morphotypes of squid

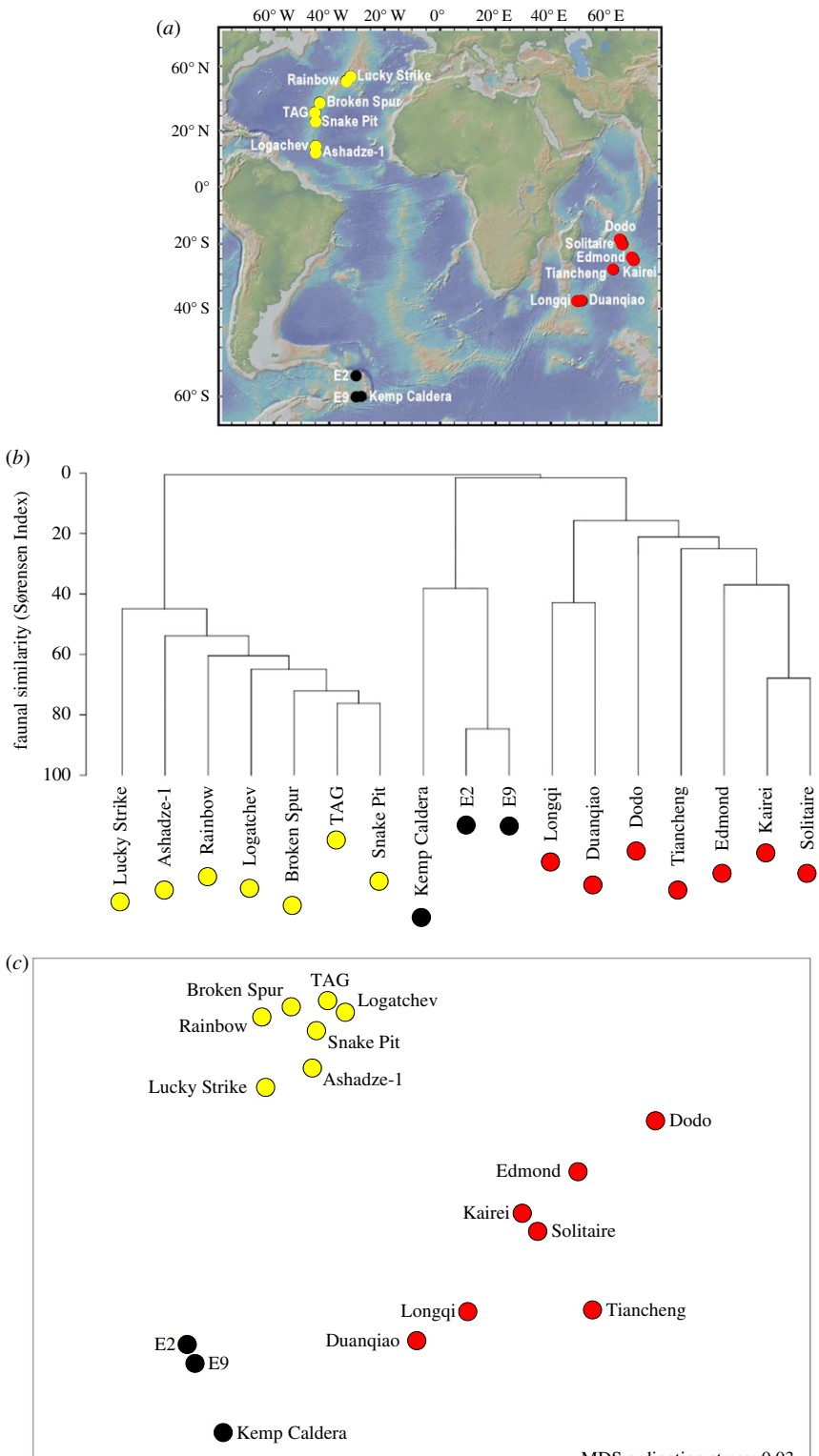

**Figure 6.** Comparison of vent faunal composition at the Kemp Caldera with 16 well-studied vent fields in neighbouring oceanic regions; red-filled circles represent vent fields in the Indian Ocean (Southwest Indian Ridge and Central Indian Ridge), yellow-filled circles represent vent fields on the Mid-Atlantic Ridge, black-filled circles represent vent fields in the Southern Ocean. (*a*) Location of hydrothermal vent fields included in multivariate analysis of faunal composition; topography shown is from the Global Multi-Resolution Topography (GMRT) synthesis (http://www.geomapapp.org/). (*b*) Hierarchical agglomerative clustering using group-average linkage for the presence/absence records of 'chemosynthetic-environment endemic' macro- and megafaunal taxa (329 records of 159 taxa across 17 vent fields, presented in the electronic supplementary material). (c) Two-dimensional non-metric multidimensional scaling (MDS) plot of Sørensen Index similarity matrix calculated from the presence/absence records of 'chemosynthetic-environment endemic' macro- and megafaunal taxa, with 5% metric MDS solution to reduce point-collapse.

were reported; a mid-sized (approx. 40 cm) red-coloured species was identified as *Alluroteuthis antarcticus* (L. Allcock 2013, personal communication), a smaller (approx. 20 cm) white species was identified as *Slosarczykovia circumantarctica* and a red-spotted squid similar to *Moroteuthis knipovitchi* was also seen but less frequently.

### 3.4.2. Benthic fauna

The general benthic faunal distribution in the Kemp Caldera were assessed based on a single SHRIMP transect from the deepest point of the caldera via the resurgent cone to the NE rim during JR224, while ROV *Isis* dived several times on the resurgent cone during JC042 and enabled a closer examination of the sediments and taxa present. The seabed in the deeper part of the caldera was covered with a thin sediment layer, interrupted by a series of lava outcrops, giving way to a cobbled seafloor closer to the resurgent cone, which is described in detail below. Mobile species of holothurians were abundant on the sedimented floor of the caldera, oriented in the same direction, along with occasional dense patches of cerianthid sea anemones and some asteroids. Lava outcrops were dominated by dense coverage of ophiuroids, with brisingid asteroids and other suspension feeders including soft coral, sponges and sea anemones also occasionally present. The caldera wall towards the rim consisted of a series of sheer faces of lava displaying hexagonal cooling joints. The fauna there was dominated by ophiacanthid ophiuroids, particularly on lava blocks at the base of the wall of hexagonal columns. The hexagonal cooling joints were occupied by yellow tube-forming sponges, with brisingids and ophiacanthids on the upper edges of pillar steps.

On the lower eastern flank of the resurgent cone around 1415 m deep, orange-coloured metalliferous, raised sediment structures were present with orange-coloured soft sediments covering the seafloor between them (figure 2*a*). No epifaunal macrobenthic specimens were visible on these sediments or on the orange structures, but notothenioid (e.g. *Notolepis annulata*) and macrourid fishes, as well as krill could be seen in the area. When the soft sediment colour changed to grey-brown and rough-edged basalt boulders appeared on the flank higher up the resurgent cone, epifaunal macrofauna specimens became present, increasing in abundance with increasing distance from the orange-coloured substrates (figure 2*b,c*). The areas of the resurgent cone were under no influence of hydrothermal activity and hosted benthic faunal assemblages similar to those known from the Antarctic and Southern Ocean continental slope and deep-sea plains. The sessile faunal component on the basalt consisted of cnidarians, especially hormathiid anemones and octocorals of the genus *Anthomastus*, bryozoans similar to *Hornea* sp., the stauromedusae *Lacunaria* sp., sabellid polychaetes, psolid holothurians, as well as synascidians, while no hexactinellid sponges were recorded. The comatulid *Promachochrinus* sp. and the brisingid *Freyella* cf. *fragilissima* were observed on raised boulders. In the sedimented areas, burrowing anemones, the pennatularian *Umbellula carpenteri* and a variety of asteroids, ophiuroids, holothurians and echinoids (*Sterechinus dentifer*, *Ctenocidaris spinosa*) were present and the bentho-pelagic shrimp *Nematocarcinus lanceopes* was also seen. The majority of observed macrofauna were mobile species like the nemertean *Parbolasia* cf. *corrugatus*, the decapod *Eualus amandae*, the gastropods *Austrodoris kerguelensis* and a pleurobranchid, several species of the polychaete families Dorvilleidae, Polynoidae, Maldanidae, Scalibregmatidae, Hesionidae, Ampharetidae, Amphinomidae, the pygnogonids *Nymphon* cf. *longicoxa* and *Colossendeis* sp. A variety of echinoderms was seen, including ophiuroids like *Ophiolimna antarctica*, *Ophioperla* sp. and ophiacanthids, asteroids, e.g. *Hymenaster* cf. *coccinatus*, *Odontaster penicillatus* and holothurians including *Bathyplotes* cf. *gourdoni*, *Protelipidia* sp., Elaspodida sp., *Scotoplanes* sp., *Molpadiodemas* sp. The ecology of *O. antarctica* collected away from the hydrothermally active sites in the Kemp Caldera was studied and no apparent influence of venting on the diet of the ophiuroid was observed [83].

The abundance and presence of the Southern Ocean benthic species decreased nearer to the hydrothermal active zones. In the 'BWM' zone, the outermost hydrothermally active biotope defined by the haliocondriid sponges, psolid holothurians were occasionally observed.

## 3.5. Microbiology

The number of prokaryotic cells (determined by flow cytometry) in water samples taken at five points within the vent plume in the Kemp Caldera ranged $1.70–2.15 \times 10^5$ cells $ml^{-1}$. Examination of the microbial composition based on 16S rDNA clone libraries from one water sample taken within the buoyant vent plume (60 m above the chimney) at 'GW' showed that Gammaproteobacteria make up 95% of the bacterial community (table 3). Within the Gammaproteobacteria, the majority of sequences

**Table 2.** Mean $\delta^{13}$C, $\delta^{15}$N and $\delta^{34}$S (‰) of hydrothermal vent fauna and non-vent fauna collected from the Kemp Caldera. Standard deviations are in parentheses.

| taxon | tissue | biotope | N | $\delta^{13}$C | $\delta^{15}$N | N | $\delta^{34}$S |
|---|---|---|---|---|---|---|---|
| Halichondriidae sp. | whole | BWM | 9 | −40.93 (0.28) | 5.68 (0.58) | 3 | 3.53 (0.25) |
| Actinostolidae sp. | tentacle | BDF | 35 | −24.59 (0.62) | 8.54 (0.52) | 13 | 15.05 (1.51) |
| Cocculinidae sp. | whole | BGW | 14 | −26.80 (2.05) | 3.46 (0.64) | 12 | 3.94 (0.73) |
| Cocculinidae sp. | whole | CSDF | 11 | −23.81 (1.59) | 6.06 (1.17) | 8 | 7.16 (0.61) |
| *Archivesica* s.l. *puertodeseadoi* | foot | FSDF | 38 | −35.61 (0.36) | −6.47 (1.73) | 9 | 4.97 (2.856) |
| *Archivesica* s.l. *puertodeseadoi* | foot | CR | 28 | −35.20 (0.38) | −3.24 (1.61) | 10 | 8.79 (0.85) |
| Maldanidae sp. | whole | CR | 1 | −26.98 | 3.55 | 0 | — |
| Terebellidae sp. | whole | CR | 1 | −27.45 | 2.81 | 0 | — |
| *Sericosura bamberi* | whole | BGW | 8 | −24.18 (0.76) | 8.59 (0.62) | 8 | 8.15 (1.93) |
| *Sericosura bamberi* | whole | CSDF | 16 | −21.89 (2.25) | 8.64 (0.68) | 15 | 8.75 (2.29) |
| *Anthomastus* sp. | polyps | non-vent | 4 | −22.45 (1.35) | 9.30 (0.29) | 1 | 18.93 |
| Echinoid sp. | gonad | non-vent | 3 | −26.96 (0.81) | 8.97 (0.81) | 0 | — |
| *Freyella* sp. | tube feet | non-vent | 3 | −23.48 (1.41) | 9.99 (0.90) | 3 | 18.34 (0.29) |
| *Bathyplotes* sp. | muscle | non-vent | 2 | −22.73 (0.24) | 7.25 (0.08) | 1 | 18.13 |
| Holothuroidea sp. 1 | muscle | non-vent | 3 | −22.44 (0.10) | 8.08 (0.67) | 3 | 17.46 (0.45) |
| *Hymenaster* sp. | tube feet | non-vent | 3 | −22.27 (0.58) | 10.92 (0.48) | 0 | — |
| *Odontaster penicillatus* | arm | non-vent | 1 | −23.33 | 14.99 | 0 | — |
| *Ophiolimna antarctica* | arm | non-vent | 3 | −22.29 (0.27) | 9.88 (0.26) | 0 | — |
| *Psolus* sp. | muscle | non-vent | 2 | −22.60 (0.99) | 9.07 (0.26) | 0 | — |
| Polynoidae sp. | whole | non-vent | 1 | −23.70 | 11.02 | 0 | — |
| *Nematocarcinus lanceopes* | muscle | non-vent | 4 | −24.01 (0.47) | 7.33 (0.26) | 0 | — |
| Macrouridae sp. | muscle | non-vent | 1 | −24.60 | 10.93 | 1 | 16.03 |
| *Notolepis annulata* | muscle | non-vent | 3 | −26.00 (0.65) | 7.53 (0.14) | 2 | 17.02 (1.08) |
| *Notolepis annulata* (dead) | muscle | BDF | 1 | −26.09 | 8.33 | 1 | 17.13 |

(57% of total bacterial sequences) were assigned to the SUP05 cluster (figure 5), with high sequence similarity (greater than 98%) to uncultured free-living bacteria found in other vent systems and moderate sequence similarity (95–97%) to vesicomyid symbionts. Archaeal sequences were split into two groups: marine group I (Crenarchaeota), making up 72% of the sequences and Thermoplasmatales (Euryarchaeota) making up the rest (28%).

## 3.6. Trophodynamics

The mean $\delta^{13}$C values of hydrothermal vent macrofauna covered a range of 19.04‰ with the minimum value of −40.93‰ (±0.28 s.d.) observed in the Halichondriidae sp. sampled at 'BWM' and the maximum value of −21.89‰ (±2.25 s.d.) in *S. bamberi* found in 'CSDF' (table 2). The large vesicomyid clam, *Archivesica* s.l. *puertodeseadoi*, was noticeably depleted in $^{12}$C relative to other hydrothermal macrofauna with $\delta^{13}$C values of −35.61‰ (±0.36 s.d.) at 'FSDF' and −35.20‰ (±0.38 s.d.) at 'CR'. Mean $\delta^{15}$N values ranged from −6.47‰ (±0.52 s.d.) in *A.* s.l. *puertodeseadoi* from 'FSDF' to 8.64‰ (±1.73 s.d.) in *S. bamberi* found in 'CSDF'. *Archivesica* s.l. *puertodeseadoi* was the only species with negative $\delta^{15}$N values, with all other hydrothermal vent macrofauna having values greater than 2.81‰,

**Table 3.** Microbial composition based on 16S rDNA clone libraries from the buoyant vent plume at 'Great Wall'.

| | sequences | % |
|---|---|---|
| **Archaea** | **47** | **100** |
| Crenarchaeota—marine group I | 34 | 72 |
| Euryarchaeota—Thermoplasmatales | 13 | 28 |
| **Bacteria** | **87** | **100** |
| Bacteroidetes—Flammeovirgaceae | 1 | 1 |
| Proteobacteria | 86 | 99 |
| Alphaproteobacteria—SAR11 | 1 | 1 |
| Deltaproteobacteria—SAR324 | 2 | 2 |
| Gammaproteobacteria | 83 | 95 |
| Alteromonadales—Alteromonadaceae | 16 | 18 |
| Oceanospirillales—Halomonadaceae | 10 | 11 |
| Oceanospirillales—SUP05 cluster | 50 | 57 |
| Oceanospirillales—other | 4 | 5 |
| Pseudomonadales | 3 | 3 |

which was found in the Terebellidae spp. from 'CR'. The minimum mean $\delta^{34}$S value was observed in the Halichondriidae sp. at 'BWM' (3.53‰ ± 0.25 s.d.) while the maximum value was observed in Actinostolidae sp. (15.05‰ ± 1.51 s.d.) in the 'BDF' biotope. The ranges of stable isotope values of the non-vent fauna in the Kemp Caldera were much narrower than those within chemosynthetic habitats: $\delta^{13}$C from −26.95‰ (±0.81 s.d.) in the echinoid to −22.29‰ (±0.27 s.d.) in the ophiuroid *Ophiolimna antarctica*; $\delta^{15}$N from 7.25‰ (±0.08 s.d.) *Bathyplotes* sp. to 14.99‰ in *Odontaster penicillatus*; and for $\delta^{34}$S the minimum value observed was 16.03‰ in the Macrouridae sp. and the maximum was 18.93‰ in the octocoral *Anthomastus* sp.

## 4. Discussion

The investigation of the Kemp Caldera resulted in discoveries of a resurgent cone with hydrothermally active venting areas and a natural whale fall (characterized by Amon *et al*. [78]) and of a wide range of marine species, distributed over several biotopes with distinct communities and environmental characteristics. The 'dead zone' of pelagic animal carcasses on the seafloor in the 'PS' biotope is similar to that reported in island-arc vents in the Pacific [53] and Caribbean [54], and native sulfur deposits found at Kemp Caldera are also common in sites in the Mariana and Kermadec arcs [84]. The most abundant megafaunal taxa in hydrothermally active venting areas of the Kemp Caldera included a large vesicomyid clam living both infaunally and epibenthically, cocculinid and lepetodrilid limpets, the sea spiders *Sericosura* spp., and actinostolid anemones. The overall biomass at Kemp Caldera's hydrothermal active areas appeared lower when compared with that of the ESR vent fields [27,36].

The vent biotopes and assemblages at the Kemp Caldera differed from those of the E2 and E9 vent fields on the nearby ESR: although seven species were shared between the Kemp Caldera and the ESR vent fields (table 4), the stalked barnacle *N. scotiaensis* and sea star *P. tyleri* were rare at Kemp, and several species that are abundant at ESR vents were not observed at Kemp Caldera, including the yeti crab *K. tyleri* and the gastropods *G. chessoia*, *Bruceiella indurata* [85] and *Provanna cooki* [86]. Although the Kemp Caldera is geographically closest to the E9 vent field (90 km distance), the vent fauna at Kemp Caldera exhibited only a 44% Sørensen Index of similarity with the vent fauna at E9, whereas E2 and E9 exhibit 85% Sørensen similarity between them despite being 440 km apart (figure 6). This difference is consistent with island-arc vent assemblages being distinct from those on seafloor spreading centres in the same region, and here may result from differences in vent fluid geochemistry, seafloor type and depth.

Hydrothermal venting at Kemp Caldera was characterized by 'white smoker' chimneys releasing fluids at temperatures between 103 and 212°C [58], with wide-ranging diffuse flow in basalt and sedimented seafloor areas from 1375 to 1487 m depth. The vent fluid composition differed from the E9

**Table 4.** Comparison of Kemp Caldera chemosynthetic fauna with other Southern Ocean hydrothermal fauna.

| | | hydrothermal site | | | | | | | sister taxa |
|---|---|---|---|---|---|---|---|---|---|
| | species | Kemp | E2 | E9 | AAR-KR1 | BS-HR | Larsen-B | Kemp-WF | |
| **Cnidaria** | | | | | | | | | |
| Anthozoa | Actinostolidae sp. 1 | + | | | | | | | Pac, Atl |
| | Actinostolidae sp. 2 | | + | | | | | | Pac, Atl |
| | Actinostolidae sp. 3 | | | + | | | | | Pac, Atl |
| **Annelida** | | | | | | | | | |
| Polychaeta | Sclerolinum contortum | (+) | | | | + | | | Pac, Atl |
| | Nicomanche lokii | + | + | | | | | | Atl |
| **Mollusca** | | | | | | | | | |
| Gastropoda | Gigantopelta chessoia | | + | + | | | | | Ind |
| | Lepetodrilus concentricus | + | + | + | | | | + | Pac, Atl, Ind |
| | Cocculinidae sp. | + | | | | | | + | Pac |
| | Provanna cooki | | + | + | | | | | Pac, Atl |
| | Brucjella indurata | + | + | + | | | | | Pac |
| Bivalvia | Spinaxinus caldarium | + | + | | | | | | Pac, Atl |
| | Parathyasira cf. dearborni | + | | | | | | | SO |
| | Archivesica s.l. puertodeseadoi | + | | | | | + | | Pac, Atl |
| **Arthropoda** | | | | | | | | | |
| Cirripedia | Neolepas scotiaensis | + | + | + | | | | | Pac |
| Decapoda | Kiwa tyleri | + | + | + | | | | | Pac, Ind |
| | Kiwa araonae | | | | + | | | | Pac, Ind |
| Pycnogonida | Sericosura bamberi | + | + | + | | | | + | Pac, Atl |
| | Sericosura curva | + | + | + | | | | + | Pac, Atl |
| | Sericosura dimorpha | + | + | + | | | | + | Pac, Atl |
| **Echinodermata** | | | | | | | | | |
| Asteroidea | Paulasterias tyleri | + | + | + | + | | | | Pac |

site, which is located 90 km away in approximately 2400 m depth, being high in hydrogen sulfide and iron concentrations and with evidence for an input of magmatic gas [29,48]. By contrast, the E2 and E9 vent fields, the latter around 440 km away in approximately 2600 m depth, were characterized by pillow basalts, sheet lavas and 'black smoker' chimneys emitting vent fluids 351 to 383°C in temperature [27], with a fluid composition similar to that reported from other basalt-hosted vent sites on seafloor-spreading centres [30].

The microbial composition in the buoyant plume of the Kemp Caldera venting chimneys was different from the microbial communities reported from the buoyant vent plumes of the black smoking chimneys from the ESR segments E2 and E9 [27]. At the Kemp Caldera, the bacterial community was with 95% highly dominated by Gammaproteobacteria, particularly of the SUP05 cluster which represented 57% of the total community, compared to the ESR sites E2 and E9, where proteobacteria represented 70% and 66% of the bacterial communities of which 58% and 55% were Gammaproteobacteria, respectively, and SUP05 contributing only 9% and 19% to the Gammaproteobacteria. The Gammaproteobacteria at the ESR sites were dominated by bacterial symbionts of vent fauna [27]. Furthermore, Alphaproteobacteria of the SAR11 clade were common at ESR (16% and 21% of the total community), but rare (1% of total community) at the Kemp Caldera. Members of the SUP05 cluster are known to be sulfur oxidizers [87–89]. The analysis of water samples from CTD cast collected over 'GW' in the Kemp Caldera showed sulfur-rich fluids (D.P. Connelly 2019, personal communication). With regard to carbon fixation, the presence of both the cbbM gene [87] and its gene product, ribulose 1,5-bisphosphate carboxylase/oxygenase (RuBisCo) [88], has been shown in SUP05 communities at other hydrothermal vent settings. This implies that the SUP05 bacteria may be dominant microbial primary producers in the water column above Kemp Caldera venting sites.

Differences in substrate may also contribute to the differences in the vent faunal assemblage at Kemp Caldera compared with the E2 and E9 vent fields on the nearby ESR. The lack of soft sediment in the immediate vicinity of the chimneys at E2 and E9 on the ESR may preclude the presence of the vesicomyid bivalves, and the sulfur and anhydrite composition of the brittle chimney structures at the Kemp Caldera may limit the available surfaces for the growth of *N. scotiaensis* barnacles. Differences in geochemistry between the ESR and the Kemp Caldera may also account for some of the discrepancy in assemblages. The depth difference between the ESR and Kemp Caldera vent fields (approx. 1700 m including the caldera rim) may also act as a dispersal filter, as discussed by Roterman *et al*. [80], favouring the dispersal of *Lepetodrilus* spp., with their small and numerous larvae, compared with taxa such as *Kiwa tyleri* that have fewer, larger and probably non-buoyant larvae [90,91]. Additionally, the presence of *Lepetodrilus concentricus* at the whale carcass in the Kemp Caldera [78], and the presence of the genus at hard and soft substrate vents, seeps and whale falls in general [92] suggests that these limpets may have a wider selection of dispersal stepping stones available to them than other fauna endemic to chemosynthesis-based ecosystems.

Multivariate analysis of faunal composition of vent fields at a regional scale shows that the Kemp Caldera is most similar to the ESR vent fields that define a previously recognized 'Southern Ocean' province of vent biogeography [24,27,31–35], and distinct from vents in neighbouring oceans (figure 6). One species found at the vents in Kemp Caldera, however, has also been recorded at hydrothermal vents on the Southwest Indian Ridge the pycnogonid *Sericosura bamberi* at the Duanqiao vent field [64]. The polychaetes *Nicomache lokii* and *Sclerolinum contortum* were the only two megafauna species collected at the Kemp Caldera for which wide-ranging, bipolar distributions have been reported [24,25], though neither have been recorded at vents in the Indian Ocean or on the Mid-Atlantic Ridge.

Further to records from the ESR and Kemp Caldera, the predatory sea star *Paulasterias tyleri* has also been collected at the South Sandwich Island, the Ross Sea and the Australian Antarctic Ridge [28,35]. Live specimens and shells of the vesicomyid *Archivesica* s.l. *puertodeseadoi* have been compared with vesicomyid shell fragments from the Larsen B extinct seepage sites [19,20] and assigned to the same species (E. Krylova 2019, personal communication). No live vesicomyid specimens have yet been collected or reported from the Larsen B or other areas covered by ice shelves [18–20], but the presence of active seepage under ice shelves has been predicted by Ingels *et al*. [93].

Fauna found in venting areas at Kemp Caldera exhibit some distributional overlaps with the natural whale fall that was also found on the resurgent cone. Amon *et al*. [78,94] reported 11 macrofaunal species on the nearby whale fall that probably harvest the bones directly or feed on bacterial mats growing on them, including three species of the bone-boring *Osedax* polychaetes [94], two species of dorvilleid *Ophryotrocha* and one capitellid polychaete, the janaerid isopod *Jaera tyleri* [95], one species of lysianassid amphipod, and one species each of lepetodrilid, cocculinid and pyropeltid gastropods. Of

these species, only the lepetodrilid and cocculinid gastropods were found in the Kemp Caldera vent biotopes, indicating that the other species reported from the whale fall may be dependent on food sources provided by decaying whale bones, but not by hydrothermal environments.

The majority of the seafloor in the caldera, away from the resurgent cone, was not affected by hydrothermal activity and the fauna resembled that reported from comparable bathyal depths elsewhere in the Southern Ocean [96,97]. Biodiversity assessments on the bathyal Southern Ocean benthic fauna are still sparse and limited to sedimented areas suitable for trawling [98–101], restricting direct faunal comparisons. Species characteristics of Southern Ocean deep waters, such as *Umbellula carpenteri* and *Sterechinus dentifer,* were present in the Kemp Caldera [102,103]. The non-vent fauna of the Kemp Caldera also included a species of hippolytid shrimp, *Eualus amandae*, which has remained undescribed until recently [104].

The macrofauna associated with the hydrothermal venting areas of the Kemp Caldera showed signs of using chemosynthetic primary production. The $\delta^{13}$C values of Halichondriidae sp. and the large vesicomyid clam, *Archivesica* s.l. *puertodeseadoi*, were clearly less than those of benthic species depending on epipelagic photosynthetic primary production in the form of sinking particulate organic matter (POM) on the ESR ($\delta^{13}$C, approx. −26 to −21‰ [37]) and those reported here. The sponge Halichondriidae sp. had particularly low $\delta^{13}$C values being less than −40‰. Methanotrophic bacteria are associated with Demospongiae from hydrothermal vent and cold seep habitats [105,106] and it may be that methanotrophic bacteria were contributing to the carbon pool of Halichondriidae sp. The $\delta^{13}$C values of *A.* s.l. *puertodeseadoi* were consistent with other species of vesicomyid collected from other hydrothermal vent settings, which range between approximately −37‰ and approximately −32‰ [107–109]. The vesicomyid endosymbionts were within the SUP05 cluster of the Gammaproteobacteria, which fixes carbon using the RuBisCo enzyme and would indicate that chemosynthetic primary production is via the Calvin–Benson–Bassham cycle.

The other species sampled within the hydrothermal vent biotopes (approx. −27 ‰ to approx. −21‰) had $\delta^{13}$C values that overlapped with those of species from non-venting areas (approx. −26‰ to approx. −22‰). These hydrothermal vent organisms are probably using chemosynthetic primary production fixed by free-living Gammaproteobacteria using the RuBisCo II enzyme, which has a lower isotopic fractionation than RuBisCo I enzyme [110], rather than using photosynthetic POM. The polychaetes, Terebellidae spp. and Maldanidae spp., have lower $\delta^{15}$N values than would be expected from consuming photosynthetic POM and are therefore likely to represent *in situ* production [111]. This is clearly evidenced in the $\delta^{34}$S values of the cocculinid limpet and *S. bamberi*, which ranged between approximately 4‰ and approximately 8‰ and are much lower than would be expected for deep-sea fauna using photosynthetic POM as an energy source [37,112]. The only exception is the actinostolid, which was found in the 'BDF' biotope, which had $\delta^{34}$S values of approximately 15‰ and therefore similar to the non-vent fauna (approx. 16‰ to approx. 18‰). This indicates that photosynthetic primary production may be contributing to the diet of this anemone. Interestingly, *S. bamberi* was observed feeding on dead carrion including *Notolepis annulata* and *Nematocarcinus lanceopes*, but the stable isotope values do not indicate that these constitute potentially high contributions to the diet.

The lowest $\delta^{13}$C values were found within *S. bamberi* (−21.89‰) and were lower than those expected for organisms using carbon fixed via the reductive tricarboxylic acid (rTCA) cycle (−15‰ to −10‰) [113]. Carbon fixation via the rTCA cycle at hydrothermal vents is predominately carried out by Epsilonproteobacteria. Hydrothermal vent fauna using this source of production are often found close to the vent orifice where temperatures are high and the reducing conditions are stronger than other areas of the vent habitat. $\delta^{13}$C values of hydrothermal vent fauna at arc volcano and caldera, including the Brothers Caldera, NW Eifuku Volcano and Sumisu hydrothermal vents, indicate that rTCA primary production can enter the food web [107,114,115]. However, none of the hydrothermal vent fauna sampled for $\delta^{13}$C analysis had values that were indicative of this production source within Kemp Caldera.

The pelagic fauna within the Kemp Caldera appeared to be high in abundance, but no pelagic sampling was carried out to verify these subjective observations, to compare with abundance data from adjacent areas like the South Sandwich Islands and to test if the pelagic fauna was using chemosynthetic primary production. Studies of the potential influence of hydrothermal vent plumes on biological communities in the water column are a neglected field [116]. Hydrothermal vent plumes are enriched with abundant and active communities of chemosynthetic prokaryotes (e.g. [117–120]) as well as POM [119]. This enhanced supply of POM has the potential to act as a food source for deep-water zooplankton communities and therefore may exert an influence on pelagic food webs in the proximity of hydrothermal vents and their plumes. Acoustic and net sampling studies over the

Endeavour Segment of the Juan de Fuca Ridge, NE Pacific, have consistently demonstrated the enhancement of deep-scattering layers overlying vent plumes at 1900–2100 m (e.g. [121–125]). These scattering layers were predominantly formed by deep-living copepods (e.g. *Neocalanus* sp.) but also comprised chaetognaths, other crustaceans and gelatinous zooplankton. Zooplankton in the vicinity of vent plumes have been demonstrated to have unusual $\delta^{15}N$ values consistent with feeding on production from the vent plume, as well as sinking organic matter and that upwelling from the seafloor [126]. It is likely in this locality that zooplankton migration and the release of eggs and larvae that move upwards in the water column are enhancing production in the region from deep waters to the epipelagic zone. Zooplankton production over the vents may be double or triple that of the background ocean [125]. High abundances of gelatinous zooplankton have been observed in the vicinity of vents in the southwest Pacific and in the Okinawa Trough (e.g. [127,128]). Large pelagic predators have also been seen within hydrothermal plumes in the shallow waters of Kavachi volcano crater in the Solomon Islands [127]. Observations from hydrophones on the Juan de Fuca Ridge also suggest that blue and fin whale may be feeding in the vicinity of the hydrothermal plumes because of the enhanced zooplankton production [129,130]. Incidental observations from other localities also hint at connectivity between vent plumes and wider ocean food webs. Larvae of vent endemic megafauna may also be enriched in vent plumes, providing another food source for pelagic organisms. However, although larvae have been sampled from vent plumes (e.g. [131,132]), their numbers have been small and understanding of the larval flux from vents to the pelagic ecosystem remains poorly defined [111]. Large pelagic megafauna, like whales, dying and decaying near vent sites might also provide a food source and stepping stone for some vent organisms [133]. The presence of the natural whale fall in the Kemp Caldera is an example of specialized species being shared between chemosynthetic food sources [78].

## 5. Conclusion

The Kemp Caldera at the southern end of the volcanic South Sandwich Arc hosts a unique hydrothermally influenced ecosystem with different biotopes and a composition of macro- and megafaunal species not found anywhere else to date. The chemosynthetic communities include species like the bivalve *Spinaxinus caldarium* and an undescribed cocculinid limpet, which are as yet specific to this location, while some species are shared with the ESR vent fields or even vent fields further away. Biogeographically, the Kemp Caldera vent field belongs to the Southern Ocean vent province. This discovery of active hydrothermal venting sites with associated chemosynthetic communities in relatively shallow bathyal waters of the Southern Ocean suggests the presence of further hydrothermal ecosystems on unexplored seamount, submarine caldera and crater sites of the South Sandwich Arc.

Ethics. Permission to collect samples in Antarctica was granted to P. Tyler by the Foreign and Commonwealth Office London (permit S3–3/2009) under §3 of the Antarctic Act 1994 and authorization was given to the persons specified in electronic supplementary material, Appendix I, to enter and remain in Antarctica for the purpose of scientific research. No special 'Animal Care Protocol' was required at the time.
Data accessibility. Our genetic data are deposited at GenBank accessions numbers MK736312–MK736351. The datasets supporting this article have been uploaded as part of the electronic supplementary material.
Authors' contributions. K.L., J.T.C., A.D.R., W.D.K.R. and K.Z. drafted the manuscript, K.L., J.T.C., V.A.I.H., A.G.C.G., L.M., W.D.K.R. and K.Z. prepared figures and tables, C.C., A.C., D.P.C., J.T.C., A.D.R., A.G.C.G., V.A.I.H., K.L., R.D.L., L.M., D.A.P., N.V.C.P., W.D.K.R., C.N.R., C.J.S., P.A.T. and K.Z. were involved in fieldwork, analyses and manuscript edits. All authors gave final approval for publication.
Competing interests. The authors declare no competing interests.
Funding. Financial support came from UK's Natural Environmental Research Council (Consortium grant no. NE/DO1249X/1, PhD studentships NE/D01429X/1, NE/F010664/1) and the Sloan Foundation (Census of Marine Life).
Acknowledgements. We thank the masters and crews of *RRS James Clark Ross* and *RRS James Cook* and the staff of the UK National Marine Facilities at NOC, especially the ROV *Isis* team, and the science teams on board for logistic, technical and shipboard support during JR224 and JC042. We are also grateful to two anonymous reviewers, who provided comments that improved the manuscript.

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
