## [Reviewer comments · Royal Society Open Science]

Review History

RSOS-190660.R0 (Original submission)

Review form: Reviewer 1 (Julian Gutt)

Is the manuscript scientifically sound in its present form?

Yes

Are the interpretations and conclusions justified by the results?

Yes

Is the language acceptable?

Yes

Is it clear how to access all supporting data?

Yes

Do you have any ethical concerns with this paper?

No

Have you any concerns about statistical analyses in this paper?

No

Recommendation?

Accept with minor revision (please list in comments)

Comments to the Author(s)

The manuscript describes the unique communities and environmental conditions of a new vent site in the Southern Ocean and, thus, contributes to a more complete global knowledge of extreme marine habitats. Comprehensive results are analyzed using suitable and modern methods. The discussion puts the findings in a broader ecological and geographical context. The manuscript is very well written and structured. Thus, it can be published almost as it is. I only have a few comments how eventually the text can be improved.

P. 2, line 9: Gutt et al. (2011) showed a seep at Larsen B in 215m depth, which should be cited here because it extends the depth range where such seeps occur in the Southern Ocean (Gutt J, Barratt I, Domack E, d'Udekem d'Acoz C, Dimmler W, Grémare A, Heilmayer O, Isla E, Janussen D, Jorgensen E, Kock K-H, Lehnert LS, López-González P, Langner S, Linse K, Manjón-Cabeza ME, Meißner M, Montiel A, Raes M, Robert H, Rose A, Sañé Schepisi E, Saucède T, Scheidat M, Schenke H-W, Seiler J, Smith C 2011. Biodiversity change after climate-induced ice-shelf collapse in the Antarctic. *Deep-Sea Res II* 58: 74-83)

P. 2, line 25: To make clear that the description etc of the studied vent site starts here I would not only write "A further deep-water area..." but something like "The focus of this study was...".

P. 3, line 9: Maybe it comes later but it would be helpful to get a bit more information on the optical resolution of the images (or minimum size of objects to be recognized and/or the average size of sea-floor area covered by a photo of the width of a video stripe).

P. 3, line 9ff: Would be nice to get a clear information, whether the images are photos or videos or grabs from videos.

P. 3, line 13: A 3-ship camera recorder or only a 3-ship camera and a recorder

P. 5, line 3: Would be helpful if "hydrothermally active area" could be specified, e.g. to which max. area around a smoker or other features described in the next paragraph do the results refer?

P. 4, line 46: Does the suggestion of an undescribed species also refer to the 94% similarity with the other species mentioned above?

P. 9, line 31: Genus name probably to be spelled out?

Fig. 1: Is georeferencing of Fig. 1E possible?

Figs 2-3: Could positions for the single photos be provided?

Review form: Reviewer 2 (Verena Tunnicliffe)

Is the manuscript scientifically sound in its present form?

Yes

Are the interpretations and conclusions justified by the results?

No

Is the language acceptable?

Yes

Is it clear how to access all supporting data?

Yes

Do you have any ethical concerns with this paper?

No

Have you any concerns about statistical analyses in this paper?

No

Recommendation?

Major revision is needed (please make suggestions in comments)

Comments to the Author(s)

Attached (Appendix A).

Decision letter (RSOS-190660.R0)

19-Jun-2019

Dear Dr Linse:

Manuscript ID RSOS-190660 entitled "The fauna of the upper bathyal hydrothermal vents of the Kemp Caldera (South Sandwich Arc, Antarctica)" which you submitted to Royal Society Open Science, has been reviewed. The comments from reviewers are included at the bottom of this letter.

In view of the criticisms of the reviewers, the manuscript has been rejected in its current form. However, a new manuscript may be submitted which takes into consideration these comments.

Please note that resubmitting your manuscript does not guarantee eventual acceptance, and that your resubmission will be subject to peer review before a decision is made.

Your resubmitted manuscript should be submitted by 17-Dec-2019. If you are unable to submit by this date please contact the Editorial Office.

Kind regards,
Alice Power
Editorial Coordinator
Royal Society Open Science

on behalf of Dr Ari Friedlaender (Associate Editor) and Kevin Padian (Subject Editor)
 openscience@royalsociety.org

Subject Editor Comments to Authors:

Both reviews were constructive but one had substantial issues that should be addressed with a substantial rewrite, which could best be done without a short timeframe such as our "major revision" has. Accordingly we suggest that you address all comments very specifically and we will look forward to your resubmission.

Reviewers' Comments to Author:

Reviewer: 1

Comments to the Author(s)

The manuscript describes the unique communities and environmental conditions of a new vent site in the Southern Ocean and, thus, contributes to a more complete global knowledge of extreme marine habitats. Comprehensive results are analyzed using suitable and modern methods. The discussion puts the findings in a broader ecological and geographical context. The manuscript is very well written and structured. Thus, it can be published almost as it is. I only have a few comments how eventually the text can be improved.

P. 2, line 9: Gutt et al. (2011) showed a seep at Larsen B in 215m depth, which should be cited here because it extends the depth range where such seeps occur in the Southern Ocean (Gutt J, Barratt I, Domack E, d'Udekem d'Acoz C, Dimmler W, Grémare A, Heilmayer O, Isla E, Janussen D, Jorgensen E, Kock K-H, Lehnert LS, López-González P, Langner S, Linse K, Manjón-Cabeza ME, Meißner M, Montiel A, Raes M, Robert H, Rose A, Sañé Schepisi E, Saucède T, Scheidat M, Schenke H-W, Seiler J, Smith C 2011. Biodiversity change after climate-induced ice-shelf collapse in the Antarctic. *Deep-Sea Res II* 58: 74-83)

P. 2, line 25: To make clear that the description etc of the studied vent site starts here I would not only write "A further deep-water area..." but something like "The focus of this study was...".

P. 3, line 9: Maybe it comes later but it would be helpful to get a bit more information on the optical resolution of the images (or minimum size of objects to be recognized and/or the average size of sea-floor area covered by a photo of the width of a video stripe).

P. 3, line 9ff: Would be nice to get a clear information, whether the images are photos or videos or grabs from videos.

P. 3, line 13: A 3-ship camera recorder or only a 3-ship camera and a recorder

P. 5, line 3: Would be helpful if "hydrothermally active area" could be specified, e.g. to which max. area around a smoker or other features described in the next paragraph do the results refer?

P. 4, line 46: Does the suggestion of an undescribed species also refer to the 94% similarity with the other species mentioned above?

P. 9, line 31: Genus name probably to be spelled out?

Fig. 1: Is georeferencing of Fig. 1E possible?

Figs 2-3: Could positions for the single photos be provided?

Reviewer: 2

Comments to the Author(s)

Attached

Author's Response to Decision Letter for (RSOS-190660.R0)

See Appendix B.

RSOS-191501.R0

Review form: Reviewer 1 (Julian Gutt)

Is the manuscript scientifically sound in its present form?

Yes

Are the interpretations and conclusions justified by the results?

Yes

Is the language acceptable?

Yes

Do you have any ethical concerns with this paper?

No

Have you any concerns about statistical analyses in this paper?

No

Recommendation?

Accept as is

Comments to the Author(s)

In my first review I considered this manuscript already as well written and providing sound and comprehensive results, which had been put into a broader ecological context. I only had minor recommendations, which had been considered in the revised and resubmitted version of the manuscript. With exceptions also the points of criticism of the other reviewer had been solved of which some additionally improved the quality of the manuscript slightly.

I would like to repeat that the manuscript provides original results but it is also a kind of review. It will be useful for specialists in ecology, biodiversity, geology and biogeochemistry in the Southern Ocean as well as experts for worldwide vent ecology. It could also stimulate the scientific community to investigate the ecology of hydrothermal vents in different oceans in more details or provide complete oceanwide metaanalyses, which was not the aim of this manuscript.

I recommend to publish the manuscript now as it is.

Review form: Reviewer 2 (Verena Tunnicliffe)

Is the manuscript scientifically sound in its present form?

Yes

Are the interpretations and conclusions justified by the results?

Yes

Is the language acceptable?

Yes

Do you have any ethical concerns with this paper?

No

Have you any concerns about statistical analyses in this paper?

No

Recommendation?

Accept with minor revision (please list in comments)

Comments to the Author(s)

Review of Linse et al. Version 2

This manuscript is notably improved by the revisions. Below are a few comments. It needs a complete re-read for the small errors, still. (For example, the first line of the title - is that really what you want?) I put line numbers on a converted Word document (Appendix C) where you can also find smaller edits - including many highlighted trivial mistakes. I'm sure you can find more :)

l. 14 in Abstract: Multivar analysis doesn't show 'how' (=mechanism) - just shows that it 'is'.

l. 287: Just an aside: I wonder if the scars mean that this species is a suspension feeder or even hosts an epibiont as does *Lepetodrilus fucensis*.

l. 407: *Sericosura venticola* does not occur on the East Pacific Rise (it's at Endeavour, JdF ridge). Ref 57 listed here did not give me any assurance.....

l. 647-652: I do not believe you can interpret seafloor producers based on water/plume samples. Please see Gregory Dick's recent review of Microbiomes of deep-sea hydrothermal vents, in which he comments:

Plumes are composed primarily of organisms derived from the water column, such as SUP05 (Gammaproteobacteria), SAR324 (Deltaproteobacteria), SAR11 (Alphaproteobacteria) and Marine Group I archaea⁹⁷⁻⁹⁹. Sea floor and/or subsurface organisms such as Epsilonproteobacteria can be present in plumes^{98,99} but are quickly diluted owing to the massive entrainment of background seawater²⁸.

Thus, I suggest you remove the water column comment here and just leave the mystery until such time that the seafloor producers are sampled.

Refs 49 thru 56 appear to be missing.

Table 4 caption: Spell out SO. Do we interpret that "Pac" means sister taxa found throughout the Pacific rather than NPac (as indicated) or WPac (*Sclerolinum* is only, WPac, I think); standardize to region or to ocean. Similarly, Ind vs SWIR.

What does the bold text mean?

Decision letter (RSOS-191501.R0)

10-Oct-2019

Dear Dr Linse,

On behalf of the Editor, I am pleased to inform you that your Manuscript RSOS-191501 entitled "The fauna of the Kemp Caldera and its the upper bathyal hydrothermal vents (South Sandwich Arc, Antarctica)" has been accepted for publication in Royal Society Open Science subject to minor revision in accordance with the referee suggestions. Please find the referees' comments at the end of this email.

The reviewers and Subject Editor have recommended publication, but also suggest some minor revisions to your manuscript. Therefore, I invite you to respond to the comments and revise your manuscript.

- Ethics statement

- Data accessibility

<http://datadryad.org/submit?journalID=RSOS&manu=RSOS-191501>

- Competing interests

- Authors' contributions

- Acknowledgements

- Funding statement

Because the schedule for publication is very tight, it is a condition of publication that you submit the revised version of your manuscript before 19-Oct-2019. Please note that the revision deadline will expire at 00.00am on this date. If you do not think you will be able to meet this date please let me know immediately.

Supplementary files will be published alongside the paper on the journal website and posted on

the online figshare repository (<https://figshare.com>). The heading and legend provided for each supplementary file during the submission process will be used to create the figshare page, so please ensure these are accurate and informative so that your files can be found in searches. Files on figshare will be made available approximately one week before the accompanying article so that the supplementary material can be attributed a unique DOI.

Kind regards,

Lianne Parkhouse
Royal Society Open Science
openscience@royalsociety.org

on behalf of Dr Ari Friedlaender (Associate Editor) and Kevin Padian (Subject Editor)
openscience@royalsociety.org

Associate Editor Comments to Author (Dr Ari Friedlaender):

To the Authors,

Based on the comments by two reviewers and my own opinion, I am satisfied with how you have handled the previous comments. I believe the manuscript is now more robust and intellectually sound and can be accepted. However, the reviewers do note that there is still considerable work to be done editorially speaking to clean up the language. I urge the authors to be diligent in this regard and please ensure that the final submission has been copy edited thoroughly. Otherwise, I congratulate the authors on their fine work and look forward to seeing this study published!

Thank you.
Ari S. Friedlaender

Reviewer comments to Author:

Reviewer: 1
Comments to the Author(s)

In my first review I considered this manuscript already as well written and providing sound and comprehensive results, which had been put into a broader ecological context. I only had minor recommendations, which had been considered in the revised and resubmitted version of the manuscript. With exceptions also the points of criticism of the other reviewer had been solved of which some additionally improved the quality of the manuscript slightly.

I would like to repeat that the manuscript provides original results but it is also a kind of review. It will be useful for specialists in ecology, biodiversity, geology and biogeochemistry in the Southern Ocean as well as experts for worldwide vent ecology. It could also stimulate the scientific community to investigate the ecology of hydrothermal vents in different oceans in more details or provide complete oceanwide metaanalyses, which was not the aim of this manuscript.

I recommend to publish the manuscript now as it is.

Reviewer: 2

Comments to the Author(s)

Review of Linse et al. Version 2

This manuscript is notably improved by the revisions. Below are a few comments. It needs a complete re-read for the small errors, still. (For example, the first line of the title - is that really what you want?) I put line numbers on a converted Word document where you can also find smaller edits - including many highlighted trivial mistakes. I'm sure you can find more :)

l. 14 in Abstract: Multivar analysis doesn't show 'how' (=mechanism) - just shows that it 'is'.

l. 287: Just an aside: I wonder if the scars mean that this species is a suspension feeder or even hosts an epibiont as does *Lepetodrilus fucensis*.

l. 407: *Sericosura venticola* does not occur on the East Pacific Rise (it's at Endeavour, JdF ridge). Ref 57 listed here did not give me any assurance.....

l. 647-652: I do not believe you can interpret seafloor producers based on water/plume samples. Please see Gregory Dick's recent review of Microbiomes of deep-sea hydrothermal vents, in which he comments:

Plumes are composed primarily of organisms derived from the water column, such as SUP05 (Gammaproteobacteria), SAR324 (Deltaproteobacteria), SAR11 (Alphaproteobacteria) and Marine Group I archaea⁹⁷⁻⁹⁹. Sea floor and/or subsurface organisms such as Epsilonproteobacteria can be present in plumes^{98,99} but are quickly diluted owing to the massive entrainment of background seawater²⁸.

Thus, I suggest you remove the water column comment here and just leave the mystery until such time that the seafloor producers are sampled.

Refs 49 thru 56 appear to be missing.

Table 4 caption: Spell out SO. Do we interpret that "Pac" means sister taxa found throughout the Pacific rather than NPac (as indicated) or WPac (*Sclerolinum* is only, WPac, I think); standardize to region or to ocean. Similarly, Ind vs SWIR.

What does the bold text mean?

Author's Response to Decision Letter for (RSOS-191501.R0)

See Appendix D.

Decision letter (RSOS-191501.R1)

23-Oct-2019

Dear Dr Linse,

I am pleased to inform you that your manuscript entitled "Fauna of the Kemp Caldera and its

upper bathyal hydrothermal vents (South Sandwich Arc, Antarctica)" is now accepted for publication in Royal Society Open Science.

on behalf of Dr Ari Friedlaender (Associate Editor) and Kevin Padian (Subject Editor)
openscience@royalsociety.org

Appendix A

Linse et al: The fauna of the upper bathyal hydrothermal vents of the Kemp Caldera (South Sandwich Arc, Antarctica)

General Comments:

This paper presents information that is valuable for the record. As the location is difficult to access, it is worth publishing most of the results. However, there is a grab-bag here that makes it a little difficult to create a coherent story. Overall, the reader is left with a poor understanding of the significance of the work. To that end, I make several suggestions:

1. A better job of the geological setting. I had to open other papers to find what should be highlighted in the Introduction. The ESR is a back-arc spreading centre that is part of the tectonic region forming the South Sandwich Arc. Kemp appears to be the southernmost volcano in this arc thus with a very close proximity to the E2 and E9 sites. A reasonable hypothesis to set up in the Intro is that, based on distance, the vent fauna should be similar. The authors appear unaware of the comparative work proceeding elsewhere on arcs vs backarcs. I feel this point is a primary contribution of this paper.
2. Following that point, the Discussion never mentions hydrothermal systems on volcanic arcs and the processes that can drive sulphur deposition and specialized faunas. Kermadec, Tonga, and Mariana Arcs have interesting comparative information. Again, a major focus can emerge to tie in the Kemp site and explain the apparent disparity with the ESR. One starter paper is Embley et al. 2007 [Exploring the Submarine Ring of Fire: Mariana Arc - Western Pacific. *Oceanography*, 20: 68-79]; (lots of animals live on sulphur deposits).
3. I'm afraid the isotope data and the "food web" need to go – they are a few data points that do not fit together and certainly don't make a food web. In this decade, we do expect samples of the potential C-N sources to support mixing extrapolations. The fact that the results are never mentioned in the Discussion underlines the irrelevance.
4. The species lists and taxonomy: I am disappointed the authors show low respect for accurate information. My grad students use papers like this one, and we are forever seeing sloppy spelling and taxonomy that have to be corrected. Far too many errors occur here.
5. Integrate the vent plume and seafloor better in a comparison to the ESR to emphasize how the geological setting (think about the fluid sources and influences) may be affecting primary production and secondary consumers. Again, more literature review would assist in interpretation of the differences.

Specific Comments

Title: The non-vent part of this paper has useful observations; perhaps a new title and keywords can open up the topic.

Introduction:

I had to get Google Earth open with the Interridge overlay to follow the vent discoveries description. Please label the ESR sites on Figure 1 as you use them later. What depths are these sites? (or adopt the figure in Cole et al)

p. 2, l. 14 verb agreement

l. 17: define 'ROV'

l. 36 verb missing

Methods:

p. 3, l.3: add “on the ROV”

l. 33: where is this caldera “next to Kemp” on Fig 1? How far “next”; is it an ancillary structure of this seamount. The Intro mentions “caldera near the Kemp Seamount” several times. What edifice supports this caldera?

I suggest you follow the precedent of Hawkes et al and call it “Kemp Caldera”.

The term “white smoker” was first applied to *Alvinella* chimneys – the ‘smoke’ was all the particulates they released. Define your white smoke.

l. 57: nonetheless, it would be very useful to include the meiofauna in the species lists for reference.

p. 4, l. 12: what does “within the vent plume mean here” – detail the indicators, intensity, depth and altitude. Is it one sample or many combined?

You flip from water to clam, then more clam and back to water again. I’m hoping the relevance of throwing in the clam endosymbiont becomes clear later. (*Later: not really but the data are interesting.*)

Results

p. 5, l.14: “too toxic for metazoans” – your comment is surmise only. ‘Pure sulphur’ environments are populated on Kermadec, Sanghi and Mariana Arcs. Nikko Seamount is one such example.

l.19: “white anhydrites” Do you know sure? – I have my doubts. Similar white smokers I have seen on NW Eifuku and northern Tonga Arc are elemental sulphur ... *after making this comment, I read Cole et al – use this reference to back your assertion.*

Were polymetallic sulphide chimneys present? *Again I feel you can do better with the introduction to set up the site description.*

l.53: if you have the sediment ... have someone determine what it is; anhydrite crystals are easy to recognize, but it is sulphur? (See de Ronde 2015 “Molten sulfur lakes of intraoceanic arc volcanoes” .

p. 6, l. 56: I don’t think you will find this ‘siboglinid’ in WORMS.

p. 6 Section 4.2.2: Your table lists seven polychaete families (each with ‘spp’ meaning you have more than one species). Given how limited the dataset is, I am disappointed you can do no better to determine identity on these animals. They comprise a major part of your collection ... and the text does not even mention them.

p.7, Biogeography: Results say 12 vent species, Sup Table has 13.

Discussion

The overall vent setting description is useful. Comment on the overall biomass at these vents – it appears to be very small and scattered; correct?

Erecting a definitive biogeographic analysis based on 12 species is tenuous. Please comment in the discussion on how robust this analysis is. I suspect finding even one more species co-shared with any one of the three regions would change the shape of the dendrogram and Sorensen’s index.

l. 50: I could not find the difference in depths mentioned: oh, on the next page – bring it up. (many vent species transcend large depth ranges such as this).

- l. 56: What does “compared to ESR sites” mean here? What dominated there; any SUP05? And note that SUP05 is very common in other settings such as OMZs.
- p. 9, l.2: “remarkable” is a bit overstated. It would be appropriate to consider the literature that compares faunal overlaps between ventfields. Try Goffredi et al. 2017 (*Proc Royal Soc B*. 284, 1859) for more perspective on inter-site differences.
- l. 9: sulphur comment is not supported by literature.

Tables and Figures:

Table 1: if you are committing to a family, use correct nomenclature (e.g. “Actinostolidae”). If you are not sure, then move up in the taxonomy. A couple spelling errors here.

Which 10 taxa “were known from non-chemosynthetic environments in the Southern Ocean”? perhaps indicate those you determine to be chemosyn-related

Table 2: Vial number not useful. What are the numbers in brackets? Spelling error.

Table 4 – I see at least three species names spelled incorrectly; we surely can do better to get the databases correct, no? (That’s not counting “Actynostolid”.)

Figure 1: enlarge font on axes and on sample sites

A box in D to indicate the location of C would help.

Figure 2: images need scales or approx. distance across. Caption could tell us more about the organisms. J – is this stuff sulphur or anhydrite?

Figure 3: Spelling! “siphons”; actinostolid – sure of family? *Sericosura*

Figure 4 – fix the spelling problems: check WORMS if you aren’t sure.

Appendix B

Dear Dr Frielaender, Dr Padian and Mrs Power in your roles as Editors and Editorial Coordinator of Royal Society Open Science,

Please receive our resubmission of Manuscript ID RSOS-190660 entitled "The fauna of the upper bathyal hydrothermal vents of the Kemp Caldera (South Sandwich Arc, Antarctica)", with a track-changes and cleaned manuscript version as well as below our specific and detailed responses to the reviewers' comments and suggestions.

We hope that you will be able to accept the revised manuscript for publication.

With kind regards in the name of all authors,

Katrin Linse

Subject Editor Comments to Authors:

Both reviews were constructive but one had substantial issues that should be addressed with a substantial rewrite, which could best be done without a short timeframe such as our "major revision" has. Accordingly we suggest that you address all comments very specifically and we will look forward to your resubmission.

We have addressed the two reviewers' comments and have rewritten parts of the introduction and discussion.

Authors revision comments to the Reviewers' Comments:

Reviewer: 1

Comments to the Author(s)

The manuscript describes the unique communities and environmental conditions of a new vent site in the Southern Ocean and, thus, contributes to a more complete global knowledge of extreme marine habitats. Comprehensive results are analyzed using suitable and modern methods. The discussion puts the findings in a broader ecological and geographical context. The manuscript is very well written and structured. Thus, it can be published almost as it is. I only have a few comments how eventually the text can be improved.

We thank Reviewer 1 for their overall comments on our manuscript.

P. 2, line 9: Gutt et al. (2011) showed a seep at Larsen B in 215m depth, which should be cited here because it extends the depth range where such seeps occur in the Southern Ocean (Gutt J, Barratt I, Domack E, d'Udekem d'Acoz C, Dimmler W, Grémare A, Heilmayer O, Isla E, Janussen D, Jorgensen E, Kock K-H, Lehnert LS, López-González P, Langner S, Linse K, Manjón-Cabeza ME, Meißner M, Montiel A, Raes M, Robert H, Rose A, Sañé Schepisi E, Saucède T, Scheidat M, Schenke H-W, Seiler J, Smith C

2011. Biodiversity change after climate-induced ice-shelf collapse in the Antarctic. *Deep-Sea Res II* 58: 74-83)

We have included the depth information in the text and added the reference as into reference list.

P. 2, line 25: To make clear that the description etc of the studied vent site starts here I would not only write "A further deep-water area..." but something like "The focus of this study was..."

We edited the sentence following reviewer 1's suggestion.

P. 3, line 9: Maybe it comes later but it would be helpful to get a bit more information on the optical resolution of the images (or minimum size of objects to be recognized and/or the average size of sea-floor area covered by a photo of the width of a video stripe).

We added "to collected video imagery of the seabed that enabled identification of megafaunal (> 3 cm) benthic invertebrates".

P. 3, line 9ff: Would be nice to get a clear information, whether the images are photos or videos or grabs from videos.

We added "to collect video imagery of the seabed" to make clear that videos were recorded in line 11 and in line 15: "or seafloor imagery "(photos, video and frame grabs of videos)" ROV Isis was

P. 3, line 13: A 3-chip camera recorder or only a 3-chip camera and a recorder

We specified this by describing it as a 3-chip domestic Panasonic camcorder.

P. 5, line 3: Would be helpful if "hydrothermally active area" could be specified, e.g. to which max. area around a smoker or other features described in the next paragraph do the results refer?

We added "characterised by presence of "white smoking" venting, diffuse flow, precipitates and/or bacterial mats". A maximum area in metrics like meters cannot be given as the area is dependent on the hydrothermal activity and this activity can spatially and temporally be different.

P. 6, line 46: Does the suggestion of an undescribed species also refer to the 94% similarity with the other species mentioned above?

We have significantly edited this section as new information on the vesicomid clam has been given to us by Elena Krylova. The clams are now assigned to *Archivesica* s.l. *puertodeseadoi* (Signorelli & Pastorino, 2015). The paragraph now reads: "Morphologically, based on shell and soft part characteristics, the specimens resemble *Archivesica* s.l. *puertodeseadoi* (Signorelli & Pastorino, 2015) (E. Krylova personal communication). BLAST searches of the COI sequences from Kemp Caldera clam specimens against sequences in GenBank resulted in *Calyptogena* sp TO-2011 AB634285, as specimen now assigned to *Archivesica* s.l. *fortunata* (Johnson et al 2017), *Laubericoncha chuni* isolate Lc4 JN563828 and *Calyptogena laubieri* ssp. 'CM-2 M' AB479083 as best matches with ~94% similarity. Like *Archivesica* s.l. *fortunata*, *Laubericoncha chuni* and *Calyptogena laubieri* were assigned to *Archivesica* s.l. in the most recent phylogenetic study [29]."

We also removed the reference to the dark black periostracum in the buried specimens.

From this point on in the text and in all tables we refer to *Archivesica* s.l. *puertodeseadoi*.

P. 9, line 31: Genus name probably to be spelled out?

We have spelled the genus of *Paulasterias tyleri* out.

Fig. 1: Is georeferencing of Fig. 1E possible?

We have georeferenced Fig 1E now.

Figs 2-3: Could positions for the single photos be provided?

We have provided the biotope and position information for the single photos in Figure 3 taken from the USBL of the dive track at the time the image was taken.

Reviewer: 2

Comments to the Author(s)

Linse et al: The fauna of the upper bathyal hydrothermal vents of the Kemp Caldera (South Sandwich Arc, Antarctica)

General Comments:

This paper presents information that is valuable for the record. As the location is difficult to access, it is worth publishing most of the results. However, there is a grab-bag here that makes it a little difficult to create a coherent story. Overall, the reader is left with a poor understanding of the significance of the work. To that end, I make several suggestions:

1. A better job of the geological setting. I had to open other papers to find what should be highlighted in the Introduction. The ESR is a back-arc spreading centre that is part of the tectonic region forming the South Sandwich Arc. Kemp appears to be the southernmost volcano in this arc thus with a very close proximity to the E2 and E9 sites. A reasonable hypothesis to set up in the Intro is that, based on distance, the vent fauna should be similar.

We extended the paragraph on the South Sandwich Arc and described the geological setting of the Sandwich Plate. We added "The minor tectonic Sandwich Plate is separated in the west from the Scotia Plate by the East Scotia Ridge (EAR), a back-arc spreading centre formed by the subduction of South American Plate on the eastern margin of the microplate, which is also forming the South Sandwich Trench [45, Leat et al 2013, Larter et al. 2003]. To the south, the Sandwich Plate border by the Antarctic Plate [45]. The active volcanic arc ranges from the northern Protector Shoal seamounts in the north via eleven volcanic islands to the seamounts in the south of which the Kemp Seamount is one of the southernmost ones [45]."

We also added a hypothesis: "We hypothesise that the chemosynthetic fauna present in the venting sites of the Kemp Caldera will be similar to the chemosynthetic fauna of the ESR segments E2 and E9, based on their relatively close distance of only ~440 km and ~90 km away."

The authors appear unaware of the comparative work proceeding elsewhere on arcs vs backarcs. I feel this point is a primary contribution of this paper.

We have added a paragraph on arc vs back arc and mid-ocean ridge vents to the introduction as well as sentences to the discussion section.

2. Following that point, the Discussion never mentions hydrothermal systems on volcanic arcs and the processes that can drive sulphur deposition and specialized faunas.

Kermadec, Tonga, and Mariana Arcs have interesting comparative information. Again, a major focus can emerge to tie in the Kemp site and explain the apparent disparity with the ESR. One starter paper is Embley et al. 2007 [Exploring the Submarine Ring of Fire: Mariana Arc - Western Pacific. *Oceanography*, 20: 68-79]; (lots of animals live on sulphur deposits).

We added a sentence to the discussion.

3. I'm afraid the isotope data and the "food web" need to go – they are a few data points that do not fit together and certainly don't make a food web. In this decade, we do expect samples of the potential C-N sources to support mixing extrapolations. The fact that the results are never mentioned in the Discussion underlines the irrelevance.

We disagree with the reviewer to delete the stable isotope data. We have edited the results and discussion sections to make sure it has a clearer fit to the rest of the manuscript. The aim is to use the stable isotope analysis to investigate trophic ecology of consumers, which make up most of the biomass. We have covered the majority of species, which were found in venting areas and also included non-vent species in order to help put the stable isotope values in context of chemosynthetic v photosynthetic primary production. Unfortunately, we do not have any C and N sources that can be used for mixing extrapolations. However, sampling pure trophic endmembers for ecologically interpretable mixing extrapolations at hydrothermal vent systems is extremely difficult. Furthermore, we do not have reliable estimates of trophic fractionation for hydrothermal vent fauna nor do we know how temperature may influence them, which will influence the results and interpretation of any mixing extrapolation.

4. The species lists and taxonomy: I am disappointed the authors show low respect for accurate information. My grad students use papers like this one, and we are forever seeing sloppy spelling and taxonomy that have to be corrected. Far too many errors occur here.

We have corrected the taxonomic names. I can assure that we do not have low respect for accurate taxonomic names.

5. Integrate the vent plume and seafloor better in a comparison to the ESR to emphasize how the geological setting (think about the fluid sources and influences) may be affecting primary production and secondary consumers. Again, more literature review would assist in interpretation of the differences.

The description of the geochemical setting and their geological sources are not part of this biological assemblage focussed manuscript but subject of a geochemical focussed manuscript. Vent plume description and comparison with the ESR fluid and fluid source comparison will be part of it.

Specific Comments

Title: The non-vent part of this paper has useful observations; perhaps a new title and keywords can open up the topic.

We have edited the keywords exchanging “food web” with “stable isotopes”.

We changed the title to “The fauna of the Kemp Caldera and its of the upper bathyal hydrothermal vents (South Sandwich Arc, Antarctica)”

Introduction:

I had to get Google Earth open with the Interridge overlay to follow the vent discoveries description. Please label the ESR sites on Figure 1 as you use them later. What depths are these sites? (or adopt the figure in Cole et al)

We added depth information of the ESR and AAR sites into the introduction text. We would like to know which figure in Cole et al the reviewer refers to as Figure 1 does not give depth information.

p. 2, l. 14 verb agreement

We edited the entire sentence to avoid the verb agreement if the sentence is read with relating “which” to “subfamily” and not to “species” as the authors intended: “The large vesicomid clam from the Larsen B area belonged to Pliocardiinae, a subfamily of the Vesicomidae, with species associated with sulphide-rich habitats and hosting endosymbionts for their chemosynthetic life style.”

l. 17: define ‘ROV’

We defined ROV as remotely operating vehicle here and removed the definition therefore from p. 3, l. 53.

l. 36 verb missing

We added the missing verb “is” after checking with SGSSI government that this legislation has not been enacted this the manuscript submission.

Methods:

p. 3, l.3: add “on the ROV”

We added “on the ROV”

I. 33: where is this caldera “next to Kemp” on Fig 1? How far “next”; is it an ancillary structure of this seamount. The Intro mentions “caldera near the Kemp Seamount” several times. What edifice supports this caldera? I suggest you follow the precedent of Hawkes et al and call it “Kemp Caldera”.

We replaced “next” with “near” to be consistent with the text in the introduction. We also replaced “caldera near to Kemp Seamount” in most positions with “Kemp Caldera”. To date the Kemp Caldera is not officially named while the Kemp Seamount is, therefore we wanted to introduce the Kemp Caldera as the caldera near to the Kemp Seamount. We have added the two names to Figure 1 D.

The term “white smoker” was first applied to *Alvinella chimneys* – the ‘smoke’ was all the particulates they released. Define your white smoke.

We follow the definition that white smokers are hydrothermal vents emitting hydrothermal fluids that are cooler than those of black smokers. This definition is in line with for example those of the International Seabed Authority (www.isa.org.jm) and Arndt (DOI: https://doi.org/10.1007/978-3-642-11274-4_1691)

I. 57: nonetheless, it would be very useful to include the meiofauna in the species lists for reference.

The faunal samples collected during JC42 by ROV were processed (e.g. sieved over 500 µm steel sieves) for macro- and megafaunal assessments. From the JC42 ROV sampling no meiofaunal taxa have been recorded from the Kemp Caldera. Therefore, we cannot provide meiofaunal taxon names in the species list.

p. 4, I. 12: what does “within the vent plume mean here” – detail the indicators, intensity, depth and altitude. Is it one sample or many combined?

We changed the text to: “For analysis of the microbial biodiversity in the water column, 30L of water from one sample within the buoyant vent plume, at a depth of 1355 m (ca 60 m above the seafloor) taken with CTD Niskin bottles were filtered through a 0.2µm pore size nitrocellulose filter (Whatman GE Healthcare). The water at the sampling depth showed a strong decrease in the redox potential of 342 mV with concomitant increase in LSS of 0.538 compared to surface water.”

You flip from water to clam, then more clam and back to water again. I’m hoping the relevance of throwing in the clam endosymbiont becomes clear later. (Later: not really but the data are interesting.)

While the reviewer states that we “flip from water to clam”, we wrote the methods in this order as it first explains how we collected the samples for the molecular microbiology analyses, then the primers, PCR conditions, sequence treatment and analysis, and later the flow cytometry.

Results

p. 5, I.14: “too toxic for metazoans” – your comment is surmise only. ‘Pure sulphur’

environments are populated on Kermadec, Sanghi and Mariana Arcs. Nikko Seamount is one such example.

We removed the sentence.

I.19: “white anhydrites” Do you know sure? – I have my doubts. Similar white smokers I have seen on NW Eifuku and northern Tonga Arc are elemental sulphur ... after making this comment, I read Cole et al – use this reference to back your assertion.

Were polymetallic sulphide chimneys present? Again I feel you can do better with the introduction to set up the site description.

We added “sulphide” and replaced “white anhydrite” with “bacterial mats” to the sentence: “... an area of dense “white smoking” sulphide chimneys covered in bacterial mats ...”

We have added the reference to Cole et al 2014 [47]. The detailed geochemical characterisation of the Kemp Caldera vents is not part of this manuscript. The detailed geochemical data analysis and write up is led by the geochemists of the ChEsSO team and will be published separately.

I.53: if you have the sediment ... have someone determine what it is; anhydrite crystals are easy to recognize, but it is sulphur? (See de Ronde 2015 “Molten sulfur lakes of intraoceanic arc volcanoes” .

We added an explanation and changed the sentence to: “The white sediments taken, coated in pale bacterial mats, in this biotope had a strong sulphur odour ...”

The detailed geochemical characterisation of the Kemp Caldera sediments and precipitates is not part of this manuscript. The detailed geochemical data analysis and write up is led by the geochemists of the ChEsSO team and will be published separately.

p. 6, l. 56: I don’t think you will find this ‘siboglinid’ in WORMS.

We replaced ‘siboglinid worms’ with “tubes of *Sclerolinum*”.

p. 6 Section 4.2.2: Your table lists seven polychaete families (each with ‘spp’ meaning you have more than one species). Given how limited the dataset is, I am disappointed you can do no better to determine identity on these animals. They comprise a major part of your collection ... and the text does not even mention them.

The text did not mention the polychaete species as apart from the Sabellidae sp. seen in the biotope “Basalt covered in white mat - BMW” von polychaetes were visible on the ROV imagery. The collected sabellid tubes unfortunately did not contain the animal to further identify this species, which was visually present in ROV video footage.

Of the polychaete taxa collected from sediment rubble, but not visible in ROV imagery, *Sclerolinum contortum* and *Nicomache lokii* have to date been identified to species. The polychaete families listed in Table 1 (Polynoidae spp, Terebellidae spp, Spionidae spp., Maldanidae spp., Amphiomidae spp. and Chaetopteridae spp.) are with our polychaete taxonomist and their team and still awaiting identification on species level. As they were not visible in the ROV imagery, but found in the residues

of the suction sampler samples collected in the biotopes “Precipitated sediment”, “Clam Road”, and “Fine Sediment in Diffuse Flow”, we decided to list their presence in Table 1. for general reference of the families’ occurrence.

p.7, Biogeography: Results say 12 vent species, Sup Table has 13.

We corrected the text to 13 species and marked the Halichondriidae sp. as the 13th species linked with the vents in Table 1.

Discussion

The overall vent setting description is useful. Comment on the overall biomass at these vents – it appears to be very small and scattered; correct?

We added a comparison with the ESR vent field biomass; “The overall biomass at Kemp Caldera’s hydrothermal active areas appeared lower when compared with the ESR vent fields [27, 36].”

Erecting a definitive biogeographic analysis based on 12 species is tenuous. Please comment in the discussion on how robust this analysis is. I suspect finding even one more species co-shared with any one of the three regions would change the shape of the dendrogram and Sorensen’s index.

As hydrothermal vents are environments with relatively low species richness compared with other deep-sea habitats, 12 vent-endemic species recorded for a vent site is well within the range of species richness recorded at well-studied sites (as a reader can tell from the Supplementary data set based on published literature), and such data are the basis for previously published analyses of vent biogeography (e.g. Rogers et al. 2012).

BUT this paper is NOT erecting a new biogeographic analysis (which would require running of MRT methods etc); rather, it presents a comparison of faunal similarity between vent fields in regions neighbouring Kemp, which shows how Kemp is most similar to vent assemblages in a previously defined Southern Ocean province of vent biogeography (in Rogers et al., 2012), but nevertheless more of an “outlier” within it, as expected for an island-arc system compared with systems on seafloor spreading centres. The comparison of faunal similarity also highlights that a few species (a pycnogonid in this case) have distributions the cross the boundaries of biogeographic provinces defined for vent fauna by MRT methods. The text has therefore been clarified to remove reference to “biogeography” where that was likely to cause confusion in the nature of the analysis being presented, as indicated by the Reviewer’s comment.

I. 50: I could not find the difference in depths mentioned: oh, on the next page – bring it up. (many vent species transcend large depth ranges such as this).

We added the depth of the Kemp Caldera, E2 and E9 venting sites into the paragraph.

I. 56: What does “compared to ESR sites” mean here? What dominated there; any SUP05? And note that SUP05 is very common in other settings such as OMZs.

We added more information to the sentence: “At Kemp Caldera, the bacterial community was with 95% highly dominated by the Gammaproteobacteria, particularly of the SUP05 cluster which represented 57% of the total community, compared to the ESR sites E2 and E9, where Proteobacteria represented 70% and 66% of the bacterial communities, with Gammaproteobacteria making up 58% and 55% of the Proteobacteria and SUP05 contributing only 9% and 19% to the Gammaproteobacteria. The Gammaproteobacteria at the ESR sites were dominated by bacterial symbionts of vent fauna. [30]. Furthermore, Alphaproteobacteria of the SAR11 clade were common at ESR (16% and 21% of the total community), but rare (1% of total community) at the Kemp Caldera.

p. 9, l.2: “remarkable” is a bit overstated. It would be appropriate to consider the literature that compares faunal overlaps between vent fields. Try Goffredi et al. 2017 (Proc Royal Soc B. 284, 1859) for more perspective on inter-site differences.

We changed “remarkable” with “noteworthy”.

l. 9: sulphur comment is not supported by literature.

We have deleted the sentence.

Tables and Figures:

Table 1: if you are committing to a family, use correct nomenclature (e.g. “Actinostolidae”). If you are not sure, then move up in the taxonomy. A couple spelling errors here. Which 10 taxa “were known from non-chemosynthetic environments in the Southern Ocean”? perhaps indicate those you determine to be chemosyn-related

We have edited the nomenclature following the reviewer’s preferred way of spelling out families and going to higher taxonomic groups, corrected the spelling and added “*” to species that have been recorded from hydrothermally active area only to follow the wording from p. 6, line 17.

Table 2: Vial number not useful. What are the numbers in brackets? Spelling error.

We have deleted the vial numbers, corrected “Benthopelagic” and explained the numbers in brackets, which are the standard deviations.

Table 4 – I see at least three species names spelled incorrectly; we surely can do better to get the databases correct, no? (That’s not counting “Actynostolid”.)

We have corrected the spelling.

Figure 1: enlarge font on axes and on sample sites. A box in D to indicate the location of C would help.

We have enlarged the font sizes on axes and sample sites. We also included a box in D for E.

Figure 2: images need scales or approx. distance across. Caption could tell us more about the organisms. J – is this stuff sulphur or anhydrite?

We have added a scale bar of approximate 10 cm to the images. In images 2B and 2K, two red laser dots give the 10 cm scale, for the other images, scale was assessed by using species’s sizes as scale.

We have corrected the legend, and explained that J) is a sulphur structure while L) is covered in anhydrites. We also refer to the organisms in the images now.

Figure 3: Spelling! “siphons”; actinostolid – sure of family? *Sericosura*

We corrected the spelling to “siphons” and *Sericosura*. Yes, we are sure of the family Actinostolidae as this was confirmed to us by an cnidarian taxonomist of the American Museum of Natural History.

Figure 4 – fix the spelling problems: check **WORMS** if you aren’t sure.

The taxa are now labelled with a higher taxonomic name (e.g. ophiuroid or ophiuroid) or a common name (e.g. vent limpets, as this includes the cocculinid sp. as well as the lepetodrilid sp.).

Appendix C

R. Soc. open sci. article template

ROYAL SOCIETY
OPEN SCIENCE

R. Soc. open sci.
doi:10.1098/not yet assigned

The fauna of the Kemp Caldera and its the upper bathyal hydrothermal vents (South Sandwich Arc, Antarctica)

Formatted: Highlight

Katrin Linse¹, Jonathan Copley², Douglas P. Connelly³, Robert D. Larter¹, David A. Pearce^{1a}, Nick V.C. Polunin⁴, Alex D. Rogers^{5b}, Chong Chen⁶, Andrew Clarke¹, Adrian G. Glover⁷, Alastair G.C. Graham^{1c}, Veerle A.I. Huvenne³, Leigh Marsh², William D.K. Reid⁴, C. Nicolai Roterman⁵, Christopher J. Sweeting⁴, Katrin Zvirgmaier^{1d}, Paul A. Tyler²

1. British Antarctic Survey, High Cross, Madingley Road, Cambridge, CB3 0ET UK
2. Ocean and Earth Science, University of Southampton, Waterfront Campus, Southampton SO14 3ZH UK
3. National Oceanography Centre, European Way, Southampton, SO14 3ZH UK
4. School of Natural and Environmental Sciences, Ridley Building, Newcastle University, Newcastle upon Tyne, NE1 7RU UK
5. Department of Zoology, University of Oxford, South Parks Road, Oxford, OX1 3PS UK
6. SUGAR, Japan Agency for Marine-Earth Science and Technology (JAMSTEC), 2-15 Natsushima-cho, Yokosuka, 237-0061 Kanagawa Pref. Japan
7. Life Sciences Department, Natural History Museum, Cromwell Road, London, SW7 5BD UK

corresponding author, email kl@bas.ac.uk, phone +44 (0)1223 221631,

Current addresses:

- a. Department of Applied Sciences, Faculty of Health and Life Sciences, University of Northumbria, Ellison Building, Newcastle Upon Tyne, NE1 8ST UK
- b. REV Ocean, Oksenøyveien 10, 1366 Lysaker Norway
- c. Physical Geography, University of Exeter, Rennes Drive, Exeter, EX4 4RJ UK
- d. Bundeswehr Institute of Microbiology, Neuherbergstr 11, 80937 Munich Germany

Keywords: Hydrothermal vents, chemosynthetic ecosystem, resurgent cone, stable isotopes, Gammaproteobacteria, biogeography

1. Summary

Faunal assemblages at hydrothermal vents associated with island-arc volcanism are less well-known than
those at vents on mid-ocean ridges and back-arc spreading centres. This study characterises chemosynthetic
biotopes at active hydrothermal vents discovered at the Kemp Caldera in the South Sandwich Arc. Sulphur
and anhydrite vent chimneys at the Kemp Caldera in 1375 to 1487 m depth emit sulphide-rich fluids with
temperatures up to 212 °C, and the microbial community of water samples in the buoyant plume rising from
the vents was dominated by sulphur-oxidising Gammaproteobacteria. In total 12 macro- and megafaunal taxa
dependent on hydrothermal activity were collected in these biotopes, of which two species were previously
undescribed, while seven species were known from the East Scotia Ridge vents and three species from vents
outside the Southern Ocean. Faunal assemblages were dominated by large vesicomid clams, actinostolid
anemones, *Sericosura* seaspiders, and lepetodrilid and cocculinid limpets, but several taxa that are abundant
at nearby East Scotia ridge hydrothermal vents are rare such as the *Neolepas scotiaensis*. Multivariate analysis
of fauna at Kemp Caldera and hydrothermal vents in neighbouring areas indicates that the Kemp Caldera is
most similar to vent fields in a previously established Southern Ocean province of vent biogeography, but
shows how the species composition at island-arc hydrothermal vents can be distinct from nearby seafloor-

Formatted: Highlight

*Author for correspondence (kl@bas.ac.uk).

†Present address: Biodiversity, Evolution and Adaptation, British Antarctic Survey, High Cross, Madingley Road, Cambridge CB3 0ET UK

spreading systems. $\delta^{13}\text{C}$ and $\delta^{15}\text{N}$ isotope values of species analysed from the Kemp Caldera were similar to
those of the same or related species at other vent fields, but none of the fauna sampled for $\delta^{13}\text{C}$ analysis at
Kemp Caldera had values indicating nutritional dependence on Epsilonproteobacteria, unlike fauna at other
island-arc hydrothermal vents.

2. Introduction

The presence of active volcanoes on the Antarctic continent and several islands in the Southern Ocean was
recorded by early polar explorers, such as Sir James Clark Ross who observed an eruption of Mount Erebus in
1841 [1]. These clearly visible volcanoes have been the subject of geological and biological research for a long
time [e.g. 2,3]. The presence of visible volcanic and measurable tectonic activity, such as eruptions, steam
fields, fumaroles and earthquakes on the Antarctic Peninsula, Marie Byrd Land, Victoria Land, the islands of
the South Shetlands and South Sandwich Arc, and their surrounding seas [e.g. 2,4] has implied the presence of
sub-ice and submarine volcanism and hydrothermally activity [5-8].

In the Antarctic and Southern Ocean marine environment, shallow-water hydrothermal fumaroles occur in the
caldera of Deception Island and are characterised by depleted marine life with low species richness and faunal
abundance compared with other Antarctic shallows [e.g. 9-11]. Hydrothermal vents and cold seeps have been
discovered in the Bransfield Strait in 990-1500 m depth [12-17], in the Larsen B ice shelf area in 215 to 850 m
water depth [18-20], and on the shelf of South Georgia in 250 to 350 m water depth [21,22], but the fauna
associated with these locations comprise elements of typical Antarctic and Southern Ocean shelf faunas, apart
from the siboglinid polychaete *Sclerolinum contortum* at Hook Ridge, Bransfield Strait [23], and a large
vesicomid clam at Larsen B, eastern Antarctic Peninsula [22], both of which are taxa associated with
chemosynthetic environments [24-26].

Hydrothermal vent fields with abundant novel fauna have been discovered on the back-arc spreading centre of
the East Scotia Ridge (ESR) at 2400-2600 m depth, and on the Australian Antarctic Ridge (AAR) in 1800-
1900 m depth [27,28]. High-temperature “black-smoker” venting occurs at two vent fields, on ESR segments
E2 and E9 [27,29,30], which are inhabited by species including the yeti crab *Kiwa tyleri*, the gastropods
*Gigantopelta chessoia*, *Lepetodrilus concentricus*, the stalked barnacle *Neolepas scotiaensis*, the seven-armed
seastar *Paulasterias tyleri*, and undescribed actinostolid anemones [31-36], in assemblages supported by
chemoautotrophic carbon fixation [37-40]. To date, the AAR sites have not been visited and imaged by
remotely operated vehicles (ROVs) but specimens of associated vent fauna *Kiwa araeonae* and *Paulasterias*
*tyleri* have been collected, indicating the presence of faunal assemblages supported by chemosynthesis
[28,41].

The South Sandwich arc of the Southern Ocean is comprised of the actively erupting volcanic South
Sandwich Islands and their associated seamounts, and situated on the Sandwich Plate [42,43]. The minor
tectonic Sandwich Plate is separated in the west from the Scotia Plate by the ESR, a back-arc spreading centre
formed by the subduction of South American Plate on the eastern margin of the microplate, which is also
forming the South Sandwich Trench [42,44,45]. To the south, the Sandwich Plate is bordered by the Antarctic
Plate [42]. The active volcanic arc ranges from the northern Protector Shoal seamounts in the north via eleven

Formatted: Font: 11 pt, Highlight

volcanic islands to the seamounts in the south of which the Kemp Seamount is one of the southernmost ones
[42]. Submarine hydrothermal vents have been reported in this island arc at Adventure Crater [46], Protector
Shoal seamounts and Quest Caldera [47], and Kemp Caldera near the Kemp Seamount [29,48], which is the
focus of this study. The Kemp Caldera is situated within a restricted zone of the South Georgia and the South
Sandwich Islands Marine Protected Area (SGSSI MPA), where both research and fisheries activities are
regulated by the SGSSI government [49]. In the SGSSI MPA commercial benthic bottom trawling is banned
and longline fisheries are restricted to depths between 700 and 2250 m depth (www.gov.gs). The SGSSI
government is in the process of enacting legislation to protect the environment of the SGSSI MPA from any
future mining or hydrocarbon activity (exploration and exploitation) to align with the Madrid Protocol of the
Antarctic Treaty, and elucidating the faunal assemblages associated with hydrothermal vents in the SGSSI
MPA also informs that legislative effort.

The ecology of deep-sea hydrothermal vents associated with island-arc volcanism is less well-known than that of
vents on mid-ocean ridges and back-arc spreading centres, despite island-arcs extending over a total distance
of 22,000 km in the oceans [50,51]. Island-arc hydrothermal vents in the Pacific and Caribbean exhibit
contrasting vent fluid geochemistry to hydrothermal systems on seafloor spreading centres. Hydrothermal
vents in the Mariana arc produce fluids with low pH, high CO₂, and low H₂S concentrations compared with
“black smoker” vents on mid-ocean ridges [51]. At vents on the Loihi Seamount of the Hawaiian archipelago,
low concentrations of H₂S favour a microbial community dominated by Fe-oxidising bacteria rather than
sulfide-oxidisers [52]. Hydrothermal vents on the Nafanua cone in the crater of Vailulu’u Seamount in the
Samoan archipelago produce fluids with pH 2.7 and droplets of liquid CO₂, and dead midwater metazoans are
abundant on the seafloor around those vents, where an acid-tolerant polynoid polychaete feeds on bacteria
colonising their carcasses [53]. A similar “dead zone” of midwater shrimp has been observed around
hydrothermal vents in the summit crater of Kick’em Jenny submarine volcano in the Lesser Antilles arc of the
Caribbean [54], which also hosts a cold-seep ecosystem on a debris avalanche deposit on its western flank
[55].

The environmental differences between island-arc hydrothermal systems and those on seafloor spreading
centres may result in differences in their associated faunal assemblages, but few faunal assemblages have been
characterised from island-arc hydrothermal vents, and none have been previously described from the Southern
Ocean. Although some species found at ESR hydrothermal vents have also been reported from the Kemp
Caldera, such as the limpet *Lepetodrilus concentricus*, the stalked barnacle *N. scotiaensis*, the seastar *P. tyleri*,
and the sea spiders *Sericosura* spp. [33-35,56], the faunal associated with this island-arc hydrothermal system
has not yet been described in detail. The aims of this study are therefore to define the different biotopes in the
hydrothermally active areas of the caldera, characterise the trophic ecology of the faunal assemblage, and
determine the level of similarity between the fauna of this island-arc vent system and assemblages at
hydrothermal vents on seafloor spreading centres in neighbouring regions.

Formatted: Font: 11 pt, Strikethrough

We hypothesise that the chemosynthetic fauna present in the venting sites of the Kemp Caldera will be similar
to the chemosynthetic fauna of the ESR segments E2 and E9, based on their relatively close distance of only
~440 km and ~90 km away.

3. Materials and Methods

Data for this study were collected during two expeditions at sea: expedition JR224 on board the *Royal*
*Research Ship (RRS) James Clark Ross* in February 2009 [57] and expedition JC042 on board the *RRS James*
*Cook* in January 2010 [58].

3.1 Hydro-acoustic data

Multibeam swath bathymetry surveys of the caldera were conducted by the *RRS James Clark Ross*' hull-
mounted Kongsberg Simrad EM120 multibeam echo sounder during expedition JR224 and by the SIMRAD
SM2000 high-resolution (200 kHz) multibeam echosounder mounted on the ROV *Isis* during expedition
JC042 [27,58,59]. ROV-mounted swath data were processed using the IFREMER software package
CARAIBES [59].

3.2 Water column and hydrothermal fluid sampling

On JC042, water column samples were collected using a titanium frame with 24 externally sprung Niskin
bottles, specifically designed for the sampling of waters with low levels of trace metals and nutrients. The
frame also included a Seabird +911 CTD, a light scattering sensor (LSS) and a reductive potential (Eh)
detector. The 10 L Niskin bottles were Teflon lined, with Teflon taps and non-metallic parts, any metallic
components were constructed using titanium or high- quality stainless steel. Water samples were analysed for
particulate, dissolved and soluble concentrations of metals and oxyanions [29] as well as microbial diversity
(detailed below).

Hydrothermal fluid sampling was achieved using titanium (Ti) samplers on the ROV, equipped with an
Inductively Coupled Link (ICL) high-temperature sensor to ensure the collection of high-quality samples. In
the case of diffuse flow fluid sampling or sampling of friable chimney structures, the Ti samplers were used in
conjunction with a specially-constructed Ti diffuse sampler, which was used to prevent entrainment of
surrounding seawater into the path of the fluid during sampling [27,448].

3.3 Seabed imagery

During expedition JR224, the towed Seabed High-Resolution IMaging Platform (SHRIMP) was deployed
once in the caldera near the Kemp Seamount to collect video imagery of the seabed that enabled
identification of megafaunal (> 3 cm) benthic invertebrates [52]. SHRIMP was equipped with three video only
recording cameras, a forward looking Simrad PAL colour CCD (Charge-coupled device) camera type
OE1364, a downward looking Bowtech PAL colour CCD camera type L3C-550 and a downward looking 3-
chip domesticPanasonic camcorder.

Formatted: Font: 11 pt, Highlight

Formatted: Font: 11 pt, Highlight

Formatted: Font: 11 pt, Highlight

In 2010, during expedition JC042 the ROV *Isis* was deployed for 8 dives with a total of 118 h deployment
time at the Kemp Caldera (Figure 1, Supplement Table 1). For seafloor imagery (photos, video and frame
grabs of videos) ROV *Isis* was equipped with a 3-chip CCD video camera (Insite Pacific Atlas), a 1080i high
definition video camera (Insite Pacific Mini Zeus) on a pan-and-tilt-mount and a 3.3-megapixel stills camera
(Insite Scorpio) [59]. Additionally, two lasers, 0.1 m apart, were mounted parallel to the focal axis of the video
camera to provide scale in images. Footage from the video cameras was recorded on DVCAM tapes and DVD
and from the still camera on a memory card [27].

**3.4 Macro- and megafauna sample treatment**

Benthic invertebrates were collected either by the ROV *Isis*'s suction sampler or scoop and brought to the
surface in ambient seawater. Once on board samples were immediately transferred to seawater in a
temperature controlled lab set to +4°C where individuals were dissected and either frozen or stored in
molecular grade ethanol for molecular analysis, frozen for isotope analysis or fixed in 10% seawater formalin
for morphological analysis.

*Molecular barcoding*

For species identification, genomic DNA was extracted from the muscle tissue of selected invertebrate species
collected in the hydrothermally active biotopes and treated for COI mtDNA amplification and sequencing
using primers LCO 1490 and HCO 2198 [60] as described in Buckeridge et al. [34]. Electropherograms were
visualized and contigs constructed using CodonCode Aligner v3.7.1.1 (CodonCode Corporation 2006).
Quality was evaluated using PHRED scores [61]. BLAST searches of the COI sequences for similarities were
carried out in GenBank. Sequences have been submitted to Genbank (accession numbers xxx – will be added
on acceptance).

*Stable isotopes*

For stable isotope analysis of the food web structure in and outside the venting areas in the Kemp Caldera
different tissues were used for different species: tube feet in the asteroids, tentacles in the anemones and foot
muscle tissue in the vesicomid clam. Pugnogonids were sampled whole while the cocculinid sp. gastropods
were removed from their shells and sampled whole. Faunal tissue samples were freeze dried and ground to a
homogenous powder using a pestle and mortar. Aliquots of fauna were tested for carbonates prior to analysis.
The samples did not effervesce, therefore no acidification was carried out. Dual $\delta^{13}\text{C}$ and $\delta^{15}\text{N}$ were measured
by EA-IRMS using a Roboprep-CN sample preparation module coupled to a Europa Scientific 20-20 IRMS on
1.0 mg of sample. $\delta^{34}\text{S}$ were measured using a SERCON Elemental Analyser coupled to a Europa Scientific
20-20 IRMS on 2 mg of sample with an additional 4 mg of vanadium pentoxide as a catalyst. All analysis was
carried out by Iso-Analytical (Crewe, UK). Stable isotope ratios were expressed in delta (δ) notation as parts
171 per thousand/ permil (‰). An external reference material of freeze dried and ground deep-sea fish white
muscle (*Antimora rostrata*) was also analysed ($\delta^{13}\text{C}$, $n = 28$, $-18.82\text{‰} \pm \text{s.d. } 0.10$; $\delta^{15}\text{N}$, $n = 28$, $13.11\text{‰} \pm \text{s.d.}$

[revised manuscript text omitted]
 characterised by an area of dense “white smoking” sulphide chimneys covered in bacterial
mats (Figure 2L) and emitting fluids with temperatures ranging from 103 to 212°C [48]. Pelagic animals,
including squid or shrimp, swimming through the white smoke were observed to change colour from red to
white and then drop to the seafloor. The only macro- and megafaunal species collected from chimney samples
and observed living on the chimney surfaces is the small vent limpet *Lepetodrilus concentricus* (Fig. 3B). The
shell of limpet specimens collected from these chimneys were covered in anhydrite and the limpets left what
appeared to be “home scars” around their sitting positions on the chimneys.

Basalt next to Great Wall (BGW)

Basalt outcrops, including rocks and boulders, and the soft sediment between them (Figure 2F), next to the
biotope “Great Wall” structure, were characterised by the rare presence of large, infaunal specimens of the
vesicomimid clam *Archivescia* s.l. *puertodeseadoi* (Figure 3E), occasional large, dark red actinostolid anemones
(Figure 3A) and *Sericosura* spp. seaspiders (Figure 3B,C) together with numerous individuals of *L. concentricus*
and a cocculinid limpet (Figure 3C). A single specimen of the stalked barnacle *Neolepas scotiaensis* (Figure 3D)
was seen in this biotope.

Basalt in Diffuse Flow (BDF)

The rigid and continuous basalts on the slope of the resurgent cone were associated with diffuse hydrothermal
flow and hosted an assemblage of abundant macrofauna (Figure 2E). The cocculinid limpet and *Lepetodrilus*
*conentricus* both occurred at very high abundances and both actinostolid anemones and *Sericosura* spp. were
frequent. Occasionally, halichondriid sponges (Figure 2D) were seen and several single specimens of *Neolepas*
*scotiaensis* were present. Towards the outer edges of this biotope very occasional non-vent-associated fauna
such as the sessile holothurians *Psolus* and cnidarians (*Anthomastus*) were recorded.

Clam Road (CR)

The bank just south of the resurgent cone hosted the Clam Road biotope (Figure 2H), characterised by rough-
edged basalt and frequent vesicomimid clams living **epilithically**. The most common species were the cocculinid
limpet, *Lepetodrilus concentricus* and pycnogonids of the genus *Sericosura*, with actinostolid anemones and
halichondriid sponges being occasionally present. Within the finer sediment rubble collected by the suction
sampler, several species of polychaetes belonging to several families were present (Table 1).

Precipitated Sediment (PS)

Near the “Winter Palace” biotope large areas of flat, sedimented seafloor were ~~found to be~~ covered in white
precipitate with the occasional basalt rock penetrating the surface (Figure 2K). A characteristic area for this
biotope is the “Glacier” locality (Figure 1E). *Lepetodrilus concentricus* and *Sericosura* spp. seaspiders were
seen on dense white precipitate, while the vesicomimid clam *Archivescia* s.l. *puertodeseadoi* was frequently seen
burrowed in areas with thinner precipitate cover. Collections in this area revealed the presence of thyasirid
bivalves and several polychaete species. Specimens of the actinostolid anemone and the cocculinid limpet were
less abundant in this biotope and mostly found on the basalt outcrops. The sediments taken, coated in pale
bacterial mats, in this biotope had a strong sulphur odour and contained tubes of *Sclerolinum* (Figure 3H). Some
of the precipitate covered seafloor areas formed “dead zones” in which with numerous dead *Nematocarcinus*
shrimps and squids which were lying on the white precipitate. *Sericosura* spp. seaspiders were seen in clusters
over individual dead shrimps and presumed to be feeding on them.

Coarse Sediment in Diffuse Flow (CSDF)

The top of the bank south of the resurgent cone was influenced by diffuse venting and covered by coarse-grained
anhydrite aggregates and basalt fragments (Figure 2 I), which lead to the name “Ash mount”. The macrofauna
was dominated by very high abundances of *Sericosura* spp. and the cocculinid sp. with the occasional *L.*

Commented [A1]: Is this a real word?

*concentricus* and actinostolid anemone. None of the vesicomid clams, common in the adjacent biotope “Clam
Road”, were present here.

Fine Sediment in Diffuse Flow (FSDF)

This biotope, defined by fine sediments in the influence of the diffuse flow, was characterised by clusters of
burrowing vesicomid clams. Thyasirid bivalves [81] lived infaunally among the vesicomid clams and the
starfish *Paulasterias tyleri* was also observed on the clam beds. Specimens of the shrimp *Nematocarcinus* sp.
were seen walking around on the sediment and occasionally siboglinid worm tubes and ophiuroids could be
seen.

Basalt covered in White Mat (BWM)

The “BWM” biotope is characterised by the presence of halichondriid sponges and a fine white bacterial or
mineral precipitate cover on the basalt (Figure 2 D). In this biotope, few macrofauna were seen apart from
sponges and occasional 7-armed starfish *Paulasterias tyleri*. Towards the border of this zone, near areas
influenced by hydrothermal activity, macrofauna associated with the hydrothermal environment including
*Sericosura* spp., cocculinid limpet. and actinostolid anemones begin to appear, initially rare then in increasing
numbers as the abundance of the halichondriid sponge decreased.

Overall, the Kemp Caldera vent field showed a consistent pattern of faunal zonation, with changing
assemblage types at different biotopes and with increasing distance from the hot vent fluid source (Figure 4).

*4.2.2. Vent biotope associated macro- and megafauna*

In total, 26 benthic species were recovered from the chemosynthetic biotopes in the Kemp Caldera (Table 1),
of which ten taxa were known from non-chemosynthetic environments in the Southern Ocean, and 13 species
have so far been discovered only at hydrothermally active areas.

The demosponges dominating the fauna in the “BWM” biotope and occasionally occurring in diffuse flow
areas were identified as a species belonging to the family Halichondriidae based on the morphology of the
simple spiculae (D. Janussen personal communication).

The actinostolid anemones with dark red tentacles were relatively large with a pedal disc >10 cm diameter. The
tentacles were armed with strong cnidocytes which stung through double-layered nitrile examination gloves
resulting in a numbing, tingling sensation (CN. Roterman, personal communication). This species occurred in
six of the biotopes but highest abundances were seen in “BDF” and “CR”.

The small vent limpet *Lepetodrilus concentricus* occurred at six of the eight hydrothermal biotopes in the Kemp
Caldera as well as on the natural whale fall [80], but was missing in the toxic environment of Great Wall and in
the outermost area influenced by diffuse flow, the “BWM” biotope. Molecular analysis of the barcoding gene
region of COI showed this species to be conspecific with *L. concentricus* reported from the ESR vents [33,82].

The cocculinid limpet was the visually most abundant species in the hydrothermal biotopes being found in six
of them and also on the natural whale fall. BLAST search resulted on COI sequences place this species into the
Cocculinidae with best matches (85% similarity) to *Cocculina subcompressa* GQ160744 and *Cocculina* sp.
L429 AB238591. The radula morphology was similar to but not in complete agreement with the genera
*Cococrater* and *Coccopigya* [83,84]

The large vesicomid clam was present in six biotopes of the Kemp Caldera, showing two different lifestyle
modes; one group was seen buried in the soft sediments while the other was seen living epilithically on basalt.
Morphologically, based on shell and soft part characteristics, the specimens resemble *Archivesica* s.l.
*puertodeseadoi* (Signorelli & Pastorino, 2015) (E. Krylova personal communication). BLAST searches of the
COI sequences from Kemp Caldera clam specimens against sequences in GenBank resulted in *Calyptogena* sp
TO-2011 AB634285, as specimen now assigned to *Archivesica* s.l. *fortunata* [26], *Laubericoncha chuni* isolate
Lc4 JN563828 and *Calyptogena laubieri* ssp. ‘CM-2 M’ AB479083 as best matches with ~94% similarity. Like
*Archivesica* s.l. *fortunata*, *Laubericoncha chuni* and *Calyptogena laubieri* have been assigned to *Archivesica*
386 s.l. in the most recent phylogenetic study [26]. To determine the presence of endosymbionts in the gill tissue,
clone libraries were constructed for 16S rDNA and resulted in 16 sequences from epifaunal and 115 sequences
from infaunal specimens. The sequences analysis showed that only one single symbiotic species of
Gammaproteobacteria from the SUP05 cluster is present in the vesicomid clams in the Kemp Caldera. The
phylogenetic identification in the arb search indicated a species of Oceanospirillales, an endosymbiont in
vesicomid clams off Florida, as the closest relative (Figure 5).

In the precipitated and diffuse flow fine sediment areas, a few specimens of thyasarid bivalves were found and
identified as *Spinaxinus caldarium* and *Parathyasira* cf. *dearborni* [50]. While *P.* cf. *dearborni* did not host
symbiotic bacteria, *S. caldarium* hosted endosymbiotic bacteria in their gills which were of the same phylotype
as symbiotic bacteria in *Spinaxinus emicatus* from the Gulf of Mexico [50]. Siboglinid polychaetes matching
the tube morphology of *Sclerolinum contortum* were found in sulphidic sediments at Precipitated Sediment (PS)
(Fig 3H). Specimens of the maldanid *Nicomache lokii* were found at “PS”, “CR” and “FSDF”.

The stalked neolepadid barnacles occasionally found on the basalt of the diffuse flow areas were *Neolepas*
*scotiaensis*, a species described from the ESR vents at segment E2 and E9 and from the Kemp Caldera [34]. As
only three barnacle specimens were collected in Kemp and used for taxonomy, no material was available for
stable isotope analysis.

The medium sized pycnogonids present in the Kemp hydrothermal biotopes belonged to three species of the
vent-affiliated genus *Sericosura* [38]. Molecular COI sequence analysis showed that *Sericosura bamberti*, *S.*
*curva* and *S. dimorpha* were closely related to each other and form a sister clade to *Sericosura venticola* from
the East Pacific Rise [56].

The multi-armed (7- to 8-armed) forcipulatacean starfish reported from clam fields and basalts in low diffuse
flow areas belong to the recently described *Paulasterias tyleri* [35]. Only four specimens of *Paulasterias tyleri*
were collected in Kemp and used for taxonomic identification.

414 **4.3 Faunal similarity of the vent assemblage at Kemp Caldera with vent fields in neighbouring regions**

Multivariate analyses of presence/absence data for 159 macrofaunal and megafaunal taxa endemic to
chemosynthetic environments from 16 vent fields in the Southern, Indian, and Atlantic Oceans (Figure 6A)
show that the vent fauna at Kemp Caldera is most similar to the faunal assemblages at the E2 and E9
hydrothermal vent fields on the East Scotia Ridge (ESR), but with a lower degree of similarity than that found
between those two ESR vent fields (Figure 6B). The vent fauna of Kemp Caldera was most similar to that
recorded at E9, showing a 44% Sørensen similarity, followed by E2 with a 32% Sørensen similarity, while E2
and E9 exhibit 85% Sørensen similarity. At a regional scale, vent fields within each ocean (Southern, Indian,
Atlantic) are more similar to each other in faunal composition than to those in other oceans (Figure 6B, C).

**4.4 Macro- and megafauna of non-venting environments**

*Pelagic fauna*

A high abundance of pelagic fauna, especially nekton, was observed in the water column of the caldera and
some seemed to follow the ROV *Isis*, presumably attracted to the lights of the UW camera systems. Several
species of megafaunal pelagic crustaceans were present, e.g. the Antarctic krill *Euphausia superba*, dark red
mysids and the benthic-pelagic shrimp *Nematocarcinus lanceopes*, as well as individuals of large jellyfish,
similar to *Poralia* (D. Lindsay personal communication), and comb jellies. In near-bottom waters different fish
species were present. Once a large toothfish *Dissostichus mawsoni* was seen, while often specimens of two
different macrourid species (rattails), a muraenolepid and the paralepidid *Notolepis annulata* were observed in
non-venting areas. Three morphotypes of squid were reported; specimens of a mid-sized (~40 cm) red coloured
species, identified as *Alluroteuthis antarcticus* (L. Allcock personal communication), and a smaller (~20 cm)
white species, identified as *Slosarczykovia circumantarctica*, while specimens of red-spotted squid similar to cf.
*Moroteuthis knipovitchi* were also seen but less frequently.

*Benthic fauna*

[revised manuscript text omitted]

(60m above the chimney) at Great Wall showed that Gammaproteobacteria make up 95% of the bacterial
community (Table 3). Within the Gammaproteobacteria, the majority of sequences (57% of total bacterial
sequences) were assigned to the SUP05 cluster (Figure 5), with high sequence similarity (>98%) to uncultured
free-living bacteria found in other vent systems and moderate sequence similarity (95-97%) to vesicomid
symbionts. Archaeal sequences were split into two groups - Marine Group I (Crenarchaeota), making up 72%
of the sequences and Thermoplasmatales (Euryarchaeota) (28%).

**4.6 Trophodynamics**

The mean $\delta^{13}\text{C}$ values of hydrothermal vent macrofauna covered a range of 19.04‰ with the minimum value
of -40.93‰ (± 0.28 standard deviation [SD]) observed in the Halichondriidae sp. sampled at BWM and the
maximum value of -21.89‰ (± 2.25 SD) in *S. bamberi* found in CSDF (Table 3). The large vesicomid clam,
*Archivesica* s.l. *puertodeseadoi*, was noticeably depleted in ^{12}C relative to other hydrothermal macrofauna
with $\delta^{13}\text{C}$ values of -35.61‰ (± 0.36 SD) at FSDF and -35.20‰ (± 0.38 SD) at CR. Mean $\delta^{15}\text{N}$ values ranged
from -6.47‰ (± 0.52 SD) in *A.s.l. puertodeseadoi* from FSDF to 8.64‰ (± 1.73 SD) in *S. bamberi* found in
CSDF. *Archivesica* s.l. *puertodeseadoi* was the only species with negative $\delta^{15}\text{N}$ values with all other
hydrothermal vent macrofauna having values greater than 2.81‰, which was found in the Terebellidae spp.
from CR. The minimum mean $\delta^{34}\text{S}$ value was observed in the Halichondriidae sp. at BWM (3.53‰ ± 0.25 SD)
while the maximum value was observed in Actinostolidae sp. (15.05‰ ± 1.51 SD) in the BDF biotope. The
ranges of stable isotope values of the non-vent fauna in the Kemp Caldera were much narrower than those
within chemosynthetic habitats: $\delta^{13}\text{C}$ from -26.95‰ (± 0.81 SD) in the echinoid to -22.29‰ (± 0.27 SD) in the
ophiuroid *Ophiolimna antarctica*; $\delta^{15}\text{N}$ from 7.25‰ (± 0.08 SD) *Bathyploetes* sp. to 14.99‰ in *Odontaster*
*penicillatus*; and for $\delta^{34}\text{S}$ the minimum value observed was 16.03‰ in the Macrouridae sp. and the maximum
was 18.93‰ in the octocoral *Anthomastus* sp.

**5. Discussion**

The investigation of the Kemp Caldera resulted in the discoveries of a resurgent cone with hydrothermally active
venting areas and a natural whale fall (characterised by Amon et al. [80]) and of a wide range of marine species,
distributed over several biotopes with distinctly different communities and environmental characteristics. The
“dead zone” of pelagic animal carcasses on the seafloor in the “Precipitated sediment” biotope is similar to that
reported in island-arc vents in the Pacific [53] and Caribbean [54], and native sulphur deposits found at Kemp
Caldera are also a common feature of sites in the Mariana and Kermadec arcs [86]. The most abundant
megafaunal taxa in hydrothermally active venting areas of Kemp Caldera were a large vesicomid clam living
both infaunally and epibenthically, cocculinid and lepetodrilid limpets, the seaspiders *Sericosura* spp., and
actinostolid anemones. The overall biomass at Kemp Caldera’s hydrothermal active areas appeared lower when
compared with the ESR vent fields [27, 36].

The vent biotopes and assemblages at the Kemp Caldera differed from those of the E2 and E9 vent fields on the
nearby East Scotia Ridge (ESR): although seven species are shared between Kemp and the ESR vent fields

(Table 4), the stalked barnacle *N. scotiaensis* and seastar *P. tyleri* were rare at Kemp, and several species that
are abundant at ESR vents were not observed at Kemp Caldera, including the yeti crab *K. tyleri* and the
gastropods *G. chessoia*, *Bruceiella indurata* [87], and *Provanna cooki* [88]. Although Kemp Caldera is
geographically closest to the E9 vent field (90 km distance), the vent fauna at Kemp exhibit a 44% Sørensen
Index of similarity with the vent fauna at E9, whereas E2 and E9 exhibit 85% Sørensen similarity between them
despite being 440 km apart (Figure 6). This difference is consistent with island-arc vent assemblages being
distinct from those on seafloor spreading centres in the same region, and here may result from differences in
vent fluid geochemistry, seafloor type, and depth.

Hydrothermal venting at Kemp Caldera was characterised by “white smoker” chimneys releasing fluids at
temperatures between 103 to 212°C [58], with wide-ranging diffuse flow in basalt and sedimented seafloor areas
at 1375 to 1487 m depth. The vent fluid composition differed from the nearby E9 site, located 90 km away in
~2400 m depth and being high in hydrogen sulphide and iron concentrations, with evidence for an input of
magmatic gas [29,48]. In contrast, the E2 and E9 vent fields, the latter around 440 km away in ~2600 m depth,
were characterised by pillow basalts, sheet lavas and “black smoker” chimneys emitting vent fluids 351 to 383°C
in temperature [27], with a composition to that reported from other basalt-hosted vent sites on seafloor spreading
centres [30].

The microbial composition in the buoyant plume of the Kemp Caldera venting chimneys was different from the
microbial communities reported from the buoyant vent plumes of the black smoking chimneys from the ESR
segments E2 and E9 [27]. At Kemp Caldera, the bacterial community was with 95% highly dominated by
Gammaproteobacteria, particularly of the SUP05 cluster which represented 57% of the total community,
compared to the ESR sites E2 and E9, where represented 70% and 66% of the bacterial communities, with
Gammaproteobacteria making up 58% and 55% of the Proteobacteria and SUP05 contributing only 9% and
19% to the Gammaproteobacteria. The Gammaproteobacteria at the ESR sites were dominated by bacterial
symbionts of vent fauna. [27]. Furthermore, Alphaproteobacteria of the SAR11 clade were common at ESR
(16% and 21% of the total community), but rare (1% of total community) at the Kemp Caldera. Members of
the SUP05 cluster are known to be sulphur oxidisers [89-91]. Analysis of water samples from CTD cast collected
over “Great Wall” in the Kemp Caldera showed sulphur-rich fluids (DP. Connelly personnel communication).
With regard to carbon fixation, both the presence of both the *cbbM* gene [89] and its gene product, ribulose 1,5-
bisphosphate carboxylase/oxygenase (RuBisCo) [90], have been shown in SUP05 communities in other
hydrothermal vent settings. This implies that the SUP05 bacteria may be the dominant microbial primary
producers at Kemp Caldera.

Differences in substrate may also contribute to differences in the vent faunal assemblage at Kemp Caldera
compared with the E2 and E9 vent fields on the nearby ESR. The lack of soft sediment in the immediate vicinity
of the chimneys at E2 and E9 on the ESR may preclude the presence of the vesicomid bivalves, and the sulphur
and anhydrite composition of the brittle chimney structures at Kemp Caldera, may limit the available surfaces
for the growth of *N. scotiaensis* barnacles. Differences in geochemistry between the ESR and Kemp may also

Commented [A2]: this clause appears to refer to E9 given the sentence structure. Rewrite if that is not case.

Formatted: Highlight

Formatted: Highlight

Formatted: Highlight

Commented [A3]: at least in the water column

Formatted: Highlight

[revised manuscript text omitted]

Gammaproteobacteria, which appears to be the main primary producer supporting the hydrothermal vent
macrofauna sampled. The unique geochemical nature of the Kemp Caldera may mean **Epsilon**proteobacteria do
not have the required ecological niche above the surface to contribute strongly to the carbon production
sustaining the hydrothermal vent macrofaunal community. This potentially makes the Kemp Caldera different
from other arc volcano and caldera habitats.

Formatted: Highlight

The pelagic fauna within the Kemp Caldera appeared to be high in abundance but no pelagic sampling was
carried out to verify the subjective observations, to compare with abundance data from adjacent areas like the
South Sandwich Islands, and to test if the pelagic fauna was utilising chemosynthetic primary production.
Studies of the potential influence of hydrothermal vent plumes on biological communities in the water column
is a neglected field [118]. Hydrothermal vent plumes are enriched with abundant and active communities of
chemosynthetic prokaryotes [e.g. 119-122] as well as particulate organic matter [121]. This enhanced supply
of particulate organic matter has the potential to act as a food source for deep-water zooplankton communities
and therefore may exert an influence on pelagic food webs in the proximity of hydrothermal vents and their
plumes. Acoustic and net sampling studies over the Endeavour Segment of the Juan ~~De~~-de Fuca Ridge, NE
Pacific, have consistently demonstrated the enhancement of deep-scattering layers overlying vent plumes at
1,900 – 2,100m [e.g. 123-127]. These scattering layers are predominantly formed by deep-living copepods
(e.g. *Neocalanus* sp.) but also comprised chaetognaths, other crustaceans and gelatinous zooplankton.
Zooplankton in the vicinity of the vent plumes have been demonstrated to have unusual $\delta^{15}\text{N}$ ratios consistent
with feeding on production from the vent plume, as well as sinking organic matter and that upwelling from the
seafloor [128]. It is likely in this locality that zooplankton migration and the release of eggs and larvae that
move upwards in the water column are enhancing production in the region from deep waters to the epipelagic
zone. Zooplankton production over the vents may be double or triple that of the background ocean [127]. High
abundances of gelatinous zooplankton have been observed in the vicinity of vents in the south west Pacific and
in the Okinawa Trough [e.g. 129,130]. Large pelagic predators have also been seen within hydrothermal
plumes in the shallow waters of Kavachi volcanic crater in the Solomon Islands [129]. Observations from
hydrophones on the Juan ~~De~~-de Fuca Ridge also suggest that blue and fin whale may be feeding in the vicinity
of the hydrothermal plumes because of the enhanced zooplankton production [131,132]. Incidental
observations from other localities also hint at connectivity between vent plumes and wider ocean food webs.
Larvae of vent endemic megafauna may also be enriched in vent plumes providing another food source for
pelagic organisms. However, although larvae have been sampled from vent plumes [e.g. 133,134] numbers
have been small and understanding of the larval flux from vents to the pelagic ecosystem remains poorly
defined [135]. Larger predators, like whales, dying and decaying near vent sites might also provide a food
source and stepping stone for some vent organisms [136]. The presence of the natural whale fall in the Kemp
Caldera is an example of specialised species being shared between chemosynthetic food sources [80, this
study].

6. Conclusion

The Kemp Caldera at the southern end of the volcanic South Sandwich Arc hosts a unique hydrothermal
ecosystem with different biotopes and a composition of macro- and megafaunal species not found anywhere
else to date. The chemosynthetic communities include species like the bivalve *Spinaxinus caldarium* and an
undescribed cocculinid limpet, which are as yet specific to this location while some species are shared with the
East Scotia Ridge vent fields or even vent fields further away. Biogeographically the Kemp Caldera vent field
belonged to the Southern Ocean vent province. This discovery of active hydrothermal venting sites with
associated ~~vivid~~ chemosynthetic communities in relatively shallow bathyal waters of the Southern Ocean
may suggest the presence of further hydrothermal ecosystems on the unexplored seamount and submarine
caldera and crater sites of the South Sandwich Arc.

**Acknowledgments**

We thank the masters and crews of *RRS James Clark Ross* and *RRS James Cook* and the staff of the UK
National Marine Facilities at NOC, especially the ROV *Isis* team, and the science teams onboard for logistic,
technical and shipboard support during JR224 and JC042. We are grateful to NERC for funding the ChEsSo
Consortium Grant (NE/DO1249X/1).

**Ethical Statement**

All necessary permits were obtained for the described field studies. Studies in the East Scotia Sea were
undertaken under the permit S3-3/2009 issued by the Foreign and Commonwealth Office, London to section 3
of the Antarctic Act 1994.

**Funding Statement**

The ChEsSo research programme was funded by a NERC Consortium Grant (NE/DO1249X/1) and supported
by the Census of Marine Life and the Sloan Foundation all of which are gratefully acknowledged. We also
acknowledge the NSF grant ANT-0739675 (TS) and NERC PhD studentships NE/D01429X/1(LM, CNR), and
NE/F010664/1 (WDKR). The funders had no role in the study design, data collection and analysis, decision to
publish, or preparation of the manuscript.

**Data Accessibility**

DNA sequences: Genbank accessions MK736312-MK736351 (<http://dx.doi.org/xxxxx>)

The datasets supporting this article have been uploaded as part of the Supplementary Material.

**Contributions**

KL, JC, ADR, WDKR and KZ drafted the manuscript, KL, JTC, VAIH, AGCG, LM, WDKR and KZ
prepared figures and tables, CC, AC, DPC, JTC, ADRG, AGCG, JH, VAIH, KL, RDL, LM, DP,
NVCP, WDKR, CNR, CJS, PAT, KZ were involved in fieldwork, analysis and manuscript edits. All
authors gave final approval for publication.

**Competing Interests**

The authors declare that they do not have any financial, personal, or professional interests that could be
construed to have influenced their paper.

References

1. Ross, JC. 1847. A Voyage of Discovery and Research in the Southern and Antarctic Regions, During the Years 1839–43. London: John Murray **1**, 216–218.
2. LeMasurier WE, Thomson JW, Baker PE, Kyle PR, Rowley PD, Smellie JL, Verwoerd WJ. (eds) 1990. Volcanoes of the Antarctic Plate and Southern Oceans. *Ant Res Ser* **48**. DOI:10.1029/AR048
3. Convey P, Lewis Smith RI. 2006. Geothermal bryophyte habitats in the South Sandwich Islands, maritime Antarctic. *J Veg Sci* **17**, 529–538.
4. Civile D, Lodolo E, Vuan A, Loreto MF. 2012. Tectonics of the Scotia–Antarctica plate boundary constrained from seismic and seismological data. *Tectonophysics* **550–553**, 17–34.
5. Blankenship DD, Brozona RB, Behrendt JC, Finn CA. 1993. Active volcanism beneath the West Antarctic ice sheet and implications for ice-sheet stability. *Nature*, 361, 526–529
6. German CR, Livermore RA, Baker ET, Bruguier NI, Connelly DP, Cunningham AP, Morris P, Rouse IP, Statham PJ, Tyler PA. 2000. Hydrothermal plumes above the East Scotia Ridge: an isolated high-latitude back-arc spreading centre. *Earth Planet Sci Lett* **184**, 241–250. [https://doi.org/10.1016/s0012-821x\(00\)00319-8](https://doi.org/10.1016/s0012-821x(00)00319-8)
7. Leat PT, Livermore RA, Millar IL, Pearce JA. P. T. 2000. Magma Supply in Back-arc Spreading Centre Segment E2, East Scotia Ridge. *J Petrol* **41**, 845–866. <https://doi.org/10.1093/ptrology/41.6.845>
8. De Vries M van Wyk, Bingham RG, Hein AS. 2018. A new volcanic province: an inventory of subglacial volcanoes in West Antarctica. In: Siegert M J, Jamieson SSR, White DA (eds). *Exploration of Subsurface Antarctica: Uncovering Past Changes and Modern Processes. Geological Society, London, Special Publications*, **461**, 231–248.
9. Gallardo VA, Castillo JG. 1968. Mass mortality in the benthic infauna of Port Foster resulting from the eruptions in Deception Island, (South Shetland Islands). *Publ Instituto Antartico Chileno* **16**, 1–11.
10. Lovell LL, Trego KD. 2003. The epibenthic megafaunal and benthic infaunal invertebrates of Port Foster, Deception Island (South Shetland Islands Antarctica). *Deep-Sea Res PT II* **50**, 1799–1819.
11. Barnes DKA, Linse K, Enderlein P, Smale D, Fraser KPP, Brown M. 2008. Marine richness and gradients at Deception Island, Antarctica. *Ant Sci* **20**, 271–279.
12. Bohrmann G, Chin C, Petersen S, Sahling H, Schwarz-Schampera U, Greinert J, Lammers S, Rehder G, Daehlmann A, Wallmann K, et al. 1999. Hydrothermal activity at Hook Ridge in the Central Bransfield Basin, Antarctica. *Geo Marine Letters* **18**, 277–284.
13. Dählmann A, Wallmann K, Sahling H, Sarthou G, Bohrmann G, Petersen S, Chin CS, Klinkhammer, GP. 2001. Hot vents in an ice-cold ocean: indications for phase separation at the southernmost area of hydrothermal activity, Bransfield Strait, Antarctica. *Earth Planet Sc Lett* **193**, 381–394.
14. Klinkhammer GP, Chin CS, Keller RA, Dahlmann A, Sahling H, Sarthou G, Petersen S, Smith F, Wilson C. 2001. Discovery of new hydrothermal vent sites in Bransfeld Strait, Antarctica. *Earth Planet Sci Lett* **193**, - 395-407.
15. Petersen S, Herzig PM, Schwarz-Schampera U, Hannington MD, Jonasson IR. 2004. Hydrothermal precipitates associated with bimodal volcanism in the Central Bransfield Strait, Antarctica. *Miner Deposita* **39**, 358–379.
16. Aquilina A, Connelly DP, Copley JT, Green DRH, Hawkes JA, Hepburn LE, Huvenne VAI, Marsh L, Mills RA, Tyler PA. 2013. Geochemical and Visual Indicators of Hydrothermal Fluid Flow through a Sediment-Hosted Volcanic Ridge in the Central Bransfield Basin (Antarctica). *PLoS ONE* **8**, e54686. doi:10.1371/journal.pone.0054686
17. Bell JB, Woulds C, Brown LE, Sweeting CJ, Reid WDK, Little CTS, Glover AG. 2016a. Macrofaunal Ecology of Sedimented Hydrothermal Vents in the Bransfield Strait, Antarctica. *Front Mar Sci* **3**, 32. doi: 10.3389/fmars.2016.00032
18. Domack E, Ishman S, Leventer A, Sylva S, Willmott V, Huber, B. 2005. A chemotrophic ecosystem found beneath Antarctic ice shelf. *EOS* **86**, 269-276.
19. Niemann H, Fischer D, Graffe D, Knittel K, Montiel A, Heilmayer O, Nöthen K, Pape T, Kasten S, Bohrmann G et al. 2009. Biogeochemistry of a low-activity cold seep in the Larsen B area, western Weddell Sea, Antarctica. *Biogeosciences* **6**, 2383–2395.
20. Gutt J, Barratt I, Domack E, d'Udekem d'Acoz C, Dimmler W, Grémare A, Heilmayer O, Isla E, Janussen D, Jorgensen E, Kock K-H, Lehnert LS, López-

- González P, Langner S, Linse K, Manjón-Cabeza ME, Meißner M, Montiel A, Raes M, Robert H, Rose A, Sañé Schepisi E, Saucède T, Scheidat M, Schenke H-W, Seiler J, Smith C. 2011. Biodiversity change after climate-induced ice-shelf collapse in the Antarctic. *Deep-Sea Res II* **58**, 74-83.
21. Römer M, Torres M, Kasten S, Kuhn G, Graham AGC, Mau S, Little CTS, Linse K, Pape T, Geprägs P, et al. 2014. First evidence of widespread methane emissions around a Sub-Antarctic island (South Georgia). *Earth Planet Sci Lett* **403**, 166-177. DOI: 10.1016/j.epsl.2014.06.036
22. Bell JB, Aquilina A, Wouds C, Glover AG, Little CTS, Reid WDK, Hepburn LE, Newton J, Mills RA. 2016. Geochemistry, faunal composition and trophic structure in reducing sediments on the southwest South Georgia margin. *R Soc Open Sci*, **3**, 160284. <http://dx.doi.org/10.1098/rsos.160284>
23. Sahling H, Wallmann K, Dähmann A, Schmaljohann R, Petersen S. 2005. The physicochemical habitat of *Sclerolinum* sp. at Hook Ridge hydrothermal vent, Bransfield Strait, Antarctica. *Limn Oceanogr* **50**, 598-606.
24. Georgieva MN, Wiklund H, Bell JB, Eilertsen MH, Mills RA, Little CTS, Glover AG. 2015. A chemosynthetic weed: the tubeworm *Sclerolinum contortum* is a bipolar, cosmopolitan species. *BMC Evol Biol* **15**, 280.
25. Eilertsen ME, Georgieva MN, Kongsrud JA, Linse K, Wiklund H, Glover AG, Rapp HT. 2018. Genetic connectivity from the Arctic to the Antarctic: *Sclerolinum contortum* and *Nicomache lokii* (Annelida) are both widespread in reducing environments. *Sci Rep* **8**, 4810.
26. Johnson SB, Krylova EM, Audzijonyte A, Sahling H, Vrijenhoek RC. 2017 Phylogeny and origins of chemosynthetic vesicomyid clams. *Syst Biodiv* **15**, 346-360. doi.org/10.1080/14772000.2016.1252438
27. Rogers AD, Tyler PA, Connelly DP, Copley JT, James R, Larter RD, Linse K, Mills RA, Naveiro-Garabato AN, Pancost RD, et al. 2012. The discovery of new deep-sea hydrothermal vent communities in the Southern Ocean and implications for biogeography. *PLoS Biology* **10**, e1001234.
28. Hahm D, Baker ET, Rhee TS, Won Y-J, Resing JA, Lupton JE, Lee W-K, Kim M, Park S-H. 2015. First hydrothermal discoveries on the Australian-Antarctic Ridge: discharge sites, plume chemistry, and vent organisms. *Geochem Geophys Geosy* **16**, 3061-3075.
29. Hawkes JA, Connelly DP, Rijkenberg MJA, and E. P. Achterberg EP. 2014. The importance of shallow hydrothermal island arc systems in ocean biogeochemistry. *Geophys Res Lett* **41**, 942-947, doi:10.1002/2013GL058817.
- 30 James R, Green DRH, Stock MJ, Alker BJ, Banerjee NR, Cole C, German CR, Huvenne VAI, Powell AM, Connelly DP. 2014. Composition of hydrothermal fluids and mineralogy of associated chimney material on the East Scotia Ridge back-arc spreading centre. *Geochim Cosmochim Acta* **139**, 47-71. doi.org/10.1016/j.gca.2014.04.024
31. Thatje S, Marsh L, Roterman CN, Mavrogordato MN, Linse K. 2015a. Adaptations to Hydrothermal Vent Life in *Kiwa tyleri*, a New Species of Yeti Crab from the East Scotia Ridge, Antarctica. *PLoS One* **10**, e0127621. doi.org/10.1371/journal.pone.0127621
32. Chen C, Linse K, Roterman CN, Copley JT, Rogers AD. 2015. A new Genus of large hydrothermal vent-endemic gastropod (Neomphalina: Peltospiridae) with two new species showing evidence of recent demographic expansion. *Zool J Linn Soc* **175**, 319-335.
33. Linse K, Rotermann CN, Chen C. 2019 A new vent limpet in the genus *Lepetodrilus* (Gastropoda: Lepetodrilidae) from Southern Ocean hydrothermal vent fields showing high phenotypic plasticity. *Frontiers in Marine Science* doi: 10.3389/fmars.2019.00381
34. Buckeridge J, Linse K, Jackson J. 2013. *Vulcanolepas scotiaensis* sp. nov., a new deep-sea scalpelliform barnacle (Eolepadidae: Neolepadinae) from hydrothermal vents in the Scotia Sea, Antarctica. *Zootaxa* **3745**, 551-568.
35. Mah C, Linse K, Copley JT, Marsh L, Rogers A, Clague D, Foltz D. 2015. Description of a New Family, including New Genera and Species of deep-sea Forcipulatacea (Asteroidea), the first known sea star from Hydrothermal Vent Settings. *Zool J Linn Soc* **174**, 93-113.36.
- Marsh L, Copley J, Huvenne V, Linse K, Reid WDK, Rogers A, Sweeting CJ, Tyler PA. 2012. Microdistribution of faunal assemblages at deep-sea hydrothermal vents in the Southern Ocean. *PLoS One* **7**, e48348
37. Reid WDK, Sweeting CJ, Wigham BD, Zwirgmaier K, Hawkes JA, McGill RAR, Linse K, Polunin NVC. 2013. Spatial differences in East Scotia Ridge hydrothermal vent food webs: influences of chemistry, microbiology and predation on trophodynamics. *PLoS One* **9**, e65553.
38. Zwirgmaier K, Reid WDK, Heywood J, Sweeting CJ, Wigham BD, Polunin NVC, Hawkes JA, Connelly DP,

- Pearce D, Linse K. 2014. Linking regional variation of epibiotic bacterial diversity and trophic ecology in a new species of Kiwaidae (Decapoda, Anomura) from East Scotia Ridge (Antarctica) hydrothermal vents. *Microbiol Open*, doi: 10.1002/mb03.227
39. Heywood J, Chen C, Pearce P, Linse K. 2017. Symbionts composition in the Southern Ocean vent gastropod *Gigantopelta chessoia*: indication for inter-generational transfer? *Polar Biol* **40**, 2335-2342. doi: 10.1007/s00300-017-2148-6
40. Chen C, Uematsu K, Linse K, Sigwart JD. 2017. By more ways than one: Rapid convergence in adaptations to hydrothermal vents shown by 3D anatomical reconstruction of *Gigantopelta* (Mollusca: Neomphalina). *BMC Evol Biol* **17**:62
41. Lee S-H, Lee W-K, Won Y-J. 2016. A new species of yeti crab, genus *Kiwa* Macpajerson, Jones and Segonac, 2005 (Decapoda: Anomura: Kiwaidae), from a hydrothermal vent on the Australian-Antarctic Ridge. *J Crust Biol* **36**, 238-247.
42. Leat PT, Fretwell PT, Tate AJ, Larter RD, Martin TJ, Smellie JL, Jokat W, Bohrmann G. 2016a. Bathymetry and geological setting of the South Sandwich Islands volcanic arc. *Ant Sci* **28**, 293-303. <https://doi.org/10.1017/S0954102016000043>
43. Leat PT, Fretwell PT, Tate AJ, Larter RD, Martin TJ, Smellie JL, Jokat W, Bohrmann G. 2016b (Table 1) Datasets used in the South Sandwich arc bathymetric compilation. *PANGAEA* <https://doi.org/10.1594/PANGAEA.868492>, (Supplement to Leat et al. 2016a)
44. Larter RD, Vanneste LE, Morris P, Smyth DK. 2003. Structure and tectonic evolution of the South Sandwich arc. In Larter RD & Leat, PT, eds. Intra-oceanic subduction systems: tectonic and magmatic processes. *Special Publication of the Geological Society of London*, 219, 255-284.
45. Leat PT, Day SJ, Tate AJ, Martin TJ, Owen MJ, Tappin DR. 2013. Volcanic evolution of the South Sandwich volcanic arc, South Atlantic, from multibeam bathymetry. *J Volcan Geotherm Res*, **265**, 60-77.
46. Tyler PA. 2012. JC055 cruise report www.bodc.ac.uk.
47. Bohrmann G. 2013. The Expedition of the Research Vessel "Polarstern" to the Antarctic in 2013 (ANT-XXIX/4). *Rep Polar Mar Res* **668**, 1-174.
48. Cole CS, James RH, Connelly DP, Hathorne EC. 2014. Rare earth elements as indicators of hydrothermal processes within the East Scotia subduction zone system. *Geochem Cosmo Acta* **140**, 20-38.
49. Larter R. 2009. https://www.bodc.ac.uk/data/information_and_inventories/cruise_inventory/report/9359/
50. Rogers AD. 2010. Cruise report JC042. www.bodc.ac.uk
51. Marsh L, Copley JT, Huvonne, VAI, Tyler PA, the Isis ROV Facility. 2013. Getting the bigger picture: Using precision Remotely Operated Vehicle (ROV) videography to acquire high-definition mosaic images of newly discovered hydrothermal vents in the Southern Ocean. *Deep-Sea Res PT I* **137**, 124-135.
52. Folmer o, Black M, Hoeh W, Lutz R, Vrijenhoek R. 1994. DNA primers for amplification of mitochondrial cytochrome c oxidase subunit I from diverse metazoan invertebrates. *Molecular Mar Biol Biotechn* **3**, 294-299.
53. Ewing B, Green P. 1998. Base-calling of automated sequencer traces using phred. II. Error probabilities. *Genome Res* **8**, 186-194. doi: 10.1101/gr.8.3.175
54. Copley JT, Marsh L, Glover AG, Hühnerbach V, Nye VE, Reid WDK, Sweeting CJ, Wigham BD, Wiklund H. 2016. Ecology and biogeography of megafauna and macrofauna at the first known deep-sea hydrothermal vents on the ultraslow-spreading Southwest Indian Ridge. *Sci Rep* **6**, 38158.
55. Zhang D, Zhou Y, Wang C, Rouse GW. 2017. A new species of *Ophryotrocha* (Annelida, Eunicida, Dorvilleidae) from hydrothermal vents on the Southwest Indian Ridge. *ZooKeys* **687**, 1-9.
56. Chen C, Zhou Y, Wang C, Copley JT. 2018. Two new hot-vent peltospirid snails (Gastropoda: Neomphalina) from Longqi hydrothermal field, Southwest Indian Ridge. *Frontiers Mar Sci* **4**, 392.
57. Watanabe HK, Chen C, Marie DP, Takai K, Fujikura K, Chan BKK. 2018. Phylogeography of hydrothermal vent stalked barnacles: a new species fills a gap in the Indian Ocean 'dispersal corridor' hypothesis. *R Soc Open Sci* **5**, 172408.
58. Zhou Y, Zhang D, Zhang R, Liu Z, Tao C, Lu B, Sun D, Xu P, Lin R, Wang J, Wang C. 2018. Characterization of vent fauna at three hydrothermal vent fields on the Southwest Indian Ridge: Implications for biogeography and interannual dynamics on slow-spreading ridges. *Deep-Sea Res PT I* **137**, 1-12.
59. Lane DJ. 1991. 16S/23S rRNA sequencing. *E Stackebrandt and M Goodfellow (ed) Nucleic acid techniques in bacterial systematics John Wiley & Sons, New York*: 115-147
60. Stackebrand E, Liesack W. 1993. Nucleic acids and classification. *Goodfellow, M and O'Donnell, AG (ed), Handbook of new bacterial systematics, Academic Press, London*: 152-189

[revised manuscript text omitted]

- chemoautotroph from the SUP05 clade of marine bacteria that produces nitrite and consumes ammonium. *ISME J* **11**, 273-271.
92. Thatje S, Smith KE, Marsh L, Tyler PA. 2015b. Evidence for abbreviated and lecithotrophic larval development in the yeti crab *Kiwa tyleri* from hydrothermal vents of the East Scotia Ridge, Southern Ocean. *Sex Early Devel Aquatic Org* **1**, 109-116.
93. Marsh L, Copley JT, Tyler PA, Thatje S. 2015. In hot and cold water: differential life-history traits are key to success in contrasting thermal deep-sea environments. *J Anim Ecol*, **84**, 898-913.
<http://doi.org/10.1111/1365-2656.12337>
94. Johnson SB, Waren A, Vrijenhoek RC. 2008. DNA barcoding of *Lepetodrilus* limpets reveals cryptic species. *J Shellfish Res* **27**, 43-51.
95. Ingels J, Aronson RB, Smith CR. 2018 The scientific response to Antarctic ice-shelf loss. *Nat Clim Change* **8**, DOI: 10.1038/s41558-018-0290-y
96. Amon DJ, Wiklund H, Dahlgren TG, Copley JT, Smith CR, Jamieson AJ, Glover AG. 2014. Molecular taxonomy of *Osedax* (Annelida: Siboglinidae) in the Southern Ocean. *Zool Scr* **43**, 405-417
doi:10.1111/zsc.12057
97. Linse K, Jackson J, Maljutina M, Brandt A. 2014. Herbivorous northern hemisphere *Jaera* (Crustacea, Isopoda, Janiridae) found on whale bones in the SO deep sea: Description of *Jaera tyleri* sp. nov. *PLoSOne* **9**, e93018
98. Brandt A, Gooday AJ, Brandao SN, Brix S, Brökeland W, Cedhagen T, Choudhury M, Cornelius N, Danis B, De Mesel I, et al. 2007. The Southern Ocean deep sea: first insights into biodiversity and biogeography. *Nature* **447**, 307-311.
99. De Broyer C, Koubbi P, Griffiths HJ, Raymond B, Udekem d'Acoz C d', Van de Putte AP, Danis B, David B, Grant S, Gutt J, et al. 2014. Biogeographic Atlas of the Southern Ocean. *Scientific Committee on Antarctic Research*, Cambridge, XII + 498 pp.
100. Saiz-Salinas JI, Ramos A, Garcia FJ, Troncoso JS, San Martin G, Sanz C, Palaci C. 1997. Quantitative analysis of macrobenthic softbottom assemblages in South Shetland waters (Antarctica). *Polar Biol* **17**, 393-400.
101. Linse K, Brandt A, Bohn J, Danis B, De Broyer C, Heterier V, Hilbig B, Janussen D, López González PJ, Schüller M, et al. 2007. Macro- and megabenthic communities in the abyssal Weddell Sea (Southern Ocean). *Deep-Sea Res PT II* **54**, 1848-1863. doi: 10.1016/j.dsr2.2007.07.011
102. Griffiths HJ, Linse K, Barnes DKA. 2008. Distribution of macrobenthic taxa across the Scotia Arc, Antarctica. *Ant Sci* **20**, 213-226.
103. Linse K, Griffiths HJ, Barnes DKA, Brandt A, Davey N, David B, DeGrave S, d'Udekem d'Acoz C, Eleaume M, Glover A, et al. 2013. The macro- and megabenthic fauna on the continental shelf of the eastern Amundsen Sea, Antarctica. *Cont Shelf Res* **68**, 80-90.
104. López-González PJ, Williams GC. 2011. A new deep-sea pennatulacean (Anthozoa: Octocorallia: Chunellidae) from the Porcupine Abyssal Plain (NE Atlantic). *Helgoland Mar Res* **65**, 309-318. doi.org/10.1007/s10152-010-0224-1
105. Diaz A, Feral J-P, David B, Saucedo T, Poulin E. 2011. Evolutionary pathways among shallow and deep-sea echinoids of the genus *Sterechinus* in the Southern Ocean. *Deep-Sea Res PT II* **58**, 205-211.
doi.org/10.1016/j.dsr2.2010.10.012
106. Nye V, Copley J, Linse K. 2013. A new species of *Eualus* Thallwitz, 1891 and new record of *Lebbeus antarcticus* (Hale, 1941) (Crustacea: Decapoda: Caridea: Hippolytidae) from the Scotia Sea. *Deep-Sea Res PT II* **92**, 145-156.
107. Vacelet J, Fiala-Medioni A, Fisher CR, Boury-Esnault N. 1996. Symbiosis between methane-oxidising bacteria and a deep-sea carnivorous cladorhizid sponge. *Marine Ecology Progress Series* **145**, 77-85.
108. Thurber AR, Kröger K, Neira C, Wiklund H, Levin LA. 2010. Stable isotope signatures and methane use by New Zealand cold seep benthos. *Marine Geology* **272**, 260-269.
109. Yamanaka T, Shimamura S, Nagashio H, Yamagami S, Onishi Y, Hyodo A, Mampuku M, Mizota C. 2015. A compilation of the stable isotopic compositions of carbon, nitrogen and sulphur in soft body parts of animals collected from deep-sea hydrothermal vent and methane seep fields: variations in energy source and importance of subsurface microbial processes in the sediment-hosted systems. In Ishibashi J., Okino K., Sunamura M (eds) *Subseafloor Biosphere Linked to Hydrothermal Systems*, Springer, Tokyo.
110. Goffredi SK, Johnson S, Tunnicliffe V, Caress D, Clague D, Escobar E, Lundsten L, Paduan JB, Rouse G, Salcedo DL, Soto LA, Spelz-Madero R, Zierenberg R, Vrijenhoek R. 2017. Hydrothermal vent fields discovered in the southern Gulf of California clarify role of habitat in augmenting regional diversity. *Proc Biol Sci*. 284(1859). pii: 20170817. doi: 10.1098/rspb.2017.0817.
111. Rau GH. 1981. Hydrothermal vent clam and tube worm 13C/12C: further evidence of nonphotosynthetic food sources. *Science* **213**, 338-340.

[revised manuscript text omitted]

1863.
doi.org/10.1098/rspb.2017.1281

Figure & Table legends

Figure 1. Map of Southern Ocean; A) Location of Kemp Caldera in Southern Ocean; B) Bathymetric West to East through the Kemp Caldera; C) Bathymetric West to East through the Kemp Caldera; D) Kemp Seamount and Kemp Caldera showing locality of cone; E) Sampling sites in the vicinity of cone in Kemp Caldera

Figure 2. In situ - Kemp Caldera biotopes; A) Ferrous sediments, 59°41.86S 28°21.53W; B) Non-vent soft sediment with ophiuroids and holothuroids, 59°41.83S 28°21.52W; C) Non-vent basalt with alcyonaceans and brisingid, 59°41.82S 28°21.51W; D) Vent halo zone with halichondriids, 59°41.64S 28°21.13W; E) Basalt in diffuse flow with actinostolid anemones, pygnoconid *Sericosura* and cocculinid limpet, 59°41.67S 28°21.10W; F) Basalt next to Great Wall with actinostolids, *Sericosura*, cocculinids and vesicomid clams *Archivesica* s.l. *puertodeseadoi*, 59°41.69S 28°21.10W; G) Fine sediment in diffuse flow with *Archivesica* s.l. *puertodeseadoi*, *Sericosura*, and cocculinids, 59°41.66S 28°20.99W, H) Clam Road with *Archivesica* s.l. *puertodeseadoi* and cocculinids, 59°42.03S 28°21.23W; I) Coarse sediment in diffuse flow with cocculinids, *Sericosura* and actinostolids, 59°42.05S 28°21.22W; J) Great Wall, sulphur structure with white bacterial mats, 59°41.68S 28°21.09W; K) Precipitated sediment with cocculinids and *Lepetodrilus concentricus*, 59°41.67S 28°21.14W; L) Winter Palace, chimneys covered in anhydrite and with *Lepetodrilus concentricus*, 59°41.69S 28°20.97W. The white bar is approximate 10 cm.

Figure 3. In situ – Kemp chemosynthetic fauna; A) Actinostolid sp., Cocculinid sp., Clam Road, 59°42.05S 28°21.23W; B) *Sericosura bambergi* (yellow arrow), Cocculinid sp., *Lepetodrilus concentricus*. Basalt next to Great Wall, 59°41.67S 28°21.09W; C) Cocculinid sp., *Sericosura bambergi* (yellow arrow) Coarse sediment in Diffuse Flow 59°42.06S 28°21.24W; D) *Neolepas scotiaensis* (yellow arrow) Basalt in Diffuse Flow, 59°42.66S 28°20.98W; E) Buried-living *Archivesica* s.l. *puertodeseadoi*., siphons (yellow arrow), Fine sediment in diffuse flow, 59°41.67S 28°21.01W; F) Epilithically-living *Archivesica* s.l. *puertodeseadoi*., siphons (yellow arrow), Clam Road, 59°42.02S 28°21.23W ; G) *Paulasterias tyleri*, Fine sediment in diffuse flow, 59°41.71S 28°21.07W; H) *Sclerolinum* sp., Precipitated sediment, 59°41.68S 28°20.99W.

Figure 4. Idealised schematic of the spatial distribution of the Kemp vent field faunal assemblages with increasing distance from sulphur vent or vent fluid exit.

Figure 5. Phylogenetic position of the Kemp 16S rRNA sequences (in bold) from the water and the vesicomid endosymbiont within the SUP05 cluster. Sequences were added to the Silva102 guide tree by parsimony. Bootstrap values (only values >50 are shown) were calculated by nearest neighbour interchange within arb.

Figure 6. Comparison of vent faunal composition at Kemp Caldera with 15 well-studied vent fields in neighbouring oceanic regions; red-filled circles represent vent fields in the Indian Ocean (Southwest Indian Ridge and Central Indian Ridge), yellow-filled circles represent vent fields on the Mid-Atlantic Ridge, black-filled circles represent vent fields in the Southern Ocean. A) Location of hydrothermal vent fields included in multivariate analysis of faunal composition; topography shown is from the Global Multi-Resolution Topography (GMRT) synthesis (<http://www.geomapapp.org/>). B) Hierarchical agglomerative clustering using group-average linkage for presence/absence records of “chemosynthetic-environment endemic” macro- and megafaunal taxa (329 records of 159 taxa across 17 vent fields, presented as Supplementary Information). C) Two-dimensional non-metric multidimensional scaling (MDS) plot of Sørensen Index similarity matrix calculated from presence/absence records of “chemosynthetic-environment endemic” macro- and megafaunal taxa, with 5% metric MDS solution to reduce point-collapse.

Table 1. Species presence in biotopes of the Kemp Caldera

Table 2. Mean $\delta^{13}\text{C}$, $\delta^{15}\text{N}$ and $\delta^{34}\text{S}$ values (‰) of hydrothermal vent fauna and non-vent fauna collected from the Kemp Caldera. Standard deviations are in parentheses..

Table 3. Microbial composition based on 16S rDNA clone libraries from the buoyant vent plume at Great Wall

Table 4. Comparison of Kemp Caldera chemosynthetic fauna with other SO hydrothermal fauna

Supplementary information

Suppl. table 1. Summary of ROV Isis deployments in the Kemp Caldera. Tasks, collections and other: Ex = experiment deployment, F = Fauna collected by ROV claw, N = Niskin, S = Suction sampler, Sw = Swath bathymetry, Ti = Titanium sampler, Tr = Trap, T-L = temperature lance, VM = Video mosaic

Suppl. table 2. Presence/absence data for vent taxa compiled from published literature for Kemp Caldera and 15 well-studied vent fields in neighbouring oceanic regions: the Southern Ocean (E2 and E9 vent fields), the Indian Ocean (Longqi, Duanqiao, Tiancheng, Kairei, Edmond, Solitaire, and Dodo fields), and Mid-Atlantic Ridge (Lucky Strike, Rainbow, Broken Spur, TAG, Snake Pit, Ashadze-1, and Logatchev fields), updating the dataset previously published by Copley et al. (2016) with subsequently published records of additional sites and taxa (Zhang et al., 2017; Chen et al., 2018; Watanabe et al., 2018; Zhou et al., 2018).

Table 1. Species presence in biotopes of the Kemp Caldera. Abundance: 0 = absent, 1 = rare, 2 = occasional, 3 = frequent, 4 = common, x = present. *= records only from hydrothermal sites

	Species	WP	GW	PS	BG	BDF	CR	CSD	FSD	BW
Porifera	Halichondriidae sp.*					2	2		2	4
Cnidaria	Actinostolidae sp.*			1	2	3	3	1		1
	Anthomastus sp.					1				
Mollusca										
Gastropoda	Lepetodrilus concentricus *	3		3	4	4	4	2		
	Cocculinidae sp.*			2	4	4	4	4		1
Bivalvia	Archivesica s.l. puertodeseadoi *			3	1		3		2	
	Spinaxinus caldarium *			1					1	
	Parathyasira cf. dearborni			1						
Cephalopoda	Octopoda sp. dead						1			
	Decapodiformes			x						
Annelida										
	Sabellidae sp									1
	Sclerolinum contortum *			2						
	Nicomache lokii *			3			1		2	
	Polynoidae spp						1			
	Terebellidae spp			2			1			
	Spionidae spp			2			1			
	Maldanidae spp						1			
	Amphiomidae spp								1	
	Chaetopteridae spp						1		1	
Crustacea										
Cirripedia	Neolepas scotiaensis *				1	1				
Decapoda	Nematocarcinus lanceopes dead								2	
	Nematocarcinus			x						
Chelicerata										
Pycnogonida	Sericosura bamberi *				2	3	4	4		1
	Sericosura curva *						1			
	Sericosura dimorpha *						1			
	Nymphon cf. longicoxa									1

Echinodermata				
Ophiuroidea	indet			2
	Paulasterias			
Asteroidea	tyleri *	1		1 1
	Sterechinus			
Echinoidea	dentifer		1	
Holothuridea	Psolus sp		1	
Chordata				
	dead Notolepis			
	annulata	x		

Table 2. Mean $\delta^{13}\text{C}$, $\delta^{15}\text{N}$ and $\delta^{34}\text{S}$ (‰) of hydrothermal vent fauna and non-vent fauna collected from the Kemp Caldera. Standard deviations are in parentheses.

Taxon	Tissue	Biotope	N	$\delta^{13}\text{C}$	$\delta^{15}\text{N}$	N	$\delta^{34}\text{S}$
Halichondriidae sp.	Whole	BWM	9	-40.93 (0.28)	5.68 (0.58)	3	3.53 (0.25)
Actinostolidae sp.	Tentacle	BDF	35	-24.59 (0.62)	8.54 (0.52)	13	15.05 (1.51)
Cocculinidae sp.	Whole	BGW	14	-26.80 (2.05)	3.46 (0.64)	12	3.94 (0.73)
Cocculinidae sp.	Whole	CSDF	11	-23.81 (1.59)	6.06 (1.17)	8	7.16 (0.61)
Archivesica s.l. puertodeseadoi	Foot	FSDF	38	-35.61 (0.36)	-6.47 (1.73)	9	4.97 (2.856)
Archivesica s.l. puertodeseadoi	Foot	CR	28	-35.20 (0.38)	-3.24 (1.61)	10	8.79 (0.85)
Maldanidae sp.	Whole	CR	1	-26.98	3.55	0	-
Terebellidae sp.	Whole	CR	1	-27.45	2.81	0	-
Sericosura bamberi	Whole	BGW	8	-24.18 (0.76)	8.59 (0.62)	8	8.15 (1.93)
Sericosura bamberi	Whole	CSDF	16	-21.89 (2.25)	8.64 (0.68)	15	8.75 (2.29)
Anthomastus sp.	Polyps	Non-vent	4	-22.45 (1.35)	9.30 (0.29)	1	18.93
Echinoid sp.	Gonad	Non-vent	3	-26.96 (0.81)	8.97 (0.81)	0	-
Freyella sp.	Tube feet	Non-vent	3	-23.48 (1.41)	9.99 (0.90)	3	18.34 (0.29)
Bathyploetes sp.	Muscle	Non-vent	2	-22.73 (0.24)	7.25 (0.08)	1	18.13
Holothuroidea sp. 1	Muscle	Non-vent	3	-22.44 (0.10)	8.08 (0.67)	3	17.46 (0.45)
Hymenaster sp.	Tube feet	Non-vent	3	-22.27 (0.58)	10.92 (0.48)	0	-
Odontaster penicillatus	Arm	Non-vent	1	-23.33	14.99	0	-
Ophiolimna antarctica	Arm	Non-vent	3	-22.29 (0.27)	9.88 (0.26)	0	-
Psolus sp.	Muscle	Non-vent	2	-22.60 (0.99)	9.07 (0.26)	0	-
Polynoidae	Whole	Non-vent	1	-23.70	11.02	0	-
Nematocarcinus lanceopes	Muscle	Non-vent	4	-24.01 (0.47)	7.33 (0.26)	0	-
Macrouridae sp.	Muscle	Non-vent	1	-24.60	10.93	1	16.03
Notolepis annulata	Muscle	Non-vent	3	-26.00 (0.65)	7.53 (0.14)	2	17.02 (1.08)
Notolepis annulata (dead)	Muscle	BDF	1	-26.09	8.33	1	17.13

Table 3. Microbial composition based on 16S rDNA clone libraries from the buoyant vent plume at Great Wall

	sequences	%
Archaea	47	100
Crenarchaeota - marine group I	34	72
Euryarchaeota - Thermoplasmatales	13	28
Bacteria	87	100
Bacteroidetes - Flammeovirgaceae	1	1
Proteobacteria	86	99
- Alphaproteobacteria - SAR11	1	1
- Deltaproteobacteria - SAR324	2	2
- Gammaproteobacteria	83	95
- - Alteromonadales - Alteromonadaceae	16	18
- - Oceanospirillales - Halomonadaceae	10	11
- - Oceanospirillales - SUP05 cluster	50	57
- - Oceanospirillales - other	4	5
- - Pseudomonadales	3	3

Table 4. Comparison of Kemp Caldera chemosynthetic fauna with other SO hydrothermal fauna

Species	Hydrothermal site							sister taxa
	Kemp	E2	E9	AAR-KR1	BS-HR	Larsen-B	Kemp-WF	
Cnidaria								
Anthozoa	Actinostolid sp. 1	+						Pac, Atl
	Actinostolid sp. 2		+					Pac, Atl
	Actinostolid sp. 3			+				Pac, Atl
Annelida								
Polychaeta	Sclerolinum contortum	(+)				+		Pac Atl
	Nicomanche lokii	+	+					N-Atl
Mollusca								
Gastropoda	Gigantopelta chessoia		+	+				SWIR
	Lepetodrilus concentricus	+	+	+			+	Pac, Atl, Ind
	Cocculinid sp.	+					+	Pac
	Provanna cooki		+	+				Pac, Atl
	Bruceiella indurata	+	+	+				Pac
Bivalvia	Spinaxinus caldarium	+						N-Pac, N-Atl
	Parathyasira cf. dearborni	+						SO
	Archivesica s.l. puertodeseadoi	+				+		Pac, Atl
Arthropoda								
Cirripedia	Neolepas scotiaensis	+	+	+				Pac
Decapoda	Kiwa tyleri		+	+				Pac, SWIR
	Kiwa aranae				+			Pac, SWIR
Pycnogonida	Sericosura bamberi	+	+	+			+	Pac, Atl
	Sericosura curva	+	+	+			+	Pac, Atl
	Sericosura dimorpha	+	+	+			+	Pac, Atl
Echinodermata								
Asteroidea	Paulasterias tyleri	+	+	+	+			Pac

Supplement table 1

Dive ISIS	Date	Length hours	Start			End			Tasks	
			Lat S	Long W	Depth (m)	Lat S	Long W	Depth (m)	Collections	other
05-06.02.2010	23.5	59°41.747	28°21.460	1325	59°41.680	28°21.147	1355	F, N, S, Ti	Tr, Sw, T-L
07-08.02.2010	25	59°41.671	28°21.089	1434	59°41.671	28°21.089	1355	F, N, S, Ti	VM
08-09.02.2010	16	59°41.674	28°21.030	1294	59°41.698	28°20.976	1430	F, N, S, Ti	Sw, T-L
09-09.02.2010	12	59°41.722	28°21.108	1354	59°41.693	28°20.961	1438	F, N, S, Ti	Sw, T-L
10-10.02.2010	14	59°42.026	28°21.161	1441	59°41.687	28°20.998	1441	F, N, S, Ti
10-11.2.2010	14	59°41.995	28°20.952	1456	59°41.607	28°21.137	1421	F, N, S, Ti	T-L
11-11.2.2010	9.5	59°41.971	28°21.030	1420	59°41.697	28°20.974	1435	F, S, Ti	Sw
12-13.2.2010	4	59°41.696	28°21.003	1405	59°41.585	28°21.163	1456	Tr	Ex

Appendix D

Dear Dr Frielaender, Dr Padian and Mrs Parkhouse in your roles as Editors and Editorial Coordinator of Royal Society Open Science,

Please receive our revision of the accepted Manuscript ID RSOS-191501 entitled "Fauna of the Kemp Caldera and its upper bathyal hydrothermal vents (South Sandwich Arc, Antarctica)", with a track-changes and cleaned manuscript version as well as below our specific and detailed responses to the reviewers' comments and suggestions.

We are pleased that you have been able to accept the revised manuscript for publication.

With kind regards in the name of all authors,

Katrin Linse

Subject Editor Comments to Authors:

However, the reviewers do note that there is still considerable work to be done editorially speaking to clean up the language. I urge the authors to be diligent in this regard and please ensure that the final submission has been copy edited thoroughly.

We edited wording, grammar and misspellings in the manuscript and have addressed Reviewer 2's comments further below, while Reviewer 1 had no further comments to act on.

We removed the section "Molecular barcoding" (L 157-164) as the barcodes are now part of two recently submitted, independent manuscripts: one describing the cocculinid gastropod species from the Kemp Caldera and the other discussing the position of the Kemp vesicomylid clam in the wider context of the Pliocardiinae.

In line with removing the molecular barcoding section, we edited the paragraphs on the cocculinid and the vesicomylid clam.

L 378 – 382: The cocculinid limpet was the visually most abundant species in the hydrothermal biotopes being found in six of them and also on the natural whale fall. The gill morphology suggests affinity to Cocculinidae within the order Cocculinida [83,84].

L 384 – 397: The large vesicomylid clam was present in six biotopes of the Kemp Caldera, showing two different lifestyle modes; one group was seen buried in the soft sediments while the other was seen living epilithically on basalt. Morphologically, based on shell and soft part characteristics, the specimens resemble *Archivesica* s.l. *puertodeseadoi* (Signorelli & Pastorino, 2015) (E. Krylova personal communication). To determine the presence of endosymbionts in the gill tissue, clone libraries were constructed for 16S rDNA and resulted in 16 sequences from epifaunal and 115 sequences from infaunal specimens. The sequences analysis showed that only one single symbiotic species of Gammaproteobacteria from the SUP05 cluster is present in the vesicomylid clams in the

Kemp Caldera. The phylogenetic identification in the arb search indicated a species of Oceanospirillales, an endosymbiont in vesicomid clams off Florida, as the closest relative (Figure 5).

Authors revision comments to the Reviewers' Comments:

Reviewer: 2

Comments to the Author(s)

Review of Linse et al. Version 2

This manuscript is notably improved by the revisions. Below are a few comments. It needs a complete re-read for the small errors, still. (For example, the first line of the title - is that really what you want?) I put line numbers on a converted Word document where you can also find smaller edits – including many highlighted trivial mistakes. I'm sure you can find more :)

We thank the reviewer for their positive comments and especially acknowledge their thorough editing, even indicating errors to us in their pdf version.

We deleted the redundant “the” from the title as well as the unnecessary “the” at the start of the title..

I. 14 in Abstract: Multivar analysis doesn't show 'how' (=mechanism) – just shows that it 'is'.

We removed the “how” and edited the sentences from “..., but shows how the species composition...” to “..., showing that the species composition ...”

L 61: Corrected “is border” to “is bordered”.

L 63: We edited the sentence and removed “ones”.

L 66: We added the space between “... study. The Kemp ...”

L 99 and 4 further places: We corrected “sea spider” by adding the missing space between the two words.

L 111: We corrected “Hydro-acoustic”.

L 137: We corrected “forward”.

L 139: We removed the extra space between “chip” and “domestic” and added a space between “domestic” and “Panasonic”.

L 212: We corrected “depth”.

I. 287: Just an aside: I wonder if the scars mean that this species is a suspension feeder or even hosts an epibiont as does *Lepetodrilus fucensis*.

We thank the reviewer for their comment and will investigate this further but not in the context of this manuscript.

L 315: To our knowledge, “epilithically” is a real word. For example it is often used describing the seaweeds, for example in Coppejans et al 2011 *Phycologia* 50 (4): 403-412.

L 321: We removed “found to be” to simplify the sentence.

L375f: We accepted the suggestion and changed the text to “... of the barcode gene region of COI...”.

I. 407: *Sericosura venticola* does not occur on the East Pacific Rise (it's at Endeavour, JdF ridge). Ref 57 listed here did not give me any assurance....

We have changed the sentence to “... to *Sericosura venticola* from North Pacific vents [56].” Reference 56 (which got accidentally cut from R1 when the track changes were accepted) is: Arango CP, Linse K. 2015. New *Sericosura* (Pycnogonida: Ammotheidae) from deep-sea hydrothermal vents in the Southern Ocean. *Zootaxa* 3995, 037-050. This publication reports *Sericosura venticola* from the East Pacific Rise based on data provided by the lead author. The COI sequence of *Sericosura venticola* is Genbank number DQ390080 and originally was published in Arango & Wheeler 2007 *Cladistics* 23: 255-293 and its locality in this publication is given as North Pacific vents.

L 455: We followed the suggestion and included asteroids after brisingid to be more precise: “... with brisingid asteroids and other ...”

L458: We followed the suggestion and included ophiuroids after ophiacanthids be more precise: “... by ophiacanthid ophiuroids, particularly ...”

L463: We corrected the spelling of metalliferous.

L546ff: We restructured the sentence to clarify that the high hydrogen sulphide and iron concentrations were measured in Kemp Caldera vent fluids: “The vent fluid composition differed from the nearby E9 site, which is located 90 km away in ~2400 m depth, being high in hydrogen sulphide and iron concentrations and with evidence for an input of magmatic gas [29,48].”

L550: We edited the sentence to “..., with a fluid composition similar to that reported from other basalt-hosted ...”.

L 557: We added the missing word “Proteobacteria” : “..., where Proteobacteria represented 70% and 66% of the bacterial communities, ...”

L559. We added the missing full stop.

L560. We deleted the extra full stop.

L 561: We deleted the extra full stop and square bracket.

L 564: We changed the position of “both” following the reviewers suggestion.

L 567: We followed the reviewers suggestion on including a water column remark: “This implies that the SUP05 bacteria may be the dominant microbial primary producers in the water column above the venting sites at Kemp Caldera.”

L 573: We corrected the spelling of *N. scotiaensis*.

L 578. We deleted the extra full stop and replaced [Amon] with the correct reference number [80].

L 602: We corrected “dorvilleid”.

L 610: We corrected the sentence by replacing “fauna” with “that”.

L 622: We corrected “Demospongiae”.

L 625: We deleted an extra “-” and added a missing “~”.

L 639: We replace “, which” with “that”.

I. 647-652: I do not believe you can interpret seafloor producers based on water/plume samples. Please see Gregory Dick’s recent review of Microbiomes of deep-sea hydrothermal vents, in which he comments:

Plumes are composed primarily of organisms derived from the water column, such as SUP05 (Gammaproteobacteria), SAR324 (Deltaproteobacteria), SAR11 (Alphaproteobacteria) and Marine Group I archaea^{97–99}. Sea floor and/or subsurface organisms such as Epsilonproteobacteria can be present in plumes^{98,99} but are quickly diluted owing to the massive entrainment of background seawater²⁸.

Thus, I suggest you remove the water column comment here and just leave the mystery until such time that the seafloor producers are sampled.

We followed the reviewer’s suggestion and deleted the sentences commenting on the water column bacteria contributing to the vent macrofauna food source.

L 669/681: We corrected the spelling of the “Juan de Fuca Ridge”.

Refs 49 thru 56 appear to be missing.

We realised that refs 49-56 were still in the “track changes version” of Revision 1 but were not included in the “clean”, track changes accepted version. We copied the missing references into the current revision. We thank the reviewer for spotting this and will check the track changes accepted version of R2 more closely.

L 699: We deleted “vivid” in the concluding remarks and corrected “chemosynthetic”.

Table 4 caption: Spell out SO. Do we interpret that “Pac” means sister taxa found throughout the Pacific rather than NPac (as indicated) or WPac (Sclerolinum is only, WPac, I think); standardize to region or to ocean. Similarly, Ind vs SWIR.

We spelled out Southern Ocean in the caption. We followed the reviewer’s suggestion and standardized to ocean.

What does the bold text mean?

The bold text is not supposed to be bold and was edited.